# Theories of Relativistic Dissipative Fluid Dynamics

**DOI:** 10.3390/e26030189

**Published:** 2024-02-22

**Authors:** Gabriel S. Rocha, David Wagner, Gabriel S. Denicol, Jorge Noronha, Dirk H. Rischke

**Affiliations:** 1Department of Physics and Astronomy, Vanderbilt University, 1221 Stevenson Center Lane, Nashville, TN 37212, USA; gabriel.soares.rocha@vanderbilt.edu; 2Instituto de Física, Universidade Federal Fluminense, 24210-346 Niterói, RJ, Brazil; gsdenicol@gmail.com; 3Institute for Theoretical Physics, Goethe University, Max-von-Laue-Str. 1, D-60438 Frankfurt am Main, Germany; dwagner@itp.uni-frankfurt.de; 4Department of Physics, West University of Timișoara, Bd. Vasile Pârvan 4, 300223 Timisoara, Romania; 5Illinois Center for Advanced Studies of the Universe, Department of Physics, University of Illinois Urbana-Champaign, Urbana, IL 61801, USA; jn0508@illinois.edu; 6Helmholtz Research Academy Hesse for FAIR, Campus Riedberg, Max-von-Laue-Str. 12, D-60438 Frankfurt am Main, Germany

**Keywords:** fluid dynamics, relativistic, kinetic theory

## Abstract

Relativistic dissipative fluid dynamics finds widespread applications in high-energy nuclear physics and astrophysics. However, formulating a causal and stable theory of relativistic dissipative fluid dynamics is far from trivial; efforts to accomplish this reach back more than 50 years. In this review, we give an overview of the field and attempt a comparative assessment of (at least most of) the theories for relativistic dissipative fluid dynamics proposed until today and used in applications.

## 1. Introduction and Summary

Fluid dynamics is an effective theory for the long-wavelength, low-frequency limit of any given many-body system. The fluid-dynamical equations of motion are based on very general principles: they describe the conservation of energy, momentum, and, possibly, charge quantum numbers in the system. As an effective theory, fluid dynamics is applicable if there is a clear separation between the microscopic scales, which are characteristic of the dynamics of the fluid’s constituents, and the macroscopic scales, which characterize the spatio-temporal variation of the fluid-dynamical variables, i.e., energy density, charge density, and fluid velocity. This scale separation is usually quantified by the so-called *Knudsen number*, the ratio of a typical microscopic to a typical macroscopic time or length scale. If the Knudsen number is sufficiently small, the state of the fluid is near local thermodynamical equilibrium, and, at least if one averages over sufficiently long time scales, dissipative effects can be considered minor corrections.

In non-relativistic applications, Navier–Stokes theory is firmly established as the theory of dissipative fluid dynamics [1]. However, for relativistic systems, the situation is much more complicated. The main problem is that the naive generalization of non-relativistic Navier–Stokes theory to the relativistic case [1,2] is acausal [3,4] and, hence, unstable to small disturbances around the equilibrium state [5]. The reason for this pathological behavior is rooted in the fact that in Navier–Stokes theory, the *dissipative currents*, the bulk viscous pressure, the charge diffusion current, and the shear-stress tensor react instantaneously to the *dissipative forces*, i.e., the gradients of the fluid-dynamical fields. This is unphysical even in the non-relativistic limit, and attempts to incorporate a finite time scale over which dissipative currents react to dissipative forces have been made even in non-relativistic dissipative fluid dynamics [6,7].

In a relativistic setting, such consideration is paramount, as it is required to repair the pathological behavior of (the naive relativistic generalization of) Navier–Stokes theory. Among the first to address this issue were Israel and Stewart [8], who developed a so-called *second-order theory* of relativistic dissipative fluid dynamics. In this theory, dissipative currents relax over a certain nonvanishing characteristic time scale to the values given by the dissipative forces. The qualifier “second-order” originates from the fact that terms which are formally of second order in the Knudsen number need to be included to account for this relaxation process[note 1].

In the past two decades, the development of causal and stable theories of relativistic dissipative fluid dynamics was, to a large extent, motivated by the attempt to describe the collective flow of hot and dense strong-interaction matter created in heavy-ion collisions at relativistic energies (for reviews, see [9,10,11] and refs. therein). These systems are so small that the separation between microscopic and macroscopic length scales is, at best, an order of magnitude in the case of collisions of large nuclei (such as gold or lead) [12,13]. Thus, the separation of scales found in such systems is certainly much smaller than in any macroscopic fluid of our daily-life experience. Nevertheless, experimental data show a surprising amount of collective behavior for the matter created in heavy-ion collisions, to the extent that the underlying collective flow can be quantitatively described using relativistic dissipative fluid dynamics [9,10,11].

Another area where relativistic fluid dynamics plays an important role as a quantitative tool is astrophysics [14]. Both the dynamics of supernovae and the binary merger of neutron stars are currently quantitatively described by relativistic fluid dynamics, coupled to Einstein’s equations of general relativity (if space-time curvature effects are non-negligible) and Maxwell’s equations of electrodynamics (if the mutual interactions between matter and electromagnetic fields are of interest). So far, most applications of fluid dynamics in this field focused on ideal fluid dynamics, and including dissipative effects is a fairly recent development [15,16,17,18].

Nowadays, there exists a plethora of formulations of relativistic dissipative fluid dynamics. This review attempts to provide a concise and comprehensive overview of these various formulations. For each version of relativistic dissipative fluid dynamics, we present a brief derivation of the equations of motion for the dissipative currents and discuss in which aspects they differ from those in other formulations. We admit that the presentation is not entirely balanced, with some formulations receiving more attention than others, which can be attributed to the particular expertise of the authors with some of these approaches, which, in turn, inadvertently receive more attention than others. Furthermore, there are several topics relevant to relativistic fluids that we do not discuss here, such as Lagrangian fluid dynamics [19], the effects of quantum anomalies [20,21,22], or hydrodynamic attractors [23,24]. Another topic which is out of the scope of the present paper is stochastic fluid dynamics [25,26,27,28,29,30,31,32,33,34,35,36,37,38,39,40] and the relationship between dissipation and fluctuations. Indeed, both a systematic understanding of the role of fluctuations and the systematic formulation of a theory of relativistic dissipative stochastic fluid dynamics are at an early stage of development. It is acknowledged that this development may significantly alter the discussions below. Overall, our list of references should be considered exemplary, not exhaustive.

This review is organized as follows. In Section 2, we first introduce the *conserved* quantities (Section 2.1), and discuss the *conservation laws* of fluid dynamics (Section 2.2). The reference point for any theory of dissipative fluid dynamics is usually fluid dynamics in the limit of vanishing dissipation, i.e., *ideal fluid dynamics*, which we present in Section 2.3. The *power counting* in terms of Knudsen and inverse Reynolds number, which guarantees the validity of fluid dynamics, is explained in Section 2.4. Theories of dissipative fluid dynamics are related to ideal fluid dynamics by the so-called *matching conditions*. These are discussed in Section 2.5, using some popular examples. This is followed by a general discussion of the *constitutive relations* for the dissipative currents in Section 2.6. There, a general classification scheme for all theories of dissipative fluid dynamics discussed in this review is introduced, which allows individual dissipative fluid-dynamical theories to be viewed as special cases within this scheme. The section closes with a brief discussion of *causality* and *stability* of relativistic fluid dynamics in Section 2.7.

Section 3 is devoted to the space-time *gradient expansion* as a way to derive dissipative fluid dynamics. Here, one considers the constitutive relations for the dissipative currents to be given by a power series in space-time gradients of the primary fluid-dynamical variables, i.e., temperature, chemical potential, and fluid velocity. We first present the general set-up of the gradient expansion in Section 3.1. We then restrict the consideration to *space-like* gradients and explicitly discuss the first- and second-order versions, i.e., *Navier-Stokes theory* (Section 3.2) and *Burnett theory* (Section 3.3), respectively. Both of these theories suffer from acausality and instability. At first-order, this is remedied in an extension of Navier–Stokes theory, called *BDNK theory*, introduced in Section 3.4, where also *time-like* gradients of the primary fluid-dynamical variables are considered. As does BDNK theory, *BRSSS theory* also features time-like gradients, although they occur only at second order. At face value, BRSSS theory possesses potentially dangerous terms that destroy the hyperbolic nature of the equations of motion, violating causality and generating instabilities. These issues are resolved by replacing those terms—using the constitutive relations at first order—by terms of linear order in the product of gradients of primary fluid-dynamical variables and dissipative currents or of second order in dissipative currents. This modification is then termed “resummed BRSSS theory”, which can be causal and stable and, thus, used in numerical simulations. In essence, this procedure results in equations of motion of relaxation type for the dissipative currents, just as in the second-order theories of Israel and Stewart.

Section 4 deals with the derivation of dissipative fluid dynamics from the *second law of thermodynamics*. This approach has the advantage that fundamental thermodynamic relations, like the Onsager principle, are respected. However, lacking an underlying microscopic theory, clearly, this approach cannot make any definite predictions for the value of the transport coefficients appearing in the constitutive relations for the dissipative currents. In Section 4.1, we first generalize fundamental thermodynamic relations to a manifestly *covariant form*. We then derive first-order Navier–Stokes theory (Section 4.2) and second-order Israel–Stewart theory (Section 4.3) from the second law of thermodynamics. The fact that both acausal and causal theories can be derived from the second law implies that imposing the latter does not necessarily lead to consistent formulations of relativistic fluid dynamics. We close this section with a discussion of *divergence-type theories* (Section 4.4). These dissipative fluid-dynamical theories require the use of a generating function for the dynamical variables, from which all conserved and dissipative currents can be derived. Unfortunately, the explicit form of this generating function is generally not known and must be postulated by an educated guess. Nevertheless, divergence-type theories can lead to potentially hyperbolic formulations that can respect the requirements of causality and stability.

In Section 5, we turn to the derivation of relativistic dissipative fluid dynamics using *kinetic theory* in the form of the *Boltzmann equation*. The advantage of such an approach is that it yields explicit expressions for the transport coefficients of sufficiently dilute systems, encoding information about the interactions of the fluid constituents at the microscopic level. In Section 5.1, we first discuss the Boltzmann equation for the evolution of the single-particle distribution function in phase space and the relationship between the latter and fluid-dynamical quantities. We then introduce the concepts of local thermodynamical equilibrium and matching conditions. In Section 5.2, we then discuss *Chapman-Enskog theory* as a means to derive Navier–Stokes theory. A modified version of Chapman–Enskog theory is introduced in Section 5.3 to derive BDNK theory. Then, we turn to the derivation of *Israel-Stewart theory in the 14-moments approximation* in Section 5.4. This is extended to include higher-order moments of the single-particle distribution function in Section 5.5, where we present theories of *Resummed Transient Fluid Dynamics*. There, we discuss two examples, *DNMR theory*, which truncates the system of moment equations by considering only the slowest eigenmodes of the linearized collision integral as dynamical variables, and *IReD theory*, which replaces higher-order moments by the corresponding dissipative currents.

In Section 6, we discuss *Zubarev’s approach* to derive dissipative fluid dynamics. Here, the underlying microscopic theory can be any quantum field theory, thus dispensing with further assumptions and approximations necessary, for instance, for the applicability of kinetic theory. The transport coefficients in this approach are then derived in the form of Kubo-type relations. We note that the first-order transport coefficients, i.e., the bulk viscosity, the particle diffusion, and the shear viscosity coefficients, can always be defined and computed via Kubo relations [41], and the appearance of such relations is not specific to the Zubarev approach.

Finally, in Section 7, we conclude this review by presenting a selection of further developments: anisotropic fluid dynamics, maximum-entropy fluid dynamics, resistive dissipative magnetohydrodynamics, and spin hydrodynamics. Several appendices provide additional material: the conversion of time-like derivatives into space-like ones using the conservation laws (Appendix A), the construction of the basis of irreducible tensors in momentum space (Appendix B), as well as transport coefficients appearing at second order in Knudsen and inverse Reynolds number in DNMR theory (Appendix C).

### Notations and Conventions

For the metric tensor of flat Minkowski space, we use the “mostly minus” convention, $g_{\mu \nu }=\mathrm{diag}\left(+,-,-,-,\right)$, unless stated otherwise. We also use natural units where $\hbar =k_{B}=c=1$. The scalar product of two 4-vectors $A^{\mu }$ and $B^{\mu }$ is written as $A^{\mu }g_{\mu \nu }B^{\nu }\equiv A^{\mu }B_{\mu }\equiv A\cdot B$. Symmetrization of indices for a rank-2 tensor $A^{\mu \nu }$ is denoted as $A^{\left(\mu \nu \right)}\equiv \frac{1}{2}\left(A^{\mu \nu }+A^{\nu \mu }\right)$, while antisymmetrization is denoted as $A^{\left[\mu \nu \right]}=\frac{1}{2}\left(A^{\mu \nu }-A^{\nu \mu }\right)$. We introduce a time-like normalized 4-vector $u^{\mu }$ (which will later be identified with the fluid velocity), i.e., u·u=1. The comoving derivative with respect to $u^{\mu }$ of a quantity *A*, u·∂A, is denoted either as $u\cdot \partial \;A\equiv \dot{A}$, or as u·∂A≡DA, if this is notationally more convenient. The projector onto the 3-space orthogonal to $u^{\mu }$ is denoted as ${∆}^{\mu \nu }\equiv g^{\mu \nu }-u^{\mu }u^{\nu }$. The projection of a 4-vector $A^{\mu }$ onto the 3-space orthogonal to $u^{\mu }$ is then $A^{\langle \mu \rangle }\equiv {∆}^{\mu \nu }A_{\nu }$. The space-like 3-space gradient of a quantity *A* is denoted as $\nabla^{\mu }A\equiv \partial^{\langle \mu \rangle }A={∆}^{\mu \nu }\partial_{\nu }A$. The symmetric and traceless rank-4 projector onto the 3-space orthogonal to $u^{\mu }$ is ${∆}_{\alpha \beta }^{\mu \nu }\equiv {∆}_{(\alpha }^{\mu }{∆}_{\beta )}^{\nu }-\frac{1}{3}{∆}^{\mu \nu }{∆}_{\alpha \beta }$. The projection of a rank-2 tensor $A^{\mu \nu }$ onto the 3-space orthogonal to $u^{\mu }$ is then defined as $A^{\langle \mu \nu \rangle }\equiv {∆}_{\alpha \beta }^{\mu \nu }A^{\alpha \beta }$.

## 2. Preliminaries

### 2.1. Conserved Quantities

Fluid dynamics is an effective theory for the small-momentum (long-wavelength), low-energy (small-frequency) limit of a given many-body system. In this limit, the system’s degrees of freedom consist of the so-called *hydrodynamic modes*, i.e., the modes without a gap in their excitation spectrum. These degrees of freedom are associated with conserved quantities, like energy, momentum, and conserved charge quantum numbers. For the sake of simplicity, we will only consider a single particle species, so here, any conserved charge quantum number is equivalent to a conserved particle number. In a relativistic theory, this always implies the conservation of the *net* particle number, i.e., the number of particles *minus* the number of antiparticles. The central quantities in fluid dynamics are thus the *(net) particle-number current* $N^{\mu }$ and the *energy-momentum tensor*$T^{\mu \nu }$. We tensor-decompose these quantities with respect to the time-like, normalized 4-vector $u^{\mu }$ introduced above,
(1a)$$\begin{array}{cc} N^{\mu } & =nu^{\mu }+n^{\mu }\;, \end{array}$$
(1b)$$\begin{array}{cc} T^{\mu \nu } & =\varepsilon u^{\mu }u^{\nu }-p{∆}^{\mu \nu }+2h^{(\mu }u^{\nu )}+\pi^{\mu \nu }\;. \end{array}$$
Here, n≡N·u is the *particle-number density* in the local rest frame of the fluid, $n^{\mu }\equiv N^{\langle \mu \rangle }$ is the *particle diffusion current* with respect to $u^{\mu }$, $\varepsilon \equiv T^{\mu \nu }u_{\mu }u_{\nu }$ is the *energy density* in the local rest frame of the fluid, $p\equiv -\frac{1}{3}T^{\mu \nu }{∆}_{\mu \nu }$ is the *isotropic pressure*, $h^{\mu }\equiv T^{\langle \mu \rangle \nu }u_{\nu }$ is the *energy-momentum diffusion current* relative to $u^{\mu }$, and $\pi^{\mu \nu }\equiv T^{\langle \mu \nu \rangle }$ is the *shear-stress tensor*. By construction,
(2)$$n\cdot u=0\;,\;h\cdot u=0\;,\;\pi^{\mu \nu }u_{\nu }=0\;,\;\pi_{\mu }^{\mu }=0\;.$$

### 2.2. Conservation Laws

The conservation of particle number, energy, and momentum is expressed by
(3a)$$\begin{array}{cc} \partial \cdot N & =0\;, \end{array}$$
(3b)$$\begin{array}{cc} \partial_{\mu }T^{\mu \nu } & =0\;. \end{array}$$
Inserting Equations ([Disp-formula FD1a-entropy-26-00189]) and projecting Equation ([Disp-formula FD3b-entropy-26-00189]) onto the directions parallel and orthogonal to $u^{\mu }$, we arrive at the tensor-decomposed form of Equations ([Disp-formula FD3a-entropy-26-00189]),
(4a)$$\begin{array}{cc} \dot{n}+n\theta -n\cdot \dot{u}+\nabla \cdot n & =0\;, \end{array}$$
(4b)$$\begin{array}{cc} \dot{\varepsilon }+\left(\varepsilon +p\right)\theta -2h\cdot \dot{u}+\nabla \cdot h-\pi^{\mu \nu }\sigma_{\mu \nu } & =0\;, \end{array}$$
(4c)$$\begin{array}{cc} \left(\varepsilon +p\right){\dot{u}}^{\mu }-\nabla^{\mu }p+{\dot{h}}^{\langle \mu \rangle }+\frac{4}{3}\theta h^{\mu }+\left(\sigma^{\mu \nu }-\omega^{\mu \nu }\right)h_{\nu }+\partial_{\nu }\pi^{\nu \langle \mu \rangle } & =0\;. \end{array}$$
Here, we used the decomposition
(5)$$\partial_{\mu }u_{\nu }=u_{\mu }{\dot{u}}_{\nu }+\frac{\theta }{3}{∆}_{\mu \nu }+\sigma_{\mu \nu }+\omega_{\mu \nu }\;,$$
where we introduced the *expansion scalar* θ≡∂·u≡∇·u, the *shear tensor* of the fluid, $\sigma_{\mu \nu }\equiv \partial_{\langle \mu }u_{\nu \rangle }$, and the *fluid vorticity*$\omega_{\mu \nu }\equiv \nabla_{[\mu }u_{\nu ]}$. We also exploited Equation (Equation 2).

Equations ([Disp-formula FD3a-entropy-26-00189]) or ([Disp-formula FD4a-entropy-26-00189]), respectively, constitute five equations for 17 independent variables: the Lorentz scalars n,ε,p, the Lorentz 4-vectors $u^{\mu }$, $n^{\mu }$, and $h^{\mu }$ (each having three independent variables), and the rank-2 Lorentz tensor $\pi^{\mu \nu }$ (five independent variables). Thus, the system of conservation laws is not closed. The task is, therefore, to provide twelve additional equations to uniquely determine the solution to Equations ([Disp-formula FD3a-entropy-26-00189]) or ([Disp-formula FD4a-entropy-26-00189]), respectively. There are several ways to achieve this goal. The most simple one is to assume that the fluid is in *local thermodynamical equilibrium*, leading to the theory of *ideal fluid dynamics*, which is discussed in the following Section 2.3. The other ways lead to different theories of *dissipative fluid dynamics* and are discussed in the remainder of this work.

### 2.3. Ideal Fluid Dynamics

As mentioned above, the simplest way to reduce the number of independent variables is to make the drastic assumption that the fluid is in *local thermodynamical equilibrium*, i.e., it is a so-called *ideal* (or *perfect*) *fluid*. Local thermodynamical equilibrium is defined by considering a fluid element at a space-time point $x^{\mu }$ which is, on the one hand, so small that the concept of locality makes sense but, on the other hand, is also so large that it contains sufficiently many particles that interact sufficiently rapidly such that the local energy density ε(x) and the local particle-number density n(x) assume values as given in a state of thermodynamic equilibrium,
(6)$$\varepsilon \equiv \varepsilon_{0}\left(T,\mu \right)\;,\;n\equiv n_{0}\left(T,\mu \right)\;,$$
where T=T(x) is the temperature and μ=μ(x) the chemical potential of the fluid at the space-time point $x^{\mu }$. The assumption of local thermodynamical equilibrium also implies that the isotropic pressure is given by a *thermodynamic equation of state*,
(7)$$p\equiv p_{0}\left(T,\mu \right)\;,$$
i.e., it is no longer an independent variable, as the temperature and chemical potential are already determined by the energy density $\varepsilon_{0}$ and particle number density $n_{0}$ via Equation (Equation 6).

An ideal fluid is *completely* characterized by its local energy density $\varepsilon_{0}\left(x\right)$, local particle-number density $n_{0}\left(x\right)$, and fluid 4-velocity $u^{\mu }\left(x\right)$. This implies that for an ideal fluid, the particle diffusion current, the energy-momentum diffusion current, and the shear-stress tensor vanish, i.e., $n^{\mu }=h^{\mu }=\pi^{\mu \nu }\equiv 0$. Together with the equation of state (Equation 7) these are twelve constraints, which reduce the number of independent variables from 17 to five: $\varepsilon_{0}$, $n_{0}$, and the three independent components of $u^{\mu }$. For an ideal fluid, the particle number 4-current and the energy-momentum tensor, therefore, read
(8a)$$\begin{array}{cc} N_{0}^{\mu } & =n_{0}u^{\mu }\;, \end{array}$$
(8b)$$\begin{array}{cc} T_{0}^{\mu \nu } & =\varepsilon_{0}u^{\mu }u^{\nu }-p_{0}{∆}^{\mu \nu }\;. \end{array}$$
The five conservation Equations ([Disp-formula FD3a-entropy-26-00189]) feature precisely five unknowns ($\varepsilon_{0}$, $n_{0}$, and $u^{\mu }$) and are thus (in principle) uniquely solvable. Consequently, the system ([Disp-formula FD3a-entropy-26-00189]) or ([Disp-formula FD4a-entropy-26-00189]), respectively, is closed.

Instead of $\varepsilon_{0}$ and $n_{0}$, one could also use the temperature *T* and the chemical potential μ as independent variables, or, as we will do throughout this paper, the *thermal potential*
$\alpha_{0}\equiv \mu /T$ and the *inverse temperature*
$\beta_{0}\equiv 1/T$. In the following, $\varepsilon_{0}$, $n_{0}$ (or *T*, μ, or $\alpha_{0}$, $\beta_{0}$), and $u^{\mu }$, will be called *primary fluid-dynamical quantities*. The quantities $n^{\mu }$, $h^{\mu }$, and $\pi^{\mu \nu }$ vanish for an ideal fluid but are nonzero for a *dissipative* fluid, i.e., a fluid whose local state *deviates* from local thermodynamical equilibrium. They are therefore called *dissipative fluid-dynamical quantities* or *dissipative currents*.

### 2.4. Power Counting in Fluid Dynamics—Knudsen and Inverse Reynolds Numbers

The applicability of fluid dynamics requires a clear separation of scales. Fluid dynamics is an effective theory valid in the large-wavelength, small-frequency limit of a given microscopic theory, i.e., on *macroscopic length and time scales*. Let *L* be the macroscopic scale, which is given by a typical time or length scale over which a certain primary fluid-dynamical variable varies, e.g., $L∼{\left[\left(\partial a\right)/a\right]}^{-1}$, with $a=\alpha_{0}$, $\beta_{0}$, or $u^{\mu }$. On the other hand, the microscopic scale, λ can, for example, be the mean free path, $\lambda_{\mathrm{mfp}}$, for dilute gases or the inverse temperature, $\beta_{0}$, for other types of fluids. Then, fluid dynamics is applicable if the so-called *Knudsen number* is sufficiently small,
(9)$$\mathrm{Kn}\equiv \frac{\lambda }{L}\ll 1\;.$$
Indeed, suppose that $\lambda \equiv \lambda_{\mathrm{mfp}}\to 0$. This means that interactions between particles occur at infinitely small distances at infinitely fast rates. Such a fluid instantaneously reaches local equilibrium and is thus described by ideal fluid dynamics, cf. Section 2.3. Consequently, the ideal-fluid limit is reached when Kn→0.

Consider now the general case where, in the presence of dissipation, ε, *n*, and *p* differ from their local-equilibrium values. We then define
(10)$$\varepsilon \equiv \varepsilon_{0}+\delta \varepsilon \;,\;n\equiv n_{0}+\delta n\;,\;p\equiv p_{0}+\Pi \;,$$
where Π is the *bulk viscous pressure*. According to the terminology introduced in Section 2.3, δε, δn, and Π are also *dissipative fluid-dynamical quantities*. Since the dissipative quantities δε, δn, Π, $n^{\mu }$, $h^{\mu }$, and $\pi^{\mu \nu }$ vanish in an ideal fluid, they can, in principle, be written as functions (or functionals) of the Knudsen number, i.e.,
(11)$$\begin{array}{cc} \delta \varepsilon & =f_{\varepsilon }\left(\mathrm{Kn}\right)\;,\;\delta n=f_{n}\left(\mathrm{Kn}\right)\;,\;\Pi =f_{\Pi }\left(\mathrm{Kn}\right)\;,\\ n^{\mu } & =f_{n}^{\mu }\left(\mathrm{Kn}\right)\;,\;h^{\mu }=f_{h}^{\mu }\left(\mathrm{Kn}\right)\;,\;\pi^{\mu \nu }=f_{\pi }^{\mu \nu }\left(\mathrm{Kn}\right)\;, \end{array}$$
with the property that $f_{\varepsilon }\left(x\right)$, $f_{n}\left(x\right)$, $f_{\Pi }\left(x\right)$, $f_{n}^{\mu }\left(x\right)$, $f_{h}^{\mu }\left(x\right)$, $f_{\pi }^{\mu \nu }\left(x\right)\to 0$ as x→0. Relations of the type (Equation 11) are examples of *constitutive relations* for the dissipative quantities. The simplest choice is, schematically,
(12)$$\begin{array}{cc} f_{\varepsilon }\left(\mathrm{Kn}\right) & =c_{\varepsilon }\mathrm{Kn}\;,\;f_{n}\left(\mathrm{Kn}\right)=c_{n}\mathrm{Kn}\;,\;f_{\Pi }\left(\mathrm{Kn}\right)=c_{\Pi }\mathrm{Kn}\;,\\ f_{n}^{\mu }\left(\mathrm{Kn}\right) & =c_{n}^{\mu }\mathrm{Kn}\;,\;f_{h}^{\mu }\left(\mathrm{Kn}\right)=c_{h}^{\mu }\mathrm{Kn}\;,\;f_{\pi }^{\mu \nu }\left(\mathrm{Kn}\right)=c_{\pi }^{\mu \nu }\mathrm{Kn}\;, \end{array}$$
with some constants $c_{\varepsilon }$, $c_{n}$, $c_{\Pi }$, $c_{n}^{\mu }$, $c_{h}^{\mu }$, and $c_{\pi }^{\mu \nu }$ of order one. In this case, the dissipative quantities are of first order in the Knudsen number. As we shall see, this is precisely the structure of Navier–Stokes theory, cf. Section 3.2, where the terms of order one in Knudsen number are represented solely by *space-like* gradients. However, more complicated functions of Kn are also possible, for instance, polynomials of second and higher order in Kn (with the correct Lorentz symmetry). Moreover, besides space-like gradients also time-like gradients may appear, as in BDNK theory, see Section 3.4. This then leads to the space-time *gradient expansion*, cf. Section 3.

However, it turns out that Navier–Stokes theory, and more generally the gradient expansion *in terms of space-like gradients*, truncated at *any order* in Kn, is *acausal and unstable*. A *causal and stable* theory of dissipative fluid dynamics can be obtained either by including *time-like gradients*, as in BDNK theory, or by demanding that the dissipative quantities fulfill *dynamical equations of motion*, cf. Section 2.6. They are thus, in principle, *independent* dynamical variables without a direct relation to the Knudsen number. Therefore, in that case, it is helpful to introduce other quantities to characterize the magnitude of the dissipative quantities compared to the primary fluid-dynamical variables with the same dimension. These quantities were termed *inverse Reynolds numbers* in [42] and are (schematically) defined as[note 2]
(13)Reε−1≡δεε0,Ren−1≡δnβ0p0,ReΠ−1≡Πp0,Ren¯−1≡|nμ|β0p0,Reh−1≡|hμ|ε0,Reπ−1≡|πμν|p0.
The inverse Reynolds numbers, as well as the Knudsen numbers, measure the deviation from equilibrium. Note that we defined Ren−1 and Ren¯−1 with $\beta_{0}p_{0}$ in the denominator, rather than $n_{0}$. The quantity $\beta_{0}p_{0}$ has the same dimension as $n_{0}$, but the latter, being the *net* particle number, can be zero if we consider a system with equal numbers of particles and antiparticles.

In most causal and stable theories of dissipative fluid dynamics, the dynamical equations are of relaxation type; see, e.g., Equation (Equation 22). Depending on the initial values for the dissipative quantities, the corresponding inverse Reynolds numbers can either be of the same order of magnitude as the Knudsen number or of vastly different magnitude. However, on certain characteristic relaxation time scales, the dissipative quantities approach the values given by Navier–Stokes theory, and thus, the associated inverse Reynolds numbers become of the same order as the Knudsen number,
(14)Reε−1,Ren−1,ReΠ−1,Ren¯−1,Reh−1,Reπ−1∼Kn.
For power-counting deviations from local equilibrium or the ideal-fluid limit, both inverse Reynolds and Knudsen numbers are usually taken to be of the same order of magnitude.

We close this section by remarking that there are additional microscopic scales which appear in the presence of thermodynamical fluctuations. Such a scale is, for instance, determined by the inverse temperature, $\beta_{0}$, and exists even in equilibrium, where the Knudsen number vanishes. However, the standard fluid-dynamical approach discussed in this review does not treat fluctuations. The latter are subject of theories of stochastic fluid dynamics [25,26,27,28,29,30,31,32,33,34,35,36,37,38,39,40], which are outside the scope of this review.

### 2.5. Matching Conditions

Fluid dynamics is a valid theory if the deviations from local thermodynamical equilibrium are small; see discussion in Section 2.4. Therefore, for a dissipative fluid, it is helpful to define a fictitious *reference state*, a fluid in local thermodynamical equilibrium characterized by its energy density $\varepsilon_{0}$ and particle-number density $n_{0}$, and expand the other fluid-dynamical quantities around this reference state [8]. The expansion around a local-equilibrium reference state implies that all dissipative quantities are small compared to the primary fluid-dynamical variables, i.e., all inverse Reynolds numbers defined in Equation (Equation 13) are much smaller than unity.

In order to solve the conservation Equations ([Disp-formula FD3a-entropy-26-00189]) in full generality, one requires so-called *constitutive relations* for the 14 quantities δε, δn, Π, $n^{\mu }$, $h^{\mu }$, and $\pi^{\mu \nu }$, see Section 2.6. The five conservation equations then determine the remaining five primary fluid-dynamical quantities $\varepsilon_{0}$, $n_{0}$ (or T,μ, or $\alpha_{0},\beta_{0}$), and $u^{\mu }$.

For some fluid-dynamical theories, the constitutive relations are particularly simple. We give a few examples in the following:(i)**Ideal fluid dynamics:** As already discussed in Section 2.3, for *ideal fluid dynamics*, all dissipative quantities vanish, i.e., their constitutive relations are trivial,
(15)$$\delta \varepsilon =\delta n=\Pi =n^{\mu }=h^{\mu }=\pi^{\mu \nu }=0\;.$$(ii)**Dissipative fluid dynamics in the Landau frame:** For dissipative fluid dynamics, a possible subset of the constitutive relations reads
(16)$$\delta \varepsilon =0\;,\;\delta n=0\;,\;h^{\mu }=0\;.$$The other constitutive relations for $n^{\mu }$, Π, and $\pi^{\mu \nu }$ have to be further specified; see discussion in the following sections. The first two relations (Equation 16) imply that $\varepsilon \equiv \varepsilon_{0}$, $n\equiv n_{0}$, i.e., the fictitious local-equilibrium reference state is chosen in such a way that its energy density and particle-number density match the energy density and the particle-number density of the actual fluid. The conditions listed in Equation (Equation 16) are also referred to as *Landau matching conditions*. The last relation (Equation 16) implies no energy diffusion, i.e., the fluid 4-velocity $u^{\mu }\equiv u_{L}^{\mu }$ is identical to the energy flow. This corresponds to choosing a particular reference frame for the motion of the fluid, which is usually called *Landau frame* [1].Projecting Equation ([Disp-formula FD1b-entropy-26-00189]) onto $u_{L,\nu }$, we derive
(17)$$T_{\;\;\nu }^{\mu }u_{L}^{\nu }=\varepsilon u_{L}^{\mu }\;,$$
i.e., the energy density is the eigenvalue of the energy-momentum tensor corresponding to the fluid 4-velocity $u_{L}^{\mu }$ as the time-like eigenvector of the latter. One can formally define the fluid 4-velocity in the Landau frame via
(18)$$u_{L}^{\mu }=\frac{T^{\mu \nu }u_{L,\nu }}{\sqrt{T^{\alpha \beta }u_{L,\beta }T_{\alpha \gamma }u_{L}^{\gamma }}}\;.$$Note that the latter definition is an implicit equation for $u_{L}^{\mu }$.(iii)**Dissipative fluid dynamics in the Eckart frame:** Another possible subset of constitutive relations reads
(19)$$\delta \varepsilon =0\;,\;\delta n=0\;,\;n^{\mu }=0\;.$$The other constitutive relations for $h^{\mu }$, Π, and $\pi^{\mu \nu }$ have to be further specified.The first two relations (Equation 19) are the same as for the Landau frame, i.e., the fictitious local-equilibrium reference state is again chosen in such a way that its energy density and particle-number density agree with the energy density and the particle-number density of the actual fluid. The conditions of Equation (Equation 19) are also referred to as *Eckart matching conditions*.The last relation (Equation 19) implies no particle diffusion, i.e., the fluid 4-velocity $u^{\mu }\equiv u_{E}^{\mu }$ is identical to the flow of particle number. This particular choice of reference frame is called *Eckart frame* [2]. Solving Equation ([Disp-formula FD1a-entropy-26-00189]) for $u_{E}^{\mu }$, we derive
(20)$$u_{E}^{\mu }\equiv \frac{N^{\mu }}{\sqrt{N\cdot N}}\;.$$

### 2.6. Constitutive Relations—General Considerations

In order to close the system of fluid-dynamical equations of motion, the conservation Equations ([Disp-formula FD3a-entropy-26-00189]) and ([Disp-formula FD4a-entropy-26-00189]), respectively, have to be supplemented by *constitutive relations* for the dissipative quantities δε, δn, Π, $n^{\mu }$, $h^{\mu }$, and $\pi^{\mu \nu }$. In this subsection, we present a general discussion of the form of these relations.

In most theories of dissipative fluid dynamics, the constitutive relation for a dissipative quantity *A* assumes the general form
(21)$$A=A_{1}+A_{2}+O_{3}\;,$$
where A=δε, δn, Π, $n^{\mu }$, $h^{\mu }$, or $\pi^{\mu \nu }$, respectively. The term $A_{1}$ comprises all terms of *first order* in Knudsen number. These are all terms that can be constructed in accordance with Lorentz symmetry from space-time *derivatives* of the *primary fluid-dynamical quantities*. Let us, for the sake of definiteness, take the latter to be $\alpha_{0}$, $\beta_{0}$, and $u^{\mu }$, and denote any one of them generically as *B*. We distinguish time-like derivatives, $\dot{B}\equiv u\cdot \partial B$, and space-like derivatives, $\nabla^{\mu }B\equiv {∆}^{\mu \nu }\partial_{\nu }B$, the nomenclature arising from the fact that $\dot{B}$ (resp. $\nabla^{\mu }B$) is a purely temporal (resp. spatial) derivative in the rest frame of the fluid. Then, the terms $A_{1}$ are general arbitrary linear combinations of terms of the form $\dot{B}$ and $\nabla^{\mu }B$, i.e., they comprise all time-like or space-like derivatives of primary fluid-dynamical quantities constructed in accordance with Lorentz symmetry. Relativistic *Navier-Stokes theory*, see Section 3.2, is a special case where *only* space-like derivatives, $\nabla^{\mu }B$, of primary fluid-dynamical quantities appear, cf. Equations ([Disp-formula FD34a-entropy-26-00189]). A more general case is BDNK theory, see Section 3.4, where also time-like derivatives occur, which ultimately can render this theory causal and stable [43], in contrast to relativistic Navier–Stokes theory, which is acausal and unstable [4,5].

The term $A_{2}$ in Equation (Equation 21) comprises all terms of *second order* in either Knudsen or inverse Reynolds number which can be constructed in accordance with Lorentz symmetry from derivatives of the primary fluid-dynamical quantities (first order in Knudsen number) and the dissipative quantities (first order in inverse Reynolds number). Thus, the second-order terms can be classified as follows:(i)*Linear order* in the *product* of derivatives of primary fluid-dynamical quantities and dissipative quantities, i.e., $∼\dot{B}A$ or $(\nabla^{\mu }B)\;A$;(ii)*Linear order* in the *derivatives* of dissipative quantities, i.e., $∼\dot{A}$ or $\nabla^{\mu }A$;(iii)*Second order* in *derivatives* of primary fluid-dynamical quantities, i.e., $∼\ddot{B}$, ${\dot{B}}^{2}$, $\dot{B}\nabla^{\mu }B$, $\nabla^{\mu }\dot{B}$, $u\cdot \partial (\nabla^{\mu }B)$, $\nabla^{\mu }\nabla^{\nu }B$, or $\left(\nabla^{\mu }B\right)\left(\nabla^{\nu }B\right)$,(iv)*Second order* in dissipative quantities, i.e., $∼A^{2}$.

Terms of type (i) and (ii) are of order O(KnRe−1), terms of type (iii) are of order $O({\mathrm{Kn}}^{2})$, and terms of type (iv) are of order O(Re−2). Terms of third order $O_{3}$ or higher are neglected in this work, but have been studied, for instance, in [44,45,46,47].

The time-like derivatives of type (ii) receive special treatment in the so-called *transient dissipative fluid-dynamical theories*. Putting them onto the left-hand side of Equation (Equation 21), we arrive at
(22)$$\tau_{A}\dot{A}+A=A_{\mathrm{NS}}+{\bar{A}}_{2}+O_{3}\;,$$
where $\tau_{A}$ is a certain characteristic time scale, and where we have restricted the terms $∼A_{1}$ to those appearing in Navier–Stokes theory. This can, at least formally, always be achieved by using the conservation laws ([Disp-formula FD4a-entropy-26-00189]), which allow one to replace a time-like derivative of a primary fluid-dynamical quantity with a space-like derivative plus higher-order terms in Knudsen and inverse Reynolds number, which can be put together with the other terms in ${\bar{A}}_{2}$. (The bar over ${\bar{A}}_{2}$ indicates that the terms $∼\dot{A}$ from the original $A_{2}$ terms have been removed and put on the left-hand side of the equation.) However, such a manipulation may affect the causality and stability of the theory, as shown by the example of BDNK theory.

An equation of the type (Equation 22) is of *relaxation type*, i.e., the dissipative quantity *A* relaxes on the time scale $\tau_{A}$ onto the value given by the right-hand side, i.e., to leading order to its corresponding Navier–Stokes value $A_{\mathrm{NS}}$. Restricting the discussion to Landau matching conditions, an application of Equation (Equation 22) to Π, $n^{\mu }$, and $\pi^{\mu \nu }$ results up to second order in the following set of equations,
(23a)$$\begin{array}{cc} \tau_{\Pi }\dot{\Pi }+\Pi & =-\zeta \theta +J+K+R\;, \end{array}$$
(23b)$$\begin{array}{cc} \tau_{n}{\dot{n}}^{\langle \mu \rangle }+n^{\mu } & =\varkappa I^{\mu }+J^{\mu }+K^{\mu }+R^{\mu }\;, \end{array}$$
(23c)$$\begin{array}{cc} \tau_{\pi }{\dot{\pi }}^{\langle \mu \nu \rangle }+\pi^{\mu \nu } & =2\eta \sigma^{\mu \nu }+J^{\mu \nu }+K^{\mu \nu }+R^{\mu \nu }\;, \end{array}$$
where $I^{\mu }\equiv \nabla^{\mu }\alpha_{0}$. Here $\tau_{\Pi }$, $\tau_{n}$, and $\tau_{\pi }$ are the relaxation times for the bulk viscous pressure, the particle diffusion current, and the shear-stress tensor, respectively. The first terms on the right-hand side are the Navier–Stokes terms; see Section 3.2. The terms J, $J^{\mu }$, and $J^{\mu \nu }$ contain all terms of terms of type (i) and (ii) allowed by Lorentz symmetry,
(24a)$$\begin{array}{cc} J & =-\delta_{\Pi \Pi }\Pi \theta -\ell_{\Pi n}\nabla \cdot n-\tau_{\Pi n}n\cdot F-\lambda_{\Pi n}n\cdot I+\lambda_{\Pi \pi }\pi^{\mu \nu }\sigma_{\mu \nu }\;, \end{array}$$
(24b)$$\begin{array}{cc} J^{\mu } & =-\delta_{nn}n^{\mu }\theta -\ell_{n\Pi }\nabla^{\mu }\Pi +\ell_{n\pi }\nabla_{\nu }\pi^{\langle \mu \rangle \nu }+\tau_{n\Pi }\Pi F^{\mu }-\tau_{n\pi }\pi^{\mu \nu }F_{\nu }\\ \; & \;\;-\lambda_{nn}\sigma^{\mu \nu }n_{\nu }+\lambda_{n\Pi }\Pi I^{\mu }-\lambda_{n\pi }\pi^{\mu \nu }I_{\nu }+\tau_{n\omega }\omega^{\mu \nu }n_{\nu }\;, \end{array}$$
(24c)$$\begin{array}{cc} J^{\mu \nu } & =-\delta_{\pi \pi }\pi^{\mu \nu }\theta +\ell_{\pi n}\nabla^{\langle \mu }n^{\nu \rangle }-\tau_{\pi n}n^{\langle \mu }F^{\nu \rangle }+\lambda_{\pi \Pi }\Pi \sigma^{\mu \nu }\\ \; & \;\;-\lambda_{\pi \pi }\pi^{\lambda \langle \mu }\sigma_{\lambda }^{\nu \rangle }+\lambda_{\pi n}n^{\langle \mu }I^{\nu \rangle }+2\tau_{\pi \omega }\pi_{\lambda }^{\langle \mu }\omega^{\nu \rangle \lambda }\;, \end{array}$$
where $F^{\mu }\equiv \nabla^{\mu }p_{0}$. As shown explicitly in Appendix A, the terms of type (i) involving *time-like* derivatives can be converted into space-like ones using the equations of motion ([Disp-formula FD4a-entropy-26-00189]), plus terms of higher order in Knudsen and/or inverse Reynolds number. This results in terms already listed in Equations ([Disp-formula FD24a-entropy-26-00189]).

The tensors K, $K^{\mu }$, and $K^{\mu \nu }$ contain all terms of type (iii),
(25a)$$\begin{array}{cc} K & ={\tilde{\zeta }}_{1}\omega_{\mu \nu }\omega^{\mu \nu }+{\tilde{\zeta }}_{2}\sigma_{\mu \nu }\sigma^{\mu \nu }+{\tilde{\zeta }}_{3}\theta^{2}+{\tilde{\zeta }}_{4}I\cdot I\\ & +{\tilde{\zeta }}_{5}F\cdot F+{\tilde{\zeta }}_{6}I\cdot F+{\tilde{\zeta }}_{7}\nabla \cdot I+{\tilde{\zeta }}_{8}\nabla \cdot F\;, \end{array}$$
(25b)$$\begin{array}{cc} K^{\mu } & ={\tilde{\varkappa }}_{1}\sigma^{\mu \nu }I_{\nu }+{\tilde{\varkappa }}_{2}\sigma^{\mu \nu }F_{\nu }+{\tilde{\varkappa }}_{3}I^{\mu }\theta +{\tilde{\varkappa }}_{4}F^{\mu }\theta \\ & +{\tilde{\varkappa }}_{5}\omega^{\mu \nu }I_{\nu }+{\tilde{\varkappa }}_{6}{∆}_{\lambda }^{\mu }\partial_{\nu }\sigma^{\lambda \nu }+{\tilde{\varkappa }}_{7}\nabla^{\mu }\theta +{\tilde{\varkappa }}_{8}\omega^{\mu \nu }F_{\nu }\;, \end{array}$$
(25c)$$\begin{array}{cc} K^{\mu \nu } & ={\tilde{\eta }}_{1}\omega_{\lambda }^{\left.\right.\langle \mu }\omega^{\nu \rangle \lambda }+{\tilde{\eta }}_{2}\theta \sigma^{\mu \nu }+{\tilde{\eta }}_{3}\sigma^{\lambda \langle \mu }\sigma_{\lambda }^{\nu \rangle }+{\tilde{\eta }}_{4}\sigma_{\lambda }^{\langle \mu }\omega^{\nu \rangle \lambda }\\ & +{\tilde{\eta }}_{5}I^{\langle \mu }I^{\nu \rangle }+{\tilde{\eta }}_{6}F^{\langle \mu }F^{\nu \rangle }+{\tilde{\eta }}_{7}I^{\langle \mu }F^{\nu \rangle }+{\tilde{\eta }}_{8}\nabla^{\langle \mu }I^{\nu \rangle }+{\tilde{\eta }}_{9}\nabla^{\langle \mu }F^{\nu \rangle }\;. \end{array}$$
As for the J-terms, all terms involving *time-like* derivatives have been converted into space-like ones using the equations of motion ([Disp-formula FD4a-entropy-26-00189]), up to higher-order terms, for details, see Appendix A. Note that the term $∼{\tilde{\varkappa }}_{8}$ was missed in [42]. It is, however, in principle allowed by Lorentz symmetry, although ${\tilde{\varkappa }}_{8}$ may vanish when matching the transport coefficients to an underlying microscopic theory.

The tensors R, $R^{\mu }$, and $R^{\mu \nu }$ consist of all terms of type (iv),
(26a)$$\begin{array}{cc} R & =\varphi_{1}\Pi^{2}+\varphi_{2}n\cdot n+\varphi_{3}\pi_{\mu \nu }\pi^{\mu \nu }\;, \end{array}$$
(26b)$$\begin{array}{cc} R^{\mu } & =\varphi_{4}\pi^{\mu \nu }n_{\nu }+\varphi_{5}\Pi n^{\mu }\;, \end{array}$$
(26c)$$\begin{array}{cc} R^{\mu \nu } & =\varphi_{6}\Pi \pi^{\mu \nu }+\varphi_{7}\pi^{\lambda \langle \mu }\pi_{\lambda }^{\nu \rangle }+\varphi_{8}n^{\langle \mu }n^{\nu \rangle }\;. \end{array}$$

The values of the transport coefficients appearing in these equations,
δΠΠ,ℓΠn,ℓnΠ,τΠn,τnΠ,λΠn,λΠπ,δnn,ℓnΠ,ℓnπ,τnΠ,τnπ,λnn,λnΠ,λnπ,τnω,δππ,ℓπn,ℓnΠ,τπn,τnΠ,λπΠ,λππ,λπn,τπω,
as well as ${\tilde{\zeta }}_{1},\ldots ,{\tilde{\zeta }}_{8}$, ${\tilde{\varkappa }}_{1},\ldots ,{\tilde{\varkappa }}_{8}$, ${\tilde{\eta }}_{1},\ldots ,{\tilde{\eta }}_{9}$, $\varphi_{1},\ldots ,\varphi_{8}$ have to be determined from an underlying microscopic theory. Note that the coefficient $\lambda_{\pi \pi }$ was denoted $\tau_{\pi \pi }$ in [42,48]. As we will see in the following sections, these coefficients assume different values in different approaches to dissipative fluid dynamics, depending on the actual approximation scheme used to derive the equations of motion.

Furthermore, using the first-order constitutive relations, it is always possible to “reshuffle” terms among the J-, K-, and R-terms. Take for instance the term $∼{\tilde{\zeta }}_{3}\theta^{2}$ in Equation ([Disp-formula FD25a-entropy-26-00189]). Using Equation ([Disp-formula FD23a-entropy-26-00189]) at *first order*, $\Pi =-\zeta \theta +O_{2}$, to replace one factor of θ one may rewrite this term as $-\left({\tilde{\zeta }}_{3}/\zeta \right)\theta \Pi +O_{3}$, i.e., it is now of the same form as the term $∼-\delta_{\Pi \Pi }\Pi \theta$ in Equation ([Disp-formula FD24a-entropy-26-00189]) (higher-order corrections are not considered anyway). Repeating this once more to replace the final factor of θ, we arrive at $\left({\tilde{\zeta }}_{3}/\zeta^{2}\right)\Pi^{2}$, i.e., it is of the same form as the term $\varphi_{1}\Pi^{2}$ in Equation ([Disp-formula FD26a-entropy-26-00189]). This shows that the transport coefficients in Equations ([Disp-formula FD23a-entropy-26-00189]) are not uniquely defined and depend on the approximations used to derive the constitutive relations for the dissipative currents. We will return to this ambiguity and point out concrete examples several times throughout this review.

### 2.7. Causality and Stability

For completeness, here we make some general comments about the causality and stability of relativistic fluids. Since the seminal works of Hiscock and Lindblom [5,49], one expects a subtle connection between causality and stability in relativistic systems. Causality states that the speed of light bounds the maximum speed of propagation of information in the fluid. Thus, as it stands, this property must be valid not only near equilibrium, where simple linear-response analyses hold [5], but this must also hold in the fully nonlinear regime. The latter is generally much harder to establish (see [50,51] for a review), with only a few general results known in the literature (see [52,53,54]).

Let us first focus on the dynamics near equilibrium, where the equations of motion of the dissipative fluid are linearized around a constant and uniform non-rotating global-equilibrium state (see [55] for recent work on inhomogeneous systems). In this case, the equations of motion are simple linear partial differential equations for the fluid variables with constant coefficients, and standard methods for computing the characteristic velocities (see, for instance, [56]) apply. These calculations can be done for any global-equilibrium state, regardless of whether or not the background fluid 4-velocity is $u^{\mu }=\left(1,0,0,0\right)$. Alternatively, since the equations of motion are linear with constant coefficients, one may also resort to a Fourier analysis of its frequency modes ω(k), determining the dispersion relations of the collective excitations near equilibrium (e.g., sound and shear disturbances). Following standard group-velocity arguments, some works have used the asymptotic quantity
(27)vf=limk→∞,k∈RReωk=limk→∞,k∈RdReωdk∈[−1,1].
to determine conditions for causality in Israel–Stewart theories. For example, the bulk and shear relaxation times and viscosities, appearing in transient theories of fluid dynamics, must satisfy the necessary condition [57],
(28)$$\frac{\zeta }{\left(\varepsilon_{0}+p_{0}\right)\tau_{\Pi }}+\frac{4}{3}\frac{\eta }{\left(\varepsilon_{0}+p_{0}\right)\tau_{\pi }}\leq 1-c_{s}^{2}\;,$$
with $c_{s}$ being the speed of sound. Many additional constraints can be derived but will not be listed here.

It has been recently shown in [58], however, that the condition (Equation 27) is not sufficient to guarantee causality as there are well-known counterexamples of theories where $v_{f}\in \left[-1,1\right]$ and the dynamics is acausal [58]. The relation between dispersion relations and causality has been elucidated in [58], where it was shown that there is no notion of causality for individual dispersion relations since no mathematical condition on the function ω(k) (such as the asymptotic group-velocity condition) can serve as a sufficient condition for subluminal propagation in dispersive media. Instead, causality can only emerge from a careful cancellation when one superimposes all the excitation branches of a physical model. This happens automatically in local theories of matter where the dispersion relations obey the covariant stability condition [59,60]
(29)Imω(k)≤|Imk|.
See also the recent works [61,62] for further discussion concerning causality, stability, and linear mode analysis.

Another recent development that has shed much light on the connection between causality and stability was discussed in [52], where it was proven that causality is necessary for stability in relativistic fluids. The original argument and proof in [52] are rather technical, involving detailed calculations using strongly hyperbolic systems. As shown in [63], none of those complications are needed. In fact, [63] generally proved that dissipation is only a Lorentz-invariant concept in causal theories. In particular, if the theory is causal, there is no need to perform calculations in a boosted Lorentz frame, i.e., causality guarantees that the conditions obtained for the stability of disturbances in the local rest frame are the same in any other Lorentz frame. This settles several questions and significantly simplifies stability calculations in relativistic fluids.

We close this section with a brief comment about the global properties of relativistic dissipative fluid-dynamics solutions in the nonlinear regime. While small-data global well-posedness for BDNK theory has been recently proven in [64] under very simplifying conditions and in flat space-time, it remains challenging to generalize this result to more realistic scenarios. Further progress was made recently in [65] in the case of Israel–Stewart theories with bulk viscosity (setting shear and diffusion effects to zero). That work showed that there exists a class of smooth initial data for which the corresponding solutions to the Cauchy problem break down in finite time, or they become acausal. Additional work is needed to determine if this truly implies singularity formation. In this regard, we note that this question has been fully answered in the non-relativistic limit of Israel–Stewart theory (with shear and bulk viscosity effects), where it was proven in [66] that the corresponding solutions of the equations of motion are generally not globally well-posed. This, in turn, implies that a vast class of non-Newtonian fluids do not have finite solutions defined at all times.

## 3. Gradient Expansions for Dissipative Fluid Dynamics

In this section, we discuss the space-time gradient expansion as a means to derive dissipative fluid dynamics. We first give some general considerations and then study the examples of Navier–Stokes and Burnett theory, which are first resp. second order in *space-like* gradients. We then turn to BDNK theory, a first-order theory containing both space- *and time-like* gradients. The existence of the latter are paramount to restoring causality and stability. Finally, we discuss resummed BRSSS theory, which is based on the second-order gradient expansion and can be brought into the form ([Disp-formula FD23a-entropy-26-00189]) to obtain an (in principle) causal and stable theory of dissipative fluid dynamics.

### 3.1. General Considerations

For the sake of simplicity, in this section we will work in the Landau frame (Equation 17), with Landau matching conditions (Equation 16), i.e., the dissipative fluid-dynamical quantities are the bulk viscous pressure, the particle-diffusion 4-current, and the shear-stress tensor. In the *gradient expansion*, these are assumed to be expressable solely in terms of powers of space-time gradients of the *primary fluid-dynamical quantities*, i.e., $\alpha_{0}$, $\beta_{0}$, and $u^{\mu }$. The dissipative currents can then be written as
(30a)$$\begin{array}{ccc} \Pi & = & \sum_{i=0}^{\infty }\sum_{j=1}^{N_{\Pi }^{\left(i\right)}}\lambda_{\Pi ,j}^{\left(i\right)}O_{j}^{\left(i\right)}\;, \end{array}$$
(30b)$$\begin{array}{ccc} n^{\mu } & = & \sum_{i=0}^{\infty }\sum_{j=1}^{N_{n}^{\left(i\right)}}\lambda_{n,j}^{\left(i\right)}O_{j}^{(i)\mu }\;, \end{array}$$
(30c)$$\begin{array}{ccc} \pi^{\mu \nu } & = & \sum_{i=0}^{\infty }\sum_{j=1}^{N_{\pi }^{\left(i\right)}}\lambda_{\pi ,j}^{\left(i\right)}O_{j}^{(i)\mu \nu }\;. \end{array}$$
Here, the quantities $O_{j}^{\left(i\right)}$, $O_{j}^{(i)\mu }$, and $O_{j}^{(i)\mu \nu }$ are terms of *i*th order in gradients of $\alpha_{0}$, $\beta_{0}$, and $u^{\mu }$, and $\lambda_{\Pi ,j}^{\left(i\right)}$, $\lambda_{n,j}^{\left(i\right)}$, and $\lambda_{\pi ,j}^{\left(i\right)}$ are the corresponding transport coefficients at this order, respectively. At any given order *i* in gradients, all terms allowed by Lorentz covariance are allowed, i.e., $O_{j}^{\left(i\right)}$ must be a Lorentz scalar, $O_{j}^{(i)\mu }$ a Lorentz vector, and $O_{j}^{(i)\mu \nu }$ a Lorentz tensor of rank 2. In addition, $O_{j}^{(i)\mu }$ must be orthogonal to $u^{\mu }$ (just as $n^{\mu }$) and $O_{j}^{(i)\mu \nu }$ must be orthogonal to $u^{\mu }$, symmetric, and trace-free (just as $\pi^{\mu \nu }$). The index *j* labels the various terms that occur at a given order *i*. In Section 3.2 and Section 3.3 we will show these terms explicitly (but restricted to space-like derivatives) to first and second order, respectively. On the other hand, the transport coefficients $\lambda_{\Pi ,j}^{\left(i\right)}$, $\lambda_{n,j}^{\left(i\right)}$, and $\lambda_{\pi ,j}^{\left(i\right)}$ are functions of temperature and chemical potential that cannot be obtained from symmetry principles alone, they have to be calculated from an underlying microscopic theory.

It is important to remark that when the system exhibits a clear separation between the typical microscopic and macroscopic scales, λ and *L*, respectively, it may be possible to truncate the expansion on the right-hand sides of Equations ([Disp-formula FD30a-entropy-26-00189]). The terms $O_{j}^{\left(i\right)}$, $O_{j}^{(i)\mu }$, and $O_{j}^{(i)\mu \nu }$ are proportional to *i* gradients of a macroscopic variable, and thus are of order $∼L^{-i}$. The microscopic scale λ is contained in the transport coefficients $\lambda_{\Pi ,j}^{\left(i\right)}$, $\lambda_{n,j}^{\left(i\right)}$, and $\lambda_{\pi ,j}^{\left(i\right)}$. Up to some overall power of λ (which restores the correct scaling dimension), $\lambda_{\Pi ,j}^{\left(i\right)}$, $\lambda_{n,j}^{\left(i\right)}$, $\lambda_{\pi ,j}^{\left(i\right)}∼\lambda^{i}$. Therefore, the transport coefficients appearing in $\lambda_{\Pi ,j}^{\left(i\right)}O_{j}^{\left(i\right)}$, $\lambda_{n,j}^{\left(i\right)}O_{j}^{(i)\mu }$, and $\lambda_{\pi ,j}^{\left(i\right)}O_{j}^{(i)\mu \nu }$ in Equations ([Disp-formula FD30a-entropy-26-00189]) are of order ${\left(\lambda /L\right)}^{i}$. Equations ([Disp-formula FD30a-entropy-26-00189]) are then nothing but series in powers of the so-called Knudsen number Kn≡λ/L, cf. Section 2.4. If Kn≪1 and these series converge, the gradient expansion of the dissipative currents can be truncated at a given order and one obtains a closed macroscopic theory for them, with a clear domain of applicability.

Ideal fluid dynamics, cf. Section 2.3, corresponds to the zeroth-order truncation of this series, i.e., when only the (i=0)-terms on the right-hand sides of Equations ([Disp-formula FD30a-entropy-26-00189]) are considered. In this case, there are no gradient terms at all. Since for ideal fluid dynamics all dissipative currents vanish, we must have $\lambda_{\Pi ,j}^{\left(0\right)}=\lambda_{n,j}^{\left(0\right)}=\lambda_{\pi ,j}^{\left(0\right)}\equiv 0$ for all *j*, so that the right-hand sides of Equations ([Disp-formula FD30a-entropy-26-00189]) vanish as well. The first-order truncation of the gradient expansion using only space-like gradients is Navier–Stokes theory, as shown below. At second order in space-like gradients, one obtains the relativistic Burnett equations, while higher orders lead to the super-Burnett equations.

We note that the convergence properties of the gradient expansion under general conditions have not been established and are a topic that is still being intensely investigated, see [67]. This considerably changes our interpretation of truncations of the gradient expansion, which no longer have a clear domain of validity in terms of the magnitude of the Knudsen number. Instead, one determines the optimal truncation of the gradient-expansion series and uses that as an effective fluid-dynamical theory. In the following, for the sake of simplicity, we ignore these issues and consider the effective theories that arise as one truncates the gradient expansion.

### 3.2. First Order: Navier–Stokes Theory

The first term in the gradient expansion can be obtained by constructing all possible tensors that can be formed from the first-order derivatives of the primary fluid-dynamical quantities. For the latter, we could take $\alpha_{0}$, $\beta_{0}$, and $u^{\mu }$. However, since gradients of the inverse temperature influence the fluid-dynamical variables only indirectly, through gradients of thermal potential and pressure, it is more physical to choose $\alpha_{0}$, $p_{0}$, and $u^{\mu }$ (instead of $\alpha_{0}$, $\beta_{0}$, and $u^{\mu }$) as primary fluid-dynamical quantities. Replacing gradients of $\beta_{0}$ by gradients of $p_{0}$ (and $\alpha_{0}$) is achieved with the help of the thermodynamic identity
(31)$$d\beta_{0}=\frac{n_{0}}{\varepsilon_{0}+p_{0}}\;d\alpha_{0}-\frac{\beta_{0}}{\varepsilon_{0}+p_{0}}\;dp_{0}\;.$$
The space-time gradients of $\alpha_{0}$, $\beta_{0}$, and $u^{\mu }$ are
(32)$$\partial_{\mu }\alpha_{0}\;,\partial_{\mu }p_{0}\;,\;\mathrm{and}\;\partial_{\mu }u_{\nu }\;.$$
Next, using these gradients, one has to construct tensors that have the same properties as the dissipative currents, i.e., the same symmetry and behavior under Lorentz transformation, and, in the case of tensors of non-vanishing rank, they have to be orthogonal to $u^{\mu }$. There must be a scalar, such as the bulk viscous pressure Π, a 4-vector orthogonal to $u^{\mu }$, such as the particle diffusion 4-current $n^{\mu }$, and a symmetric, traceless second-rank tensor orthogonal to $u^{\mu }$, such as the shear-stress tensor $\pi^{\mu \nu }$. In the traditional gradient expansion, all possible terms involving first-order *time-like* derivatives of $\alpha_{0}$, $\beta_{0}$, and $u^{\mu }$ are converted into space-like ones using the equations of motion ([Disp-formula FD4a-entropy-26-00189]), up to higher-order terms, see Appendix A. Thus, the only possibilities are
(33a)$$\begin{array}{cc} O_{1}^{\left(1\right)} & =\nabla \cdot u\equiv \theta \;, \end{array}$$
(33b)$$\begin{array}{cc} O_{1}^{(1)\mu } & =\nabla^{\mu }\alpha_{0}\equiv I^{\mu }\;,\;O_{2}^{(1)\mu }=\nabla^{\mu }p_{0}\equiv F^{\mu }\;, \end{array}$$
(33c)$$\begin{array}{cc} O_{1}^{(1)\mu \nu } & ={∆}_{\alpha \beta }^{\mu \nu }\partial^{\alpha }u^{\beta }\equiv \sigma^{\mu \nu }\;. \end{array}$$
However, in order to respect the second law of thermodynamics, there cannot be a term $∼\nabla^{\mu }\beta_{0}$, see the discussion in [48], and thus a possible term $∼\nabla^{\mu }p_{0}$ is simply $∼\nabla^{\mu }\alpha_{0}$, by virtue of Equation (Equation 31). Without loss of generality we therefore can assume that $\lambda_{n,2}^{\left(1\right)}=0$, and thus the term $O_{2}^{(1)\mu }$ does not appear. We therefore have $N_{\Pi }^{\left(1\right)}=N_{n}^{\left(1\right)}=N_{\pi }^{\left(1\right)}=1$. Consequently, the most general relations satisfied by Π, $n^{\mu }$, and $\pi^{\mu \nu }$, up to first order in Kn, are
(34a)$$\begin{array}{ccc} \Pi & = & \lambda_{\Pi ,1}^{\left(1\right)}\theta \equiv -\zeta \theta \;, \end{array}$$
(34b)$$\begin{array}{ccc} n^{\mu } & = & \lambda_{n,1}^{\left(1\right)}I^{\mu }\equiv \varkappa I^{\mu }\;, \end{array}$$
(34c)$$\begin{array}{ccc} \pi^{\mu \nu } & = & \lambda_{\pi ,1}^{\left(1\right)}\sigma^{\mu \nu }\equiv 2\eta \sigma^{\mu \nu }\;, \end{array}$$
which corresponds to relativistic Navier–Stokes theory [1], with the bulk-viscosity coefficient $\zeta \equiv -\lambda_{\Pi ,1}^{\left(1\right)}$, the diffusion coefficient $\varkappa \equiv \lambda_{n,1}^{\left(1\right)}$, and the shear-viscosity coefficient $\eta \equiv \lambda_{\pi ,1}^{\left(1\right)}/2$, respectively.

We note that relativistic Navier–Stokes theory has no practical applications since it is acausal and unstable. This can be shown, for instance, via a simple linear stability analysis, although one has to go to a moving frame to discover the instability, see, e.g., [48]. Acausality follows from the fact that, in the absence of time-like derivatives in the constitutive relations, hyperbolicity is invariably lost when using Equations ([Disp-formula FD34a-entropy-26-00189]) to obtain the equations of motion ([Disp-formula FD4a-entropy-26-00189]), namely
(35a)$$\begin{array}{cc} {\dot{n}}_{0} & =-n_{0}\theta -\partial_{\mu }\left(\varkappa I^{\mu }\right)\;, \end{array}$$
(35b)$$\begin{array}{cc} {\dot{\varepsilon }}_{0} & =-\left(\varepsilon_{0}+p_{0}-\zeta \theta \right)\theta +2\eta \sigma_{\alpha \beta }\sigma^{\alpha \beta }\;, \end{array}$$
(35c)$$\begin{array}{cc} \left(\varepsilon_{0}+p_{0}-\zeta \theta \right){\dot{u}}^{\mu } & =F^{\mu }-\nabla^{\mu }\left(\zeta \theta \right)-2{∆}_{\alpha }^{\mu }\partial_{\beta }\left(\eta \sigma^{\alpha \beta }\right)\;. \end{array}$$
These equations are known as the relativistic Navier–Stokes equations. In this formulation, the state of a dissipative fluid is described by the same variables as in the case of an ideal fluid, i.e., by the primary fluid-dynamical variables $\alpha_{0}$, $\beta_{0}$, and $u^{\mu }$. The only difference is the existence of dissipative processes corresponding to new forms of particle and energy-momentum transfer, which occur due to purely space-like gradients of the primary fluid-dynamical variables. The first and last Equation ([Disp-formula FD35a-entropy-26-00189]) become explicitly parabolic in the linear regime, containing modes that propagate infinitely fast [48].

### 3.3. Second Order: Burnett Theory

At second order in gradients, the Lorentz scalar terms are
(36a)$$\begin{array}{cc} O_{1}^{\left(2\right)} & =\omega_{\mu \nu }\omega^{\mu \nu }\;,\;O_{2}^{\left(2\right)}=\sigma_{\mu \nu }\sigma^{\mu \nu }\;,\;O_{3}^{\left(2\right)}=\theta^{2}\;,\;O_{4}^{\left(2\right)}=I\cdot I\;,\\ O_{5}^{\left(2\right)} & =F\cdot F\;,\;O_{6}^{\left(2\right)}=I\cdot F\;,\;O_{7}^{\left(2\right)}=\nabla \cdot I\;,\;O_{8}^{\left(2\right)}=\nabla \cdot F\;, \end{array}$$
while the Lorentz vectors orthogonal to $u^{\mu }$ are
(36b)$$\begin{array}{cc} O_{1}^{(2)\mu } & =\sigma^{\mu \nu }I_{\nu }\;,\;O_{2}^{(2)\mu }=\sigma^{\mu \nu }F_{\nu }\;,\;O_{3}^{(2)\mu }=I^{\mu }\theta \;,\;O_{4}^{(2)\mu }=F^{\mu }\theta \;,\\ O_{5}^{(2)\mu } & =\omega^{\mu \nu }I_{\nu }\;,\;O_{6}^{(2)\mu }={∆}_{\lambda }^{\mu }\partial_{\nu }\sigma^{\lambda \nu }\;,\;O_{7}^{(2)\mu }=\nabla^{\mu }\theta \;,\;O_{8}^{(2)\mu }=\omega^{\mu \nu }F_{\nu }\;, \end{array}$$
and the Lorentz tensors of rank 2, which are symmetric, traceless, and orthogonal to $u^{\mu }$, read
(36c)$$\begin{array}{cc} O_{1}^{(2)\mu \nu } & =\omega_{\lambda }^{\;\langle \mu }\omega^{\nu \rangle \lambda }\;,\;O_{2}^{(2)\mu \nu }=\theta \sigma^{\mu \nu }\;,\;O_{3}^{(2)\mu \nu }=\sigma_{\lambda }^{\;\langle \mu }\sigma^{\nu \rangle \lambda }\;,\\ O_{4}^{(2)\mu \nu } & =\sigma_{\lambda }^{\;\langle \mu }\omega^{\nu \rangle \lambda }\;,\;O_{5}^{(2)\mu \nu }=I^{\langle \mu }I^{\nu \rangle }\;,\;O_{6}^{(2)\mu \nu }=F^{\langle \mu }F^{\nu \rangle }\;,\\ O_{7}^{(2)\mu \nu } & =I^{\langle \mu }F^{\nu \rangle }\;,\;O_{8}^{(2)\mu \nu }=\nabla^{\langle \mu }I^{\nu \rangle }\;,\;O_{9}^{(2)\mu \nu }=\nabla^{\langle \mu }F^{\nu \rangle }\;. \end{array}$$
As before, any term containing a time-like derivative of a primary fluid-dynamical variable allowed by symmetry is replaced by terms containing only space-like derivatives using the conservation laws.

Including the above terms in the expressions for the dissipative currents, Equations ([Disp-formula FD30a-entropy-26-00189]), and denoting ${\tilde{\zeta }}_{j}\equiv \lambda_{\Pi ,j}^{\left(2\right)}$, ${\tilde{\varkappa }}_{j}\equiv \lambda_{n,j}^{\left(2\right)}$, and ${\tilde{\eta }}_{j}\equiv \lambda_{\pi ,j}^{\left(2\right)}$ we obtain the relativistic Burnett equations [68,69],
(37a)$$\begin{array}{c} \Pi =-\zeta \theta +{\tilde{\zeta }}_{1}\omega_{\mu \nu }\omega^{\mu \nu }+{\tilde{\zeta }}_{2}\sigma_{\mu \nu }\sigma^{\mu \nu }+{\tilde{\zeta }}_{3}\theta^{2}+{\tilde{\zeta }}_{4}I\cdot I\\ +{\tilde{\zeta }}_{5}F\cdot F+{\tilde{\zeta }}_{6}I\cdot F+{\tilde{\zeta }}_{7}\nabla \cdot I+{\tilde{\zeta }}_{8}\nabla \cdot F\;, \end{array}$$
(37b)$$\begin{array}{c} n^{\mu }=\;\varkappa I^{\mu }+{\tilde{\varkappa }}_{1}\sigma^{\mu \nu }I_{\nu }+{\tilde{\varkappa }}_{2}\sigma^{\mu \nu }F_{\nu }+{\tilde{\varkappa }}_{3}I^{\mu }\theta +{\tilde{\varkappa }}_{4}F^{\mu }\theta \\ \;\;+{\tilde{\varkappa }}_{5}\omega^{\mu \nu }I_{\nu }+{\tilde{\varkappa }}_{6}{∆}_{\lambda }^{\mu }\partial_{\nu }\sigma^{\lambda \nu }+{\tilde{\varkappa }}_{7}\nabla^{\mu }\theta +{\tilde{\varkappa }}_{8}\omega^{\mu \nu }F_{\nu }\;, \end{array}$$
(37c)$$\begin{array}{c} \pi^{\mu \nu }=\;2\eta \sigma^{\mu \nu }+{\tilde{\eta }}_{1}\omega_{\lambda }^{\left.\right.\langle \mu }\omega^{\nu \rangle \lambda }+{\tilde{\eta }}_{2}\theta \sigma^{\mu \nu }+{\tilde{\eta }}_{3}\sigma^{\lambda \langle \mu }\sigma_{\lambda }^{\nu \rangle }+{\tilde{\eta }}_{4}\sigma_{\lambda }^{\langle \mu }\omega^{\nu \rangle \lambda }\\ \;\;+{\tilde{\eta }}_{5}I^{\langle \mu }I^{\nu \rangle }+{\tilde{\eta }}_{6}F^{\langle \mu }F^{\nu \rangle }+{\tilde{\eta }}_{7}I^{\langle \mu }F^{\nu \rangle }+{\tilde{\eta }}_{8}\nabla^{\langle \mu }I^{\nu \rangle }+{\tilde{\eta }}_{9}\nabla^{\langle \mu }F^{\nu \rangle }\;. \end{array}$$
Obviously, this theory is a particular case of the more general second-order equations of motion ([Disp-formula FD23a-entropy-26-00189]), with vanishing relaxation times, $\tau_{\Pi }=\tau_{n}=\tau_{\pi }=0$, and vanishing J- and R-terms. We note that Burnett’s theory, like Navier–Stokes theory, is expected to be linearly acausal and unstable[note 3] and, thus, has no practical applications. As a matter of fact, the Burnett equations are unstable even in the non-relativistic regime [71].

We note that terms with comoving (time-like) derivatives of primary fluid-dynamical variables could be introduced on the right-hand sides of Equation ([Disp-formula FD37a-entropy-26-00189]). However, as already mentioned and shown explicitly in Appendix A, using the equations of motion ([Disp-formula FD4a-entropy-26-00189]) to leading order, these terms can be expressed in terms of those already present in Equations ([Disp-formula FD37a-entropy-26-00189]). This would then merely lead to a redefinition of some of the transport coefficients in these equations. Nevertheless, it is important to remark that even though such terms can be arbitrarily reshuffled if one indiscriminately follows this scheme, this procedure may considerably affect the mathematical properties of the equations of motion. This will become clearer when we discuss BDNK and resummed BRSSS theories in the following.

### 3.4. BDNK Theory

In the global-equilibrium state, $\beta_{\mu }=\beta_{0}u_{\mu }\equiv u_{\mu }/T$ is a Killing vector [14], so $\partial_{(\mu }\beta_{\nu )}=0$ and $\partial_{\mu }\alpha_{0}=0$ [72]. It is natural to assume that if one is sufficiently close to the global-equilibrium state, the energy-momentum tensor and the particle-number current can be written as a truncated Taylor expansion around global equilibrium as follows:
(38a)$$\begin{array}{cc} N^{\mu } & =N_{0}^{\mu }+Y^{\mu \rho \sigma }T\partial_{\rho }\beta_{\sigma }+Z^{\mu \rho }\partial_{\rho }\alpha_{0}+O\left(\partial^{2}\right)\;, \end{array}$$
(38b)$$\begin{array}{cc} T^{\mu \nu } & =T_{0}^{\mu \nu }+H^{\mu \nu \rho \sigma }T\partial_{\rho }\beta_{\sigma }+X^{\mu \nu \rho }\partial_{\rho }\alpha_{0}+O\left(\partial^{2}\right)\;, \end{array}$$
with $N_{0}^{\mu }$ and $T_{0}^{\mu \nu }$ being the conserved currents in local equilibrium, cf. Equations ([Disp-formula FD8a-entropy-26-00189]), and the remaining tensors corresponding to the most general tensor structures that can be constructed from the 4-velocity and the metric tensor,
(39a)$$\begin{array}{cc} Y^{\mu \rho \sigma } & =\nu_{1}u^{\mu }u^{\rho }u^{\sigma }+\nu_{2}u^{\mu }{∆}^{\rho \sigma }+\gamma_{1}u^{\sigma }{∆}^{\mu \rho }+\gamma_{2}u^{\rho }{∆}^{\mu \sigma }\;, \end{array}$$
(39b)$$\begin{array}{c} H^{\mu \nu \rho \sigma }=\varepsilon_{1}u^{\mu }u^{\nu }u^{\rho }u^{\sigma }+\varepsilon_{2}u^{\mu }u^{\nu }{∆}^{\rho \sigma }+\pi_{1}u^{\rho }u^{\sigma }{∆}^{\mu \nu }+\pi_{2}{∆}^{\mu \nu }{∆}^{\rho \sigma }\\ \;\;\;+\theta_{1}\left(u^{\mu }u^{\sigma }{∆}^{\nu \rho }+u^{\nu }u^{\sigma }{∆}^{\mu \rho }\right)+\theta_{2}\left(u^{\nu }u^{\rho }{∆}^{\mu \sigma }+u^{\mu }u^{\rho }{∆}^{\nu \sigma }\right)+2\eta \;{∆}^{\mu \nu \rho \sigma }\;, \end{array}$$
(39c)$$\begin{array}{cc} Z^{\mu \rho } & =\nu_{3}u^{\mu }u^{\rho }+\gamma_{3}{∆}^{\mu \rho }\;, \end{array}$$
(39d)$$\begin{array}{cc} X^{\mu \nu \rho } & =\varepsilon_{3}u^{\mu }u^{\nu }u^{\rho }+\pi_{3}u^{\rho }{∆}^{\mu \nu }+\theta_{3}\left(u^{\nu }{∆}^{\mu \rho }+u^{\mu }{∆}^{\nu \rho }\right)\;. \end{array}$$
In the sense of effective field theory, this procedure provides the local-equilibrium values of the conserved currents with the most general corrections of the corresponding Lorentz-tensor rank, which can be constructed using first-order derivatives of the primary fluid-dynamical variables. The main difference to the formalism presented in the previous subsection is that time-like derivatives are no longer systematically replaced by space-like ones by employing conservation laws. Even though such a replacement appears to be allowed by the power-counting scheme itself, it was shown to considerably affect the mathematical properties of the resulting equations of motion, at least for first-order truncations of this gradient expansion [43,52,73,74,75]. In principle, this procedure could be continued at a higher order in derivatives, although the total number of possible terms will increase significantly. The general constitutive relations truncated at first order, as shown above, together with the conservation laws ([Disp-formula FD3a-entropy-26-00189]) is now known as the BDNK theory of fluid dynamics [43,52,73,74,75].

Above, $\varepsilon_{0}$, $p_{0}$, and $n_{0}$ are (as before) interpreted as the equilibrium energy density, pressure, and particle-number density (related via the equilibrium equation of state). At the same time, the remaining parameters (which can depend on the thermodynamic functions) describe dissipative effects (shear, bulk, heat flow, or particle diffusion) in an arbitrary fluid-dynamical frame [73]. For example, $\varepsilon_{1,2,3}$ denote corrections to the energy density, while $\pi_{1,2,3}$ denote corrections to the pressure. Indeed, Equations ([Disp-formula FD39a-entropy-26-00189]) imply the following constitutive relations for the dissipative currents
(40a)$$\begin{array}{cc} \Pi \; & =-\pi_{3}D\alpha_{0}-\pi_{2}\frac{D\beta_{0}}{\beta_{0}}-\pi_{1}\theta \;,\;\delta n=\nu_{3}D\alpha_{0}+\nu_{2}\frac{D\beta_{0}}{\beta_{0}}+\nu_{1}\theta \;, \end{array}$$
(40b)$$\begin{array}{cc} \delta \varepsilon \; & =\varepsilon_{3}D\alpha_{0}+\varepsilon_{2}\frac{D\beta_{0}}{\beta_{0}}+\varepsilon_{1}\theta \;,\;n^{\mu }=\gamma_{3}I^{\mu }+\gamma_{1}\frac{1}{\beta_{0}}\nabla^{\mu }\beta_{0}+\gamma_{2}Du^{\mu }\;, \end{array}$$
(40c)$$\begin{array}{cc} h^{\mu }\; & =\theta_{3}I^{\mu }+\theta_{1}\frac{1}{\beta_{0}}\nabla^{\mu }\beta_{0}+\theta_{2}Du^{\mu }\;,\;\pi^{\mu \nu }=2\eta \sigma^{\mu \nu }\;. \end{array}$$
We note that several constraints arise from thermodynamic consistency, from the requirement that Equations ([Disp-formula FD38a-entropy-26-00189]) correctly reduce to their first-principles expressions in global-equilibrium states. First, $\varepsilon_{0}$, $p_{0}$, and $n_{0}$ must satisfy the standard thermodynamic identities $\varepsilon_{0}+p_{0}=T{\left(\partial p_{0}/\partial T\right)}_{\mu }+\mu {\left(\partial p_{0}/\partial \mu \right)}_{T}$ and $n_{0}={\left(\partial p_{0}/\partial \mu \right)}_{T}$, and second, we must have $H^{\mu \nu \rho \sigma }=H^{\mu \nu \sigma \rho }$ and $Y^{\mu \rho \sigma }=Y^{\mu \sigma \rho }$ (which were already enforced from the very beginning). Furthermore, $\gamma_{1}=\gamma_{2}$, $\theta_{1}=\theta_{2}$ [73]. These conditions can be viewed as necessary and sufficient conditions for the first-order terms in Equations ([Disp-formula FD38a-entropy-26-00189]), representing dissipative corrections to vanish in states of global equilibrium, which are defined by $\partial_{(\mu }\beta_{\nu )}=0$ and $\partial_{\mu }\alpha_{0}=0$.

The BDNK formalism can describe causal and stable dissipative fluid dynamics with shear, bulk, and particle- or energy-diffusion effects, see [52]. The equations of motion display interesting properties, such as strong hyperbolicity and local well-posedness[note 4] of the initial-value problem [50,52,76,77], even when the fluid is dynamically coupled to Einstein’s equations—no other formalism has been shown to accomplish this including effects of shear, bulk, heat flow, and/or particle diffusion. In fact, one can show that the causality and instability issues originally found in theories of Eckart and Landau-Lifshitz [5] are not necessarily present in BDNK theory [43,52,73,74,75]. The crucial difference resides in the fact that the constitutive relations in BDNK theory display time-like derivatives of the primary fluid-dynamical variables. These terms effectively change the nature of the differential equation and make it possible to restore hyperbolicity and causality and to render the theory stable against small perturbations around equilibrium [43]—although the theory is formally of the same order as Navier–Stokes theory in the power-counting scheme discussed so far. When substituted into the local conservation laws ([Disp-formula FD3a-entropy-26-00189]), the constitutive relations ([Disp-formula FD40a-entropy-26-00189]) lead to
(41a)$$\begin{array}{cc} 0 & =Dn_{0}+\left(n_{0}+\nu_{3}D\alpha_{0}+\nu_{2}\frac{D\beta_{0}}{\beta_{0}}+\nu_{1}\theta \right)\theta +\partial_{\mu }\left[\gamma_{3}I^{\mu }+\gamma_{1}\frac{1}{\beta_{0}}\nabla^{\mu }\beta_{0}+\gamma_{2}Du^{\mu }\right]\;, \end{array}$$
(41b)$$\begin{array}{cc} 0 & =D\varepsilon_{0}+D\left(\varepsilon_{3}D\alpha_{0}+\varepsilon_{2}\frac{D\beta_{0}}{\beta_{0}}+\varepsilon_{1}\theta \right)+\left(\varepsilon_{0}+\varepsilon_{3}D\alpha_{0}+\varepsilon_{2}\frac{D\beta_{0}}{\beta_{0}}+\varepsilon_{1}\theta \right)\theta \\ \; & \;\;+\left(p_{0}-\pi_{3}D\alpha_{0}-\pi_{2}\frac{D\beta_{0}}{\beta_{0}}-\pi_{1}\theta \right)\theta -2\eta \sigma^{\mu \nu }\sigma_{\mu \nu }\\ \; & \;\;+\left(-2Du_{\mu }+\nabla_{\mu }\right)\left[\theta_{3}I^{\mu }+\theta_{1}\frac{1}{\beta_{0}}\nabla^{\mu }\beta_{0}+\theta_{2}Du^{\mu }\right]\;, \end{array}$$
(41c)$$\begin{array}{cc} 0 & =\left(\varepsilon_{0}+\varepsilon_{3}D\alpha_{0}+\varepsilon_{2}\frac{D\beta_{0}}{\beta_{0}}+\varepsilon_{1}\theta +p_{0}-\pi_{3}D\alpha_{0}-\pi_{2}\frac{D\beta_{0}}{\beta_{0}}-\pi_{1}\theta \right)Du^{\mu }\\ \; & \;\;-\nabla^{\mu }\left(p_{0}-\pi_{3}D\alpha_{0}-\pi_{2}\frac{D\beta_{0}}{\beta_{0}}-\pi_{1}\theta \right)+2{∆}^{\mu \nu }\partial_{\alpha }\left(\eta \sigma_{\;\nu }^{\alpha }\right)\\ \; & \;\;+\left(\frac{4}{3}\theta {∆}^{\mu \nu }+\sigma^{\mu \nu }-\omega^{\mu \nu }+{∆}^{\mu \nu }D\right)\left[\theta_{3}I_{\nu }+\theta_{1}\frac{1}{\beta_{0}}\nabla_{\nu }\beta_{0}+\theta_{2}Du_{\nu }\right]\;, \end{array}$$
which should be contrasted with the corresponding equations of motion in Navier–Stokes theory, cf. Equations ([Disp-formula FD35a-entropy-26-00189])[note 5].

One may consider BDNK theory as a way to regularize Navier–Stokes theory by appropriately changing its non-hydrodynamic modes using transformations of the fluid-dynamical frame [73], obtaining a causal and stable evolution without adding an extended set of dynamical variables (such as in Israel–Stewart theory). The original BDNK formalism has recently been used and extended in several ways. For example, [78] used BDNK theory to derive a causal and stable theory of relativistic magnetohydrodynamics, while [79] investigated its applications in cosmology. The connection between second-order theories and BDNK theories was investigated in [80], showing that BDNK theories can emerge as the first-order truncation of a generalized second-order theory formulated in a general fluid-dynamical frame. In that case, one finds that causality and stability of the second-order theory implies causality and stability for its BDNK truncation. Numerical simulations of BDNK theory have been investigated in [81,82,83,84], and the properties of shock waves in this approach have been investigated in [85]. The properties of BDNK theory in the presence of anomalies have been derived in [86], while the BDNK formulation of spin hydrodynamics was worked out in [87]. Finally, the stochastic formulation of BDNK theory has been recently discussed in [38,39,40]. In this review, we will discuss in Section 5.3 how the BDNK coefficients shown in Equations ([Disp-formula FD39a-entropy-26-00189]) can be computed from relativistic kinetic theory, following the systematic procedure derived in [88] (see also [43,74,89] for related work).

### 3.5. Resummed BRSSS Theory

The original BRSSS paper [90] dealt with a conformal theory, i.e., a system where $T_{\mu }^{\mu }=0$ and the dynamical variables and equations of motion change covariantly under Weyl transformations. In such systems at zero chemical potential, thermodynamic quantities such as the energy density scale as $\varepsilon_{0}∼T^{4}$ and the bulk viscous pressure Π vanishes identically, assuming Landau matching conditions. Furthermore, the system considered in [90] carried no conserved charge (besides energy-momentum), thus $\alpha_{0}=0$ and $n^{\mu }=I^{\mu }=0$. Therefore, the only dissipative quantity is the shear-stress tensor $\pi^{\mu \nu }$. For $\alpha_{0}=0$, the thermodynamic identity (Equation 31) allows us to replace $F^{\mu }$ by $\nabla^{\mu }\beta_{0}\equiv J^{\mu }$. The gradient expansion ([Disp-formula FD37a-entropy-26-00189]) then immediately gives
(42)$$\begin{array}{cc} \pi^{\mu \nu } & =2\eta \sigma^{\mu \nu }+{\tilde{\eta }}_{1}\omega_{\lambda }^{\;\langle \mu }\omega^{\nu \rangle \lambda }+{\tilde{\eta }}_{2}\theta \sigma^{\mu \nu }+{\tilde{\eta }}_{3}\sigma_{\lambda }^{\;\langle \mu }\sigma^{\nu \rangle \lambda }+{\tilde{\eta }}_{4}\sigma_{\lambda }^{\;\langle \mu }\omega^{\nu \rangle \lambda }+{\bar{\eta }}_{6}J^{\langle \mu }J^{\nu \rangle }+{\bar{\eta }}_{9}\nabla^{\langle \mu }J^{\nu \rangle }\;, \end{array}$$
where ${\bar{\eta }}_{6}$ and ${\bar{\eta }}_{9}$ are linear combinations of ${\tilde{\eta }}_{6}$ and ${\tilde{\eta }}_{9}$, multiplied by thermodynamic functions. The discussion in [90] also accounts for terms coming from a non-trivial space-time curvature, which we neglect here for the sake of simplicity. Requiring that Equation (Equation 42) transforms homogeneously under Weyl transformations imposes certain restrictions on the transport coefficients, such that the terms $∼{\tilde{\eta }}_{2}$, ${\bar{\eta }}_{6}$, and ${\bar{\eta }}_{9}$ can be replaced by the linear combination ${\dot{\sigma }}^{\langle \mu \nu \rangle }+\frac{1}{3}\theta \sigma^{\mu \nu }$. Thus, one arrives, after a suitable redefinition of transport coefficients, at
(43)$$\pi^{\mu \nu }=2\eta \sigma^{\mu \nu }-2\eta \tau_{\pi }\left({\dot{\sigma }}^{\langle \mu \nu \rangle }+\frac{1}{3}\theta \sigma^{\mu \nu }\right)+4\lambda_{1}\;\sigma_{\lambda }^{\;\langle \mu }\sigma^{\nu \rangle \lambda }+2\lambda_{2}\;\sigma_{\lambda }^{\;\langle \mu }\omega^{\nu \rangle \lambda }+\lambda_{3}\;\omega_{\lambda }^{\;\langle \mu }\omega^{\nu \rangle \lambda }\;.$$
This is, by itself, a constitutive relation for $\pi^{\mu \nu }$ and, when inserting this into the conservation laws ([Disp-formula FD4a-entropy-26-00189]) and neglecting nonlinear terms, one obtains non-hyperbolic partial differential equations that violate causality and are subject to instabilities. However, following the argument presented at the end of Section 2.6, one can turn the otherwise unsuitable expression (Equation 43) into a hyperbolic differential equation (at least in the linear regime), which, under certain conditions, respects causality and is stable. The idea is to replace $\sigma^{\mu \nu }$ in the second-order terms on the right-hand side by its first-order expression given by Navier–Stokes theory, Equation ([Disp-formula FD34c-entropy-26-00189]), i.e., $\sigma^{\mu \nu }\to \pi^{\mu \nu }/\left(2\eta \right)$, which up to second order in gradients is certainly allowed. This then leads to
(44)$$\tau_{\pi }{\dot{\pi }}^{\langle \mu \nu \rangle }+\pi^{\mu \nu }=2\eta \sigma^{\mu \nu }-\frac{4}{3}\tau_{\pi }\theta \pi^{\mu \nu }+\frac{\lambda_{1}}{\eta^{2}}\;\pi_{\lambda }^{\;\langle \mu }\pi^{\nu \rangle \lambda }+\frac{\lambda_{2}}{\eta }\;\pi_{\lambda }^{\;\langle \mu }\omega^{\nu \rangle \lambda }+\lambda_{3}\;\omega_{\lambda }^{\;\langle \mu }\omega^{\nu \rangle \lambda }\;,$$
where one has made use of the fact that, because of conformal invariance, the shear-viscosity coefficient fulfills $\dot{\eta }=-\eta \theta$ to leading order. Comparing this to the general expression ([Disp-formula FD23c-entropy-26-00189]) for second-order theories, we identify $\varphi_{7}\equiv \lambda_{1}/\eta^{2}$, $2\tau_{\pi \omega }\equiv \lambda_{2}/\eta$, and ${\tilde{\eta }}_{1}\equiv \lambda_{3}$. All other transport coefficients appearing in $J^{\mu \nu }$, $K^{\mu \nu }$, and $R^{\mu \nu }$ vanish.

Considering linear disturbances around a non-rotating equilibrium state (such that the term $∼\lambda_{3}$ does not contribute), the system of equations can be hyperbolic, causal, and stable provided the relaxation time $\tau_{\pi }$ is sufficiently large compared to the shear-viscosity coefficient η [57,91], see Equation (Equation 28) at vanishing bulk viscosity and for $c_{s}^{2}=1/3$. Conditions that establish causality even in the nonlinear regime can be obtained by using the constraints found in [53,54] (assuming $\lambda_{2}=\lambda_{3}=0$), which have been investigated in simulations [92,93,94].

The approach of [90] was later extended to nonconformal theories, where the bulk viscous pressure is non-zero [95]. Resurrecting the term $∼\dot{\theta }$ from the gradient-expansion result ([Disp-formula FD37a-entropy-26-00189]) via Equation ([Disp-formula FD266b-entropy-26-00189]), one obtains
(45)$$\Pi =-\zeta \theta +\zeta \tau_{\Pi }\dot{\theta }+\xi_{1}\sigma^{\mu \nu }\sigma_{\mu \nu }+\xi_{2}\theta^{2}+\xi_{3}\omega^{\mu \nu }\omega_{\mu \nu }+\xi_{4}{\left(\varepsilon_{0}+p_{0}\right)}^{2}F\cdot F\;,$$
where terms from non-vanishing curvature have again been omitted. Now employing the first-order result ([Disp-formula FD34a-entropy-26-00189]), we again arrive at an equation of relaxation type,
(46)$$\begin{array}{cc} \tau_{\Pi }\dot{\Pi }+\Pi & =-\zeta \theta +\tau_{\Pi }\left(A\frac{\partial \ln\zeta }{\partial \alpha_{0}}+B\frac{\partial \ln\zeta }{\partial \beta_{0}}\right)\Pi \theta +\frac{\xi_{1}}{4\eta^{2}}\pi^{\mu \nu }\pi_{\mu \nu }\\ \; & \;\;+\frac{\xi_{2}}{\zeta^{2}}\Pi^{2}+\xi_{3}\omega^{\mu \nu }\omega_{\mu \nu }+\xi_{4}{\left(\varepsilon_{0}+p_{0}\right)}^{2}F\cdot F\;, \end{array}$$
where we have used Equation (Equation 257) as well as replaced $\sigma^{\mu \nu }\to \pi^{\mu \nu }/\left(2\eta \right)$. Comparison with Equation ([Disp-formula FD23a-entropy-26-00189]) reveals that, in this theory,
(47a)$$\begin{array}{cc} \delta_{\Pi \Pi } & \equiv -\tau_{\pi }\left(A\frac{\partial \ln\zeta }{\partial \alpha_{0}}+B\frac{\partial \ln\zeta }{\partial \beta_{0}}\right)\;,\;\varphi_{1}\equiv \frac{\xi_{2}}{\zeta^{2}}\;,\;\varphi_{2}\equiv \frac{\xi_{1}}{4\eta^{2}}\;, \end{array}$$
(47b)$$\begin{array}{cc} {\tilde{\zeta }}_{1} & \equiv \xi_{3}\;,\;{\tilde{\zeta }}_{5}\equiv \xi_{4}{\left(\varepsilon_{0}+p_{0}\right)}^{2}\;, \end{array}$$
while all other transport coefficients in Equation ([Disp-formula FD23a-entropy-26-00189]) are zero. Of course, while formally correct to the order we are considering in the equations of motion, employing the replacements θ→−Π/ζ, $\sigma^{\mu \nu }\to \pi^{\mu \nu }/\left(2\eta \right)$ is to some extent arbitrary, in the sense that, for instance, instead of generating a term $∼\Pi^{2}$ from the term $∼\theta^{2}$, one could as well have replaced only one factor θ, and one would then have “reshuffled” the coefficient $\varphi_{1}$ into the (suitably redefined) coefficient $\delta_{\Pi \Pi }$. The same argument also holds for the other terms where the first-order constitutive relations were used to replace gradients of primary fluid-dynamical variables.

## 4. Dissipative Fluid Dynamics from the Second Law of Thermodynamics

In this section, we derive several formulations of relativistic fluid dynamics by imposing the second law of thermodynamics. For the sake of simplicity, all derivations shall be performed in the Landau frame. Nevertheless, the derivation procedure is general and can be adapted to arbitrary matching conditions (see [80]). We start with a discussion of covariant thermodynamics. After that, we present the derivation of Navier–Stokes and Israel–Stewart theory. We close the section with a brief discussion of divergence-type theories, where the second law of thermodynamics is instrumental to the construction of these theories.

### 4.1. Covariant Thermodynamics

The phenomenological approach to deriving dissipative fluid dynamics is based on the second law of thermodynamics. The advantage is that it is very general and fulfills basic requirements such as the Onsager–Casimir principle [96]. However, because it is so general, it of course cannot provide explicit expressions for the transport coefficients. Furthermore, we shall see below that imposing the second law of thermodynamics does not imply that the corresponding theory is causal and stable. As a preparatory step, we first present the covariant formulation of thermodynamics of a system in local thermodynamical equilibrium. This will naturally lead to the generalization to a system out of local equilibrium, which is discussed in the following subsections.

The entropy 4-current in local thermodynamical equilibrium is defined as
(48)$$S_{0}^{\mu }=s_{0}u^{\mu }\;,$$
where, according to Euler’s relation, $s_{0}\equiv \beta_{0}\left(\varepsilon_{0}+p_{0}\right)-\alpha_{0}n_{0}$ is the entropy density. Defining the time-like 4-vector
(49)$$\beta^{\mu }\equiv \beta_{0}u^{\mu }\;,$$
we first introduce a covariant version of Euler’s relation,
(50)$$S_{0}^{\mu }=p_{0}\beta^{\mu }+T_{0}^{\mu \nu }\beta_{\nu }-\alpha_{0}N_{0}^{\mu }\;,$$
which is consistent with Equation (Equation 48), as can be readily proven by inserting the local-equilibrium particle 4-current $N_{0}^{\mu }$ and energy-momentum tensor $T_{0}^{\mu \nu }$ from Equations ([Disp-formula FD8a-entropy-26-00189]). Following Israel and Stewart [8,97], we then postulate a covariant version of the first law of thermodynamics,
(51)$$dS_{0}^{\mu }=\beta_{\nu }dT_{0}^{\mu \nu }-\alpha_{0}dN_{0}^{\mu }\;.$$
From this and the covariant Euler relation (Equation 50) follows a covariant version of the Gibbs-Duhem relation
(52)$$d\left(p_{0}\beta^{\mu }\right)=N_{0}^{\mu }d\alpha_{0}-T_{0}^{\mu \nu }d\beta_{\nu }\;.$$
The covariant relations (Equation 50)–(Equation 52) are defined such that when contracted with $u_{\mu }$, using $u_{\mu }du^{\mu }=0$ they reduce to the standard thermodynamic relations. They do not contain more information than the latter because when projecting onto the 3-space orthogonal to $u^{\mu }$, they are trivially fulfilled [48].

The first law of thermodynamics (Equation 51) now leads to a conservation equation for the entropy 4-current,
(53)$$\partial \cdot S_{0}=\beta_{\nu }\partial_{\mu }T_{0}^{\mu \nu }-\alpha_{0}\partial \cdot N_{0}=0\;,$$
where we used the equations of motion ([Disp-formula FD3a-entropy-26-00189]) with the particle 4-current and the energy-momentum tensor replaced by their local-equilibrium counterparts ([Disp-formula FD8a-entropy-26-00189]). Equation (Equation 53) is nothing but the statement that, in local thermodynamical equilibrium, entropy is conserved. In tensor-decomposed form, it reads
(54)$$\partial \cdot S_{0}={\dot{s}}_{0}+s_{0}\theta =0\;.$$

### 4.2. Navier–Stokes Theory

In the presence of dissipative currents, entropy is no longer conserved. We now derive the form of the entropy 4-current in this case. We do this order by order in powers of the dissipative currents. At first order, we will recover Navier–Stokes theory, while at second order, we will derive the phenomenological Israel–Stewart equations, cf. Section 4.3. In order to preserve Lorentz covariance, the entropy 4-current must assume the form
(55)$$S^{\mu }=S_{0}^{\mu }+Q^{\mu }\;,$$
with the local-equilibrium entropy 4-current (Equation 48) and a yet-to-be-specified 4-current $Q^{\mu }$, which must be a function of the dissipative currents, such that in local thermodynamical equilibrium $Q_{0}^{\mu }=0$. Assuming the matching conditions (Equation 16), we may decompose the full particle 4-current and energy-momentum tensor as
(56a)$$\begin{array}{cc} N^{\mu } & =N_{0}^{\mu }+n^{\mu }\;, \end{array}$$
(56b)$$\begin{array}{cc} T^{\mu \nu } & =T_{0}^{\mu \nu }-\Pi {∆}^{\mu \nu }+\pi^{\mu \nu }\;, \end{array}$$
with $N_{0}^{\mu }$, $T_{0}^{\mu \nu }$ given by Equations ([Disp-formula FD8a-entropy-26-00189]). For the equilibrium quantities, Equation (Equation 51) still remains valid, such that
(57)$$\partial \cdot S_{0}=\beta_{0}u_{\nu }\partial_{\mu }T_{0}^{\mu \nu }-\alpha_{0}\partial \cdot N_{0}\;,$$
but now the right-hand side of Equation (Equation 53) no longer vanishes, as in Equation (Equation 53), since the dissipative currents are non-zero. In fact, using the conservation equations ([Disp-formula FD3a-entropy-26-00189]) with Equations ([Disp-formula FD56a-entropy-26-00189]) we obtain
(58)$$\partial \cdot S_{0}=\alpha_{0}\partial \cdot n+\beta_{0}\left(-\Pi \theta +\pi^{\mu \nu }\sigma_{\mu \nu }\right)\;.$$
By decomposing the first term on the right-hand side as $\alpha_{0}\partial \cdot n=\partial_{\mu }\left(\alpha_{0}n^{\mu }\right)-n\cdot I$, Equation (Equation 58) can be written in a more convenient form,
(59)$$\partial_{\mu }\left(S_{0}^{\mu }-\alpha_{0}n^{\mu }\right)=-n\cdot I-\beta_{0}\Pi \theta +\beta_{0}\pi^{\mu \nu }\sigma_{\mu \nu }\equiv Q\;.$$
It is very tempting to identify the term on the left-hand side of Equation (Equation 59) as the 4-divergence of the (non-equilibrium) entropy 4-current
(60)$$S^{\mu }\equiv S_{0}^{\mu }-\alpha_{0}n^{\mu }=s_{0}u^{\mu }-\alpha_{0}n^{\mu }\;,$$
and the terms on the right-hand side, *Q*, as the source terms for entropy production. This identification was proposed by Eckart [2] and by Landau and Lifshitz [1] and we adapt it here to derive relativistic Navier–Stokes theory[note 6].

Relativistic Navier–Stokes theory is then obtained by applying the second law of thermodynamics to each fluid element, i.e., by requiring that the entropy production obtained in Equation (Equation 59) must always be positive semi-definite, Q≥0. The simplest way to satisfy this condition for all possible fluid configurations is to assume that Π, $n^{\mu }$, and $\pi^{\mu \nu }$ are given by Equations ([Disp-formula FD34a-entropy-26-00189]), with the bulk-viscosity coefficient ζ, the particle-diffusion coefficient ϰ, and the shear-viscosity coefficient η. Then, substituting Equations ([Disp-formula FD34a-entropy-26-00189]) into Equation (Equation 59), entropy production becomes a quadratic function of the dissipative currents,
(61)$$Q=\frac{\beta_{0}}{\zeta }\Pi^{2}-\frac{1}{\varkappa }n\cdot n+\frac{\beta_{0}}{2\eta }\pi_{\mu \nu }\pi^{\mu \nu }\;.$$
Note that n·n<0, since $n^{\mu }$ is a space-like vector, while $\pi_{\mu \nu }\pi^{\mu \nu }$ is positive and, therefore, as long as ζ, ϰ, η>0, *Q* is, in fact, always positive semi-definite. The equations of fluid dynamics are obtained by substituting Equations ([Disp-formula FD34a-entropy-26-00189]) into the conservation laws ([Disp-formula FD4a-entropy-26-00189]), resulting in Equations ([Disp-formula FD35a-entropy-26-00189]).

As already mentioned, Navier–Stokes theory is acausal and, consequently, unstable. Thus, it is unable to describe any relativistic fluid existing in Nature. The source of the acausality can be understood from the constitutive relations satisfied by the dissipative currents, Equations ([Disp-formula FD34a-entropy-26-00189]). Such linear relations imply that any inhomogeneity of $\alpha_{0}$, $\beta_{0}$, and $u^{\mu }$, will *instantaneously* give rise to a dissipative current. This instantaneous creation of currents from (space-like) gradients of the primary fluid-dynamical variables leads to second-order gradients in the equations of motion ([Disp-formula FD35a-entropy-26-00189]) and renders them non-hyperbolic. In a relativistic theory, this leads to instabilities [63].

### 4.3. Israel–Stewart Theory

It is possible to derive stable and causal fluid-dynamical equations from the second law of thermodynamics. First, note that the entropy current (Equation 60) of Navier–Stokes theory is of *first order* in the dissipative quantities: the term proportional to $n^{\mu }$ is a first-order term allowed by Lorentz symmetry (the other one would be $∼\beta_{0}\Pi u^{\mu }$, but this term cannot occur if the entropy is supposed to assume its maximum value in equilibrium). As shown by Israel and Stewart [8,99], to arrive at a stable and causal theory of relativistic dissipative fluid dynamics, one has to extend the entropy current (Equation 60) to include terms of *second order* in dissipative quantities, i.e., $S^{\mu }$ must be a function of quadratic order in Π, $n^{\mu }$, and $\pi^{\mu \nu }$. Thus [8,99],
(62)$$\begin{array}{ccc} S^{\mu } & = & S_{0}^{\mu }-\alpha_{0}n^{\mu }+Q^{\mu }+O_{3}\;, \end{array}$$
where $O_{3}$ denotes terms of third order or higher in the dissipative currents and
(63)$$Q^{\mu }\equiv -\frac{1}{2}u^{\mu }\left(\delta_{0}\Pi^{2}-\delta_{1}n\cdot n+\delta_{2}\pi_{\alpha \beta }\pi^{\alpha \beta }\right)-\gamma_{0}\Pi n^{\mu }-\gamma_{1}\pi_{\;\nu }^{\mu }n^{\nu }$$
is of second order, $Q^{\mu }∼O_{2}$. The expansion coefficients, $\delta_{0}$, $\delta_{1}$, $\delta_{2}$, $\gamma_{0}$, and $\gamma_{1}$, are functions of $\alpha_{0}$ and $\beta_{0}$ and can for instance be obtained by matching this expansion with an underlying microscopic theory. Note that the entropy 4-current used to derive relativistic Navier–Stokes theory is recovered by taking $Q^{\mu }=0$. It is important to remember that $Q^{\mu }$ is not orthogonal to the fluid 4-velocity and, consequently,
(64)$$s\equiv S\cdot u=s_{0}+Q\cdot u=s_{0}-\frac{1}{2}\left(\delta_{0}\Pi^{2}-\delta_{1}n\cdot n+\delta_{2}\pi_{\alpha \beta }\pi^{\alpha \beta }\right)\neq s_{0}\;,$$
i.e., the non-equilibrium entropy density in the local rest frame, *s*, does not equal the entropy density computed using the fictitious equilibrium state, $s_{0}\left(\varepsilon_{0},n_{0}\right)$.

We now calculate the entropy production using the more general entropy 4-current (Equation 62),
(65)$$\partial \cdot S=\beta_{0}u_{\nu }\partial_{\mu }T_{0}^{\mu \nu }-\alpha_{0}\partial \cdot N_{0}-\partial_{\mu }\left(\alpha_{0}n^{\mu }\right)+\partial \cdot Q\;,$$
where we employed Equation (Equation 51). The conservation laws ([Disp-formula FD4a-entropy-26-00189]) then lead to
(66)$$\partial \cdot S=-\beta_{0}\Pi \theta +\beta_{0}\pi^{\mu \nu }\sigma_{\mu \nu }-n\cdot I+\partial \cdot Q\;.$$
Using Equation (Equation 63), we derive
(67)$$\begin{array}{ccc} \partial \cdot Q & = & -\delta_{0}\Pi \;\dot{\Pi }+\delta_{1}n\cdot \dot{n}-\delta_{2}\pi_{\mu \nu }{\dot{\pi }}^{\mu \nu }-\frac{1}{2}\left(\Pi^{2}{\dot{\delta }}_{0}-n\cdot n\;{\dot{\delta }}_{1}+\pi_{\mu \nu }\pi^{\mu \nu }{\dot{\delta }}_{2}\right)\\ & & -\frac{1}{2}\left(\delta_{0}\Pi^{2}-\delta_{1}n\cdot n+\delta_{2}\pi_{\mu \nu }\pi^{\mu \nu }\right)\theta -\gamma_{0}\Pi \;\partial \cdot n-\gamma_{0}\;n\cdot \nabla \Pi -\Pi \;n\cdot \nabla \gamma_{0}\\ & & -\gamma_{1}\pi_{\mu \nu }\nabla^{\langle \mu }n^{\nu \rangle }-\pi_{\mu \nu }n^{\langle \mu }\nabla^{\nu \rangle }\gamma_{1}-\gamma_{1}n_{\nu }\partial_{\mu }\pi^{\mu \nu }\;. \end{array}$$
Then, substituting Equation (Equation 67) into Equation (Equation 66), we obtain the more general entropy-production equation
(68)$$\begin{array}{c} \partial \cdot S=\beta_{0}\Pi \left(-\theta -\frac{\delta_{0}}{\beta_{0}}\dot{\Pi }-\frac{{\dot{\delta }}_{0}}{2\beta_{0}}\Pi -\frac{\delta_{0}}{2\beta_{0}}\Pi \theta -\frac{\gamma_{0}}{\beta_{0}}\;\partial \cdot n-\frac{1-r}{\beta_{0}}n\cdot \nabla \gamma_{0}\right)\\ \;\;\;\;\;+n_{\mu }\left(-I^{\mu }+\delta_{1}{\dot{n}}^{\langle \mu \rangle }+\frac{{\dot{\delta }}_{1}}{2}n^{\mu }+\frac{\delta_{1}}{2}n^{\mu }\theta -\gamma_{0}\nabla^{\mu }\Pi -r\Pi \nabla^{\mu }\gamma_{0}-\gamma_{1}\partial_{\nu }\pi^{\mu \nu }-y\pi^{\mu \nu }\nabla_{\nu }\gamma_{1}\right)\\ \;\;\;\;\;+\beta_{0}\pi_{\mu \nu }\left(\sigma^{\mu \nu }-\frac{\delta_{2}}{\beta_{0}}{\dot{\pi }}^{\langle \mu \nu \rangle }-\frac{{\dot{\delta }}_{2}}{2\beta_{0}}\pi^{\mu \nu }-\frac{\delta_{2}}{2\beta_{0}}\pi^{\mu \nu }\theta -\frac{\gamma_{1}}{\beta_{0}}\nabla^{\langle \mu }n^{\nu \rangle }-\frac{1-y}{\beta_{0}}n^{\langle \mu }\nabla^{\nu \rangle }\gamma_{1}\right)\;, \end{array}$$
where r,y are arbitrary constants. This arbitrariness arises because it is ambiguous how to distribute the respective terms in Equation (Equation 67) between the first and second, and second and third lines of Equation (Equation 68), respectively.

As argued before, the only way to explicitly satisfy the second law of thermodynamics is to assure that the entropy production is a positive semi-definite quadratic function of the dissipative currents, i.e.,
(69)$$\partial \cdot S\equiv Q=\frac{\beta_{0}}{\zeta }\Pi^{2}-\frac{1}{\varkappa }n\cdot n+\frac{\beta_{0}}{2\eta }\pi_{\mu \nu }\pi^{\mu \nu }\;,$$
cf. Equation (Equation 61), with Q≥0 and therefore positive definite transport coefficients ζ, ϰ, and η. This further implies that the dissipative currents must satisfy the *dynamical* equations of the form ([Disp-formula FD23a-entropy-26-00189]), with the relaxation times given by
(70)$$\tau_{\Pi }\equiv \zeta \frac{\delta_{0}}{\beta_{0}}\;,\;\tau_{n}\equiv \varkappa \delta_{1}\;,\;\tau_{\pi }\equiv 2\eta \frac{\delta_{2}}{\beta_{0}}\;,$$
and the other second-order transport coefficients are given as follows:
(71a)$$\begin{array}{cc} \delta_{\Pi \Pi } & \equiv \frac{\tau_{\Pi }}{2}\left(1+A\frac{\partial \ln\delta_{0}}{\partial \alpha_{0}}+B\frac{\partial \ln\delta_{0}}{\partial \beta_{0}}\right)\;,\;\ell_{\Pi n}\equiv \frac{\tau_{\Pi }\gamma_{0}}{\delta_{0}}\;,\\ \lambda_{\Pi n} & \equiv \ell_{\pi n}\left(1-r\right)\left(\frac{\partial \ln\gamma_{0}}{\partial \alpha_{0}}+\frac{n_{0}}{\varepsilon_{0}+p_{0}}\frac{\partial \ln\gamma_{0}}{\partial \beta_{0}}\right)\;,\\ \tau_{\Pi n} & \equiv -\ell_{\Pi n}\frac{1}{\varepsilon_{0}+p_{0}}\left[1+\left(1-r\right)\frac{\partial \ln\gamma_{0}}{\partial \ln\beta_{0}}\right]\;,\;\lambda_{\Pi \pi }\equiv 0\;, \end{array}$$
(71b)$$\begin{array}{cc} \delta_{nn} & \equiv \frac{\tau_{n}}{2}\left(1+A\frac{\partial \ln\delta_{1}}{\partial \alpha_{0}}+B\frac{\partial \ln\delta_{1}}{\partial \beta_{0}}\right)\;,\;\ell_{n\Pi }\equiv -\frac{\tau_{n}\gamma_{0}}{\delta_{1}}\;,\;\ell_{n\pi }\equiv \tau_{n}\frac{\gamma_{1}}{\delta_{1}}\;,\\ \tau_{n\Pi } & \equiv \frac{r\ell_{n\Pi }}{\varepsilon_{0}+p_{0}}\frac{\partial \ln\gamma_{0}}{\partial \ln\beta_{0}}\;,\;\tau_{n\pi }\equiv \frac{\ell_{n\pi }}{\varepsilon_{0}+p_{0}}\left(1+y\frac{\partial \ln\gamma_{1}}{\partial \ln\beta_{0}}\right)\;,\;\lambda_{nn}\equiv 0\;,\\ \lambda_{n\Pi } & \equiv -r\ell_{n\Pi }\left(\frac{\partial \ln\gamma_{0}}{\partial \alpha_{0}}+\frac{n_{0}}{\varepsilon_{0}+p_{0}}\frac{\partial \ln\gamma_{0}}{\partial \beta_{0}}\right)\;,\;\lambda_{n\pi }\equiv -y\ell_{n\pi }\left(\frac{\partial \ln\gamma_{1}}{\partial \alpha_{0}}+\frac{n_{0}}{\varepsilon_{0}+p_{0}}\frac{\partial \ln\gamma_{1}}{\partial \beta_{0}}\right)\;, \end{array}$$
(71c)$$\begin{array}{cc} \tau_{n\omega } & \equiv 0\;,\\ \delta_{\pi \pi } & \equiv \frac{\tau_{\pi }}{2}\left(1+A\frac{\partial \ln\delta_{2}}{\partial \alpha_{0}}+B\frac{\partial \ln\delta_{2}}{\partial \beta_{0}}\right)\;,\;\ell_{\pi n}\equiv -\frac{\tau_{\pi }\gamma_{1}}{\delta_{2}}\;,\;\tau_{\pi n}\equiv \ell_{\pi n}\frac{1-y}{\varepsilon_{0}+p_{0}}\frac{\partial \ln\gamma_{1}}{\partial \ln\beta_{0}}\;,\\ \lambda_{\pi \Pi } & =\lambda_{\pi \pi }\equiv 0\;,\;\lambda_{\pi n}\equiv \ell_{\pi n}\left(1-y\right)\left(\frac{\partial \ln\gamma_{1}}{\partial \alpha_{0}}+\frac{n_{0}}{\varepsilon_{0}+p_{0}}\frac{\partial \ln\gamma_{1}}{\partial \beta_{0}}\right)\;,\;\tau_{\pi \omega }\equiv 0\;. \end{array}$$
Here, we have made use of Equation (Equation 257) to convert comoving derivatives of $\alpha_{0}$, $\beta_{0}$ into terms ∼θ. All other transport coefficients, i.e., those in the K- and R-terms vanish. Since the relaxation times (Equation 70) must be positive, the expansion coefficients $\delta_{0}$, $\delta_{1}$, and $\delta_{2}$ must all be larger than zero. This implies, inspecting Equation (Equation 64), that $s\leq s_{0}$, i.e., the entropy density in the non-equilibrium state is smaller than in equilibrium (as it should be because the equilibrium state maximizes the entropy).

Note that several transport coefficients vanish, i.e., the equations of motion for the dissipative currents derived using the second law of thermodynamics do not feature all possible terms that may appear. In Section 5, we shall see which terms precisely are missing in this type of derivation when we derive the equations of fluid dynamics from microscopic theory. A derivation from microscopic theory also allows us to fix the as-of-yet arbitrary constants r,y uniquely.

For the sake of completeness, we note that in [80], the most general Israel–Stewart equations of motion that arise from entropy-production arguments were determined in a general fluid-dynamical frame. Those equations reduce to the ones shown in this section when written in the Landau frame. Furthermore, see [100] for a derivation of the Israel–Stewart equations for multiple conserved charges from the entropy-production arguments discussed in this section.

Finally, we emphasize that Navier–Stokes theory can be derived from second-law arguments alone, as discussed above, and such a theory is acausal and unstable. Therefore, imposing the validity of the second law of thermodynamics alone is insufficient to derive viable relativistic dissipative fluid-dynamical theories. This may seem puzzling at first, given the importance and usual constraining power of the second law of thermodynamics in non-relativistic fluid dynamics [1]. This matter was clarified in a series of papers published recently [101,102,103].

In the original work of Hiscock and Lindblom [49], the stability of Israel–Stewart theory was demonstrated using a very convenient quadratic functional of the fluctuations around equilibrium, whose properties are defined by the solutions of the linearized equations of motion. This method is very powerful, as it is inherently covariant and valid for demonstrating stability around rotating global-equilibrium states. At the time, as remarked by Hiscock and Lindblom, there was no systematic method for obtaining such a functional. This was remedied in [102] by introducing the so-called Gibbs stability criterion, which systematically establishes thermodynamic stability in relativistic systems that strictly obey the second law of thermodynamics. Incidentally, this also implies causality in the linear regime, as shown in [103], and a very nice interpretation of this result in terms of information theory has also been presented in the same work. This new development, motivated initially to understand the connections between stability and causality in relativistic fluids, has now been applied to solve outstanding problems in related areas, such as the definition and construction of new universality classes in fluid dynamics [104,105], and the formulation of relativistic stochastic fluid dynamics [38,39].

### 4.4. Divergence-Type Theories

This subsection briefly summarizes relativistic fluid-dynamical theories of divergence type [106]. Such theories provide a far-reaching subject with many important contributions to the physics of fluids, kinetic theory, and out-of-equilibrium phenomena. However, a comprehensive summary of these results is beyond the scope of this work as this would be a significant task that requires a dedicated review by itself. Given that this is not discussed in detail in other works, here we focus on statements concerning causality, local well-posedness, and stability of divergence-type theories, which can be compared to other approaches such as BDNK theory and Israel–Stewart-like approaches. In this subsection, we use a “mostly plus” metric signature to compare to the original notation and results of [106].

In a nutshell, the explanation for why divergence-type theories do not necessarily provide a stable, causal, strong-hyperbolic, and local well-posed theory of relativistic dissipative fluids is that divergence-type theories are more of a general formalism and not a specific theory of fluids. This general formalism can accommodate theories with varying and opposing physical and mathematical properties without providing systematic principles for constructing specific models where one or more of the desired properties (stability, causality, etc.) can be enforced. This is clearly illustrated by the fact that Eckart’s theory, known to be acausal and unstable, is formally of divergence type [106]. This situation is well summarized in [14] (Chapter 6), where the authors state that “the construction of a formulation that is cast in a divergence type is not, per se, sufficient to guarantee hyperbolicity”.

We will now briefly explain the remarks above in more detail. We follow closely what was done in [106], which provides a systematic yet concise study of divergence-type theories[note 7], with further input from [7,14,107]. More recent results on divergence-type theories are discussed further below.

A divergence-type theory is a fluid-dynamical theory satisfying the following properties: (i) The dynamical variables can be taken to be the particle-number current $N^{\mu }$ and the (symmetric) energy-momentum tensor $T^{\mu \nu }$; (ii) the dynamical equations are the conservation laws
(72a)$$\begin{array}{cc} \partial_{\mu }N^{\mu } & =0\;, \end{array}$$
(72b)$$\begin{array}{cc} \partial_{\mu }T^{\mu \nu } & =0\;, \end{array}$$
(72c)$$\begin{array}{cc} \partial_{\mu }F^{\mu \nu \sigma } & =I^{\nu \sigma }\;, \end{array}$$
where $F^{\mu \nu \sigma }$ and $I^{\nu \sigma }$ are algebraic functions of $N^{\mu }$ and $T^{\mu \nu }$ and the tensors $F^{\mu \nu \sigma }$ and $I^{\nu \sigma }$ are symmetric and trace-free in ν and σ; (iii) there exists an entropy current $S^{\mu }$ that satisfies
(73)$$\begin{array}{c} \partial_{\mu }S^{\mu }=A\;, \end{array}$$
where *A* is an algebraic function of $N^{\mu }$ and $T^{\mu \nu }$. Observe that the set $\left(N^{\mu },T^{\mu \nu }\right)$ carries 14 degrees of freedom, i.e., the same number of equations as in Equations ([Disp-formula FD72a-entropy-26-00189]).

Within a kinetic-theory approach (see Section 5), the fields $N^{\mu }$, $T^{\mu \nu }$, and $F^{\mu \nu \sigma }$ are the first, second, and third moments of the distribution function and $I^{\nu \sigma }$ is the source term in the equation of motion for the third moment of the distribution function. Thus, one can compute exactly what these fields are from a microscopic kinetic approach. However, in practice, one is interested in constructing a fluid-dynamical theory precisely because one cannot solve the full microscopic theory, so one seeks an effective theory that “integrates out” the short-wavelength, high-frequency dynamics. This is accomplished by some coarse-graining procedure, which, inevitably, introduces some *modeling choices* that can lead to different equations of motion. Thus, the choice of the functional form for the dependence of $F^{\mu \nu \sigma }$, $I^{\nu \sigma }$, $S^{\mu }$, and *A* in terms of $N^{\mu }$ and $T^{\mu \nu }$, which define the *constitutive relations*, can be viewed as parametrizing different coarse-graining procedures.

As it stands, the above formalism remains too general to be applied to concrete problems. Thus, in divergence-type theories, one further assumes that (iv) there exists a generating function χ, which is an algebraic function of a new set of dynamical variables $\left(\zeta ,\zeta_{\mu },\zeta_{\mu \nu }\right)$, with $\zeta_{\mu \nu }$ trace-free and symmetric, such that
(74a)$$\begin{array}{cc} N^{\mu } & =\frac{\partial^{2}\chi }{\partial \zeta \partial \zeta_{\mu }}\;, \end{array}$$
(74b)$$\begin{array}{cc} T^{\mu \nu } & =\frac{\partial^{2}\chi }{\partial \zeta_{\mu }\partial \zeta_{\nu }}\;, \end{array}$$
(74c)$$\begin{array}{cc} F^{\mu \nu \sigma } & =\frac{\partial^{2}\chi }{\partial \zeta_{\mu }\partial \zeta_{\nu \sigma }}\;, \end{array}$$
(74d)$$\begin{array}{cc} S^{\mu } & =\frac{\partial \chi }{\partial \zeta_{\mu }}-\zeta N^{\mu }-\zeta_{\nu }T^{\mu \nu }-\zeta_{\nu \sigma }F^{\mu \nu \sigma }\;, \end{array}$$
(74e)$$\begin{array}{cc} A & =-\zeta_{\nu \sigma }I^{\nu \sigma }\;. \end{array}$$
Although assumption (iv) might seem ad hoc at first sight, it is motivated by the fact that conditions (i)–(iii) plus a set of natural assumptions on the constitutive relations imply the existence of such a χ ([14], Chapter 6).

Equations ([Disp-formula FD74a-entropy-26-00189]) can be viewed as a change of variables[note 8] from the dynamical variables $\left(N^{\mu },T^{\mu \nu }\right)$ to the new dynamical variables $\left(\zeta ,\zeta_{\mu },\zeta_{\mu \nu }\right)$, with Equations ([Disp-formula FD74c-entropy-26-00189]) plus a choice of $I^{\nu \sigma }$ providing the constitutive relations. In practice, a divergence-type theory is determined by specifying a generating function χ, defining the fields of the theory by Equations ([Disp-formula FD74a-entropy-26-00189]), and imposing the equations of motion ([Disp-formula FD72a-entropy-26-00189]) (Equation (Equation 73) will be automatically satisfied). Writing $\zeta_{A}$ to represent the variables $\left(\zeta ,\zeta_{\mu },\zeta_{\mu \nu }\right)$ and $I^{A}$ to represent the source $I^{\nu \sigma }$, i.e., $\zeta_{A}=\left(\zeta ,\zeta_{\mu },\zeta_{\mu \nu }\right)$ and $I^{A}=\left(0,0,I^{\nu \sigma }\right)$, the equations of motion ([Disp-formula FD72a-entropy-26-00189]) can be written as
(75)$$\begin{array}{c} \frac{\partial^{2}\chi^{\mu }}{\partial \zeta_{A}\partial \zeta_{B}}\partial_{\mu }\zeta_{B}=I^{A}\;, \end{array}$$
where
$$\begin{array}{c} \chi^{\mu }=\frac{\partial \chi }{\partial \zeta_{\mu }}\;. \end{array}$$
The matrices $M^{\mu }$ are given by
$$\begin{array}{c} {\left(M^{\mu }\right)}^{AB}\equiv M^{AB\mu }=\frac{\partial^{2}\chi^{\mu }}{\partial \zeta_{A}\partial \zeta_{B}}\;, \end{array}$$
and they are symmetric since $\frac{\partial^{2}\chi^{\mu }}{\partial \zeta_{A}\partial \zeta_{B}}=\frac{\partial^{2}\chi^{\mu }}{\partial \zeta_{B}\partial \zeta_{A}}$. Thus, the system’s evolution cast in terms of the variables $\zeta_{A}$, i.e., following Equation (Equation 75), is automatically a *symmetric* system of first-order partial differential equations. Following standard definitions [56], the system (Equation 75) will be *hyperbolic* in the direction of a future-directed time-like vector $w_{\mu }$ if $M^{AB\mu }w_{\mu }$ is negative definite[note 9], and the system will be *causal* (thus consistent with the principles of relativity) if $M^{AB\mu }w_{\mu }$ is negative definite for all future-directed time-like vectors[note 10]$w_{\mu }$.

While the above presentation of divergence-type theories was very concise, it should be clear that very little can be said at this level of generality. As it stands, the actual fluid content of a theory resides in the choice of a generating function. To the best of our knowledge, *there is no systematic principle guiding the choice of χ.* That is not to say that informed choices of χ cannot be made based on physical intuition or other motivating factors. Therefore, while some physical principles are relatively easy to enforce—for example, one wants to enforce A≥0 in view of the second law of thermodynamics (see Equations (Equation 73) and ([Disp-formula FD74e-entropy-26-00189])); other principles, such as causality, depend on the detailed structure of the generating function χ and are much harder to fulfill.

Most importantly, one wants more than obtaining a hyperbolic and causal theory for some arbitrary set of variables $\zeta_{A}$—one needs to make connections with actual fluid properties. Thus, it is not enough to write down a generating function χ in terms of some general variables $\zeta_{A}$ for which hyperbolicity and causality hold. One needs some well-defined and physically meaningful procedure that allows to recover quantities such as the flow velocity, the energy density, etc., out of the variables $\zeta_{A}$.

One could imagine that the difficulties outlined in the above two paragraphs are not major obstacles, i.e., that it is only a matter of investing enough effort in finding a “right” generating functional that gives a causal and hyperbolic theory for which one can clearly identify the variables that play the role of the energy density, the flow velocity, etc. While this might certainly be the case, as we explain below, the results obtained so far within the framework of divergence-type theories are far from achieving this goal.

In [106], the authors provide two important examples—i.e., two generating functions—of fluid-dynamical theories of divergence type, which we now discuss. Their first example is simply Eckart’s theory (which is neither causal nor stable). In doing so, the authors do not write explicitly the terms in first-order derivatives of the velocity, etc., as is usually done in Eckart’s formalism. Rather, definitions of the flow velocity, shear stress, etc., are given in terms of $N^{\mu }$ and $T^{\mu \nu }$ using usual tensor decompositions, and these definitions are then related to $\zeta_{A}$. The evolution for $\zeta_{A}$ and, consequently, for the flow velocity, shear stress, etc., is obtained from Equation (Equation 75). One can, however, recognize that this indeed corresponds to Eckart’s theory in its familiar form if we consider the evolution of those quantities that involve first-order derivatives, such as the shear stress (recall that Equation (Equation 75) is a system of first-order partial differential equations).

Their second example involves a fluid theory in the variables $\zeta_{A}$, which is causal and hyperbolic, forming, thus, in particular, a first-order symmetric hyperbolic system. More precisely, the authors proceed as follows. They consider the same relations giving $\zeta_{A}$ in terms of the $\left(N^{\mu },T^{\mu \nu }\right)$ (thus giving the flow velocity, shear stress, etc., in terms of $\left(N^{\mu },T^{\mu \nu }\right)$) as in Eckart’s theory[note 11] but modify the generating function. They establish hyperbolicity and causality as follows. First, they consider the matrices $M^{AB\mu }$ evaluated in perfect-fluid states ${\left.\zeta_{A}\right|}_{per}$, i.e., states where dissipative contributions vanish, obtaining causality and hyperbolicity (provided standard conditions for the perfect fluid hold, e.g., the sound speed is not superluminal). Next, they argue that since the causality and hyperbolicity conditions vary continuously with $\zeta_{A}$, since they hold for ${\left.\zeta_{A}\right|}_{per}$, they must thus also hold for $\zeta_{A}$ sufficiently close to ${\left.\zeta_{A}\right|}_{per}$. While this argument has merits, one should not overlook that it is *qualitative* and, consequently, significantly limited for practical studies of relativistic dissipative fluids. *No quantitative estimate of how close $\zeta_{A}$ needs to be to ${\left.\zeta_{A}\right|}_{per}$ in order to guarantee causality and hyperbolicity is given.*

This should be contrasted with the results for BDNK theory, where exact inequalities among the transport coefficients and fluid-dynamical variables are obtained as causality and hyperbolicity conditions, and also with inequalities of the same sort that ensure causality of the Israel–Stewart theory [54]. Needless to say, one needs such quantitative estimates for applications to concrete problems, including investigations of the causality of numerical solutions. We should add that obtaining quantitative bounds of this nature for divergence-type theories seems daunting: this would require a careful analysis of the system’s characteristics and the properties of the matrices $M^{AB\mu }$. There is no reason to expect that this analysis would be significantly simpler than the one carried out for the BDNK [52] and Israel–Stewart theories [54], which required developing a series of new ideas *tailored to those theories specifically.*

The results of [106] have been extended in [107], where a more thorough and systematic study of divergence-type theories has been carried out. In particular, more general divergence-type theories have been introduced in [107]. Concerning causality and hyperbolicity, however, the results obtained in [107] are not different from the ones reviewed above: they are qualitative, asserting causality and hyperbolicity for fluid states significantly close to ${\left.\zeta_{A}\right|}_{per}$ without quantitative estimates. Similarly, in [108,109], qualitative causality and hyperbolicity results are obtained.

More recently, the formalism of divergence-type theories has been applied to study relativistic dissipative fluids in the context of heavy-ion collisions in [110,111,112], for the case of a conformal fluid see also [113,114]. Once again, the results are valid for $\zeta_{A}$ sufficiently close to ${\left.\zeta_{A}\right|}_{per}$ without quantifying how close this has to be. In summary, divergence-type theories provide a rich formalism for studying relativistic dissipative fluids. When it comes to causality and local well-posedness, however, the results obtained for divergence-type theories lack specificity and applicability as compared to those obtained for BDNK theory [52] and, to a lesser degree, Israel–Stewart-like theories [53,54], where such properties can be easily checked by verifying a set of inequalities involving the dynamical variables.

## 5. Dissipative Fluid Dynamics from Kinetic Theory

At a conceptual level, the thermodynamic arguments and general consistency properties such as causality and stability are extremely important to obtain a valid fluid-dynamical theory. However, they are insufficient for explicitly deriving the values of the various transport coefficients that enter dissipative fluid dynamics. To that end, a non-equilibrium microscopic theory is necessary. Then, a suitable power-counting procedure is applied to reduce that theory’s non-equilibrium degrees of freedom to the fluid-dynamical ones, i.e., the components of $N^{\mu }$ and $T^{\mu \nu }$.

In this section, we discuss how some variants of relativistic dissipative fluid dynamics can be obtained from relativistic kinetic theory in the form of the Boltzmann equation [48,115]. The conservation laws, the cornerstone of any fluid-dynamical formulation, emerge naturally as the zeroth and first moment of the Boltzmann equation. However, to construct a fluid-dynamical theory, more information is needed in the form of constitutive relations for the dissipative currents, which provide the necessary information to close the system of equations of motion. Such equations are expected to appear in the so-called fluid-dynamical regime, in which the fields $\alpha_{0}$, $\beta_{0}$, and $u^{\mu }$ vary very slowly in space and time when compared to the typical microscopic scales of the system. This wide separation of scales, quantified by the Knudsen number (cf. Section 2.4), allows a derivation of an effective theory that can describe the fluid in terms of a reduced number of degrees of freedom. We shall see that different implementations of this idea lead to distinct fluid-dynamical theories. First, we shall discuss the microscopic derivation of Navier–Stokes theory within the Chapman–Enskog procedure, which solves the Boltzmann equation in a perturbative scheme [48,115]. After that, we discuss a recently proposed modified version of this Chapman–Enskog expansion necessary for the derivation of BDNK theory [88,116], which has emerged as an alternative to remedy the infamous acausality and instability problems of Navier–Stokes theory. Finally, we discuss transient second-order theories of fluid dynamics, in which the dissipative currents obey independent equations of motion. We divide the microscopic derivation of these equations of motion into two parts. In the first part, the traditional 14-moment truncation procedure is discussed. The second part is devoted to resummed transient fluid-dynamical theories, which have been employed more recently.

### 5.1. Boltzmann Equation, Fluid-Dynamical Quantities, Local Equilibrium, and Matching Conditions

Let us consider a relativistic, sufficiently dilute gas of monatomic particles, whose non-equilibrium state is determined by the single-particle distribution function f(x,k) in phase space. The dynamics of the single-particle distribution function is given by the Boltzmann equation. In the absence of external forces (e.g., a strong electromagnetic or gravitational field) and considering that the system is composed of only one particle species that undergoes binary elastic collisions, the relativistic Boltzmann equation reads
(76)$$k\cdot \partial f_{k}=\frac{1}{2}\int dQ\;dQ^{\prime }\;dK^{\prime }\;W_{kk^{\prime }↔qq^{\prime }}\left({\tilde{f}}_{k}{\tilde{f}}_{k^{\prime }}f_{q}f_{q^{\prime }}-f_{k}f_{k^{\prime }}{\tilde{f}}_{q}{\tilde{f}}_{q^{\prime }}\right)\equiv C\left[f_{k}\right]\;,$$
where, for notational compactness we employ $f_{k}\equiv f\left(x,k\right)$ and ${\tilde{f}}_{k}=1-af_{k}$, with a=1 (a=−1) for fermions (bosons) and a→0 for classical particles obeying Maxwell–Boltzmann statistics. On the right-hand side, in the collision term $C[f_{k}]$, we also defined the Lorentz-invariant integration measure for on-shell particles ($k\cdot k=m^{2}$), $dK\equiv d^{3}k/\left[{\left(2\pi \right)}^{3}k^{0}\right]$ and the transition rate
(77)$$\begin{array}{cc} \; & W_{kk^{\prime }↔qq^{\prime }}={\left(2\pi \right)}^{6}s\sigma \left(s,\Theta \right)\delta^{\left(4\right)}\left(k^{\mu }+k^{\prime \mu }-q^{\mu }-q^{\prime \mu }\right)\;, \end{array}$$
where σ(s,Θ) is the differential cross section for the corresponding interaction. The transition rate is expressed in terms of the Mandelstam variable $s\equiv \left(k+k^{\prime }\right)\cdot \left(k+k^{\prime }\right)$ and the scattering angle Θ,
(78)$$\begin{array}{cc} \; & \cos\Theta \equiv \frac{\left(k-k^{\prime }\right)\cdot \left(q-q^{\prime }\right)}{{\left(k-k^{\prime }\right)}^{2}}=1+\frac{2t}{s-4m^{2}}={\left.\frac{k\cdot q}{\left|k||q\right|}\right|}_{CM}\;, \end{array}$$
where ${|}_{CM}$ denotes that the momenta are taken in the center-of-momentum frame and we have made use of the Mandelstam variable $t={\left(k-q\right)}^{2}={\left(k^{\prime }-q^{\prime }\right)}^{2}$.

The relativistic Boltzmann equation also emerges from quantum field theory [115,117,118,119,120,121]. In this case, a truncation of a hierarchy containing the full quantum non-equilibrium dynamics is performed. Then, the relevant phase-space distribution is the so-called Wigner function. With the latter, quantum averages are expressed as phase-space integrals of this function, in analogy with the classical case [115]. Nevertheless, since the Wigner function is not positive semi-definite, it cannot be interpreted as a *probability* distribution and, thus, the analogy with the classical case is incomplete. From the dynamics of the Wigner function, the Boltzmann equation is recovered at next-to-leading order in the semi-classical expansion. At higher orders, one arrives at kinetic equations that feature non-local collision terms due to quantum fluctuations. These non-localities are especially relevant in formulations of fluid dynamics that take into account the conservation of angular momentum, as in so-called spin hydrodynamics [122,123,124,125,126,127], whose general formulation shall be described in Section 7.4.

Macroscopic quantities are obtained from moments of the single-particle distribution function, i.e., momentum-space integrals over the latter weighted by powers of momentum. For instance, the particle-number 4-current and energy-momentum tensor, respectively, are given by
(79a)$$\begin{array}{cc} N^{\mu } & =\int dK\;k^{\mu }\;f_{k}\;, \end{array}$$
(79b)$$\begin{array}{cc} T^{\mu \nu } & =\int dK\;k^{\mu }k^{\nu }\;f_{k}\;. \end{array}$$
The local conservation laws ([Disp-formula FD3a-entropy-26-00189]) for these currents are obtained by integrating both sides of the Boltzmann equation (Equation 76), multiplied with 1 and $k^{\nu }$, respectively, over momentum space and using the fact that particle number and energy-momentum is conserved in binary elastic collisions, which gives rise to the following properties of the collision term:(80)$$\int dK\;C\left[f_{k}\right]=0\;,\;\int dK\;k^{\mu }\;C\left[f_{k}\right]=0\;.$$
The definitions ([Disp-formula FD79a-entropy-26-00189]) naturally imply the following expressions for the particle-number density *n*, the particle diffusion current $n^{\mu }$, the isotropic pressure *p*, the energy-momentum diffusion current $h^{\mu }$, and the shear-stress tensor $\pi^{\mu \nu }$, respectively,
(81a)$$\begin{array}{cc} n & \equiv \int dK\;E_{k}\;f_{k}\;,\;\varepsilon \equiv \int dK\;E_{k}^{2}\;f_{k}\;,\;p\equiv -\frac{1}{3}\int dK\;{∆}_{\mu \nu }k^{\mu }k^{\nu }\;f_{k}\;, \end{array}$$
(81b)$$\begin{array}{cc} n^{\mu } & \equiv \int dK\;k^{\langle \mu \rangle }\;f_{k}\;,\;h^{\mu }\equiv \int dK\;E_{k}\;k^{\langle \mu \rangle }f_{k}\;,\;\pi^{\mu \nu }\equiv \int dK\;k^{\langle \mu }k^{\nu \rangle }\;f_{k}\;, \end{array}$$
where $E_{k}\equiv u\cdot k$ is the energy of a particle with 4-momentum $k^{\mu }$ in the rest frame of the fluid element.

Another central concept in the kinetic-theory derivation of fluid dynamics is that of local equilibrium. This state can be defined using the H-theorem, which states that there exists a 4-vector functional of $f_{k}$, $S^{\mu }$, such that ∂·S≥0. One identifies this functional with the entropy 4-current and uses the condition $\partial \cdot S_{0}=0$ as the definition of a local-equilibrium state with entropy 4-current $S_{0}^{\mu }$, see also Section 4. Such a functional and its 4-divergence read
(82a)$$\begin{array}{cc} S^{\mu } & =-\int dK\;k^{\mu }\left(\frac{{\tilde{f}}_{k}}{a}\ln{\tilde{f}}_{k}+f_{k}\ln f_{k}\right)\;, \end{array}$$
(82b)$$\begin{array}{cc} \partial \cdot S & =-\int dK\;C\left[f_{k}\right]\left(\ln f_{k}-\ln{\tilde{f}}_{k}\right)\;, \end{array}$$
where the Boltzmann equation has been used in the derivation of the second equation. Then, from the assumption that ${\tilde{f}}_{k}$ and $f_{k}$ are non-negative functions, it can be shown that indeed ∂·S≥0 [115]. The particular case $\partial \cdot S_{0}=0$, which defines the local-equilibrium distribution $f_{0k}$, is attained if and only if the detailed-balance condition ${\tilde{f}}_{0k}{\tilde{f}}_{0k^{\prime }}f_{0q}f_{0q^{\prime }}=f_{0k}f_{0k^{\prime }}{\tilde{f}}_{0q}{\tilde{f}}_{0q^{\prime }}$ is satisfied. This is guaranteed by the following distribution function
(83)$$\begin{array}{cc} \; & f_{0k}=\frac{g}{\exp\left(\beta \cdot k-\alpha_{0}\right)+a}\;, \end{array}$$
which reduces to the Maxwell–Boltzmann, Bose–Einstein, and Fermi–Dirac equilibrium distributions for a=0,−1,1, respectively. In the above equation, *g* is the degeneracy factor of a state with momentum $k^{\mu }$, $\alpha_{0}=\mu /T$ is the (local) thermal potential, and $\beta^{\mu }=\beta_{0}u^{\mu }\equiv u^{\mu }/T$ is the inverse-temperature 4-vector.

Analogously to Equation (Equation 10), the particle-number density, energy density, and isotropic pressure are then separated into equilibrium and non-equilibrium parts,
(84)$$n\equiv n_{0}\left(T,\mu \right)+\delta n\;,\;\varepsilon \equiv \varepsilon_{0}\left(T,\mu \right)+\delta \varepsilon \;,\;p\equiv p_{0}\left(T,\mu \right)+\Pi \;,$$
with
(85a)$$\begin{array}{cc} n_{0}\left(T,\mu \right) & \equiv \int dK\;E_{k}\;f_{0k}\;,\;\varepsilon_{0}\left(T,\mu \right)\equiv \int dK\;E_{k}^{2}\;f_{0k}\;,\;p_{0}\left(T,\mu \right)\equiv -\frac{1}{3}\int dK\;{∆}_{\mu \nu }k^{\mu }k^{\nu }\;f_{0k}\;, \end{array}$$
(85b)$$\begin{array}{cc} \delta n & \equiv \int dK\;E_{k}\;\delta f_{k}\;,\;\delta \varepsilon \equiv \int dK\;E_{k}^{2}\;\delta f_{k}\;,\;\Pi \equiv -\frac{1}{3}\int dK\;{∆}_{\mu \nu }k^{\mu }k^{\nu }\;\delta f_{k}\;. \end{array}$$
With this decomposition, the fluid is described in terms of the following 19 fields: $n_{0}$, δn, $\varepsilon_{0}$, δε, Π, $u^{\mu }$, $n^{\mu }$, $h^{\mu }$, and $\pi^{\mu \nu }$, respectively. We note that these 19 degrees of freedom exceed the 14 independent components of $N^{\mu }$ and $T^{\mu \nu }$. The additional five degrees of freedom are eliminated by the matching conditions, which provide five additional constraints that define $n_{0}$, $\varepsilon_{0}$ (or μ, *T*), and $u^{\mu }$.

In the realm of kinetic theory, the matching conditions can be conveniently expressed as integral constraints for the single-particle distribution function. In the Landau frame, the conditions (Equation 16) can be expressed as
(86)$$\int dK\;E_{k}\;\delta f_{k}\equiv 0\;,\;\int dK\;E_{k}^{2}\;\delta f_{k}\equiv 0\;,\;\int dK\;E_{k}\;k^{\langle \mu \rangle }\;\delta f_{k}\equiv 0\;.$$
On the other hand, in the Eckart frame, the conditions (Equation 19) lead to
(87)$$\int dK\;E_{k}\;\delta f_{k}\equiv 0\;,\;\int dK\;E_{k}^{2}\;\delta f_{k}\equiv 0\;,\;\int dK\;k^{\langle \mu \rangle }\;\delta f_{k}\equiv 0\;.$$
The recent development of BDNK theory has created the demand for more general definitions of the reference equilibrium state [43,52,74]. And indeed, in *kinetic theory*, the constraints (Equation 86) and (Equation 87) can be generalized by considering arbitrary moments of the single-particle distribution function. The matching conditions can be written in the following more general form:(88)$$\int dK\;g_{k}\;\delta f_{k}=0\;,\;\int dK\;h_{k}\;\delta f_{k}=0\;,\;\int dK\;q_{k}\;k^{\langle \mu \rangle }\;\delta f_{k}=0\;,$$
where $g_{k}$ and $h_{k}$ are linearly independent functions of $E_{k}$ and $q_{k}$ is an arbitrary function of $E_{k}$. In [88,116,128,129], the following forms of these functions have been employed:(89)$$g_{k}=E_{k}^{q}\;,\;h_{k}=E_{k}^{s}\;,\;q_{k}=E_{k}^{z}\;,$$
which yield Landau matching conditions for (q,s,z)=(1,2,1) and Eckart matching conditions for (q,s,z)=(1,2,0). So far, alternative matching conditions of the form (Equation 88) do not have a clear physical interpretation and serve as a mathematical tool to provide closure to the equations of motion.

### 5.2. Chapman–Enskog Expansion and Navier–Stokes theory

The Navier–Stokes equations were proposed in the 19th century and are the most widespread theory of non-relativistic fluid mechanics [130,131]. A systematic derivation of this theory and its corresponding transport coefficients was developed independently in the early 20th century by S. Chapman [132,133] and D. Enskog [134]. Their formulation was built on the fundamental contribution by D. Hilbert [135], who proposed the first systematic derivation of a dissipative fluid-dynamical theory from a microscopic theory [136,137], in the non-relativistic regime. Hilbert’s idea was to convert the problem of solving the Boltzmann equation into a perturbative problem in a small dimensionless parameter, which is, in essence, the Knudsen number introduced in Section 2.4. The Hilbert expansion does not lead to the Navier–Stokes equations but to a subset of normal solutions of the Boltzmann equation, which are configurations fully determined by the fluid-dynamical state [137]. In this section, we discuss the procedure developed by Chapman and Enskog in the relativistic regime and show how it is applied to close the fluid-dynamical equations.

The starting point of Chapman–Enskog theory is the introduction of a book-keeping parameter, ϵ, on the left-hand side of the Boltzmann equation,
(90)$$\epsilon \left(E_{k}Df_{k}+k\cdot \nabla f_{k}\right)=C\left[f_{k}\right]\;,$$
where we have used $\partial_{\mu }=u_{\mu }D+\nabla_{\mu }$. We note that the derivatives of $f_{k}$ on the left-hand side of this equation are of order $∼O(L^{-1})$, while the collision term on the right-hand side is of order $∼O(\lambda_{\mathrm{mfp}}^{-1})$. Thus, the book-keeping parameter ϵ is an effective way to construct a power series in the Knudsen number $\mathrm{Kn}=\lambda_{\mathrm{mfp}}/L$. One then makes the following series Ansatz for the solution,
(91)$$f_{k}=\sum_{i=0}^{\infty }\epsilon^{i}f_{k}^{\left(i\right)}\;.$$
Then, solutions are found iteratively order by order in ϵ. The series (Equation 91) is in general an asymptotic series [137,138] and successive truncations should hence recover asymptotic solutions of the Boltzmann equation as ϵ→1. To zeroth order in this expansion, one obtains the following nonlinear equation for $f_{k}^{\left(0\right)}$,
(92)$$0=C[f_{k}^{\left(0\right)}].$$
As argued in Section 5.1, this equation is solved by the local-equilibrium distribution function (Equation 83), $f_{k}^{\left(0\right)}\equiv f_{0k}$.

So far, this procedure is identical to the one proposed by Hilbert. The new ingredient introduced by Chapman and Enskog was to consider that also the comoving (or time-like) derivative of $f_{k}$[note 12] must be expanded in ϵ [115],
(93)$$Df_{k}=\sum_{i=0}^{\infty }\epsilon^{i}{\left[Df_{k}\right]}^{\left(i\right)}\;.$$
This imposes a resummation of the original series proposed by Hilbert. At this point, it is important to point out that ${\left[Df_{k}\right]}^{\left(i\right)}$, the resummed contribution of $Df_{k}$ at *i*-th order should *not be confused* with the comoving derivative of the *i*-th correction of $f_{k}$, which we shall denote by $Df_{k}^{\left(i\right)}$. These quantities are, nevertheless, related. Indeed, $Df_{k}^{\left(i\right)}$ can have contributions of all orders in ϵ,
(94)$$Df_{k}^{\left(i\right)}=\sum_{j=1}^{\infty }\epsilon^{j}{\left[Df_{k}^{\left(i\right)}\right]}^{\left(j\right)},\;i=0,1,2,\ldots \;.$$
Then, we shall identify the resummed contribution of the comoving derivative at *n*-th order, ${\left[Df_{k}\right]}^{\left(n\right)}$, with the sum of all contributions of each $Df_{k}^{\left(j\right)}$ at (n+1)-th order [48]
(95)$${\left[Df_{k}\right]}^{\left(n\right)}=\sum_{j=0}^{n}{\left[Df_{k}^{\left(j\right)}\right]}^{\left(n+1\right)},\;i=0,1,2,\ldots \;.$$
To illustrate this resummation procedure, we shall proceed to the first-order Chapman– Enskog equation,
(96)$$E_{k}{\left[Df_{k}\right]}^{\left(0\right)}+k\cdot \nabla f_{0k}=f_{0k}\hat{L}\phi_{k}\;,$$
where $\phi_{k}\equiv f_{k}^{\left(1\right)}/\left(f_{0k}{\tilde{f}}_{0k}\right)$ defines the first correction to the local-equilibrium distribution and $\hat{L}$ is the linearized collision operator,
(97)$$f_{0k}\hat{L}\phi_{k}\equiv \int dQ\;dQ^{\prime }\;dK^{\prime }\;W_{kk^{\prime }↔qq^{\prime }}f_{0k}f_{0k^{\prime }}{\tilde{f}}_{0q}{\tilde{f}}_{0q^{\prime }}\left(\phi_{q}+\phi_{q^{\prime }}-\phi_{k}-\phi_{k^{\prime }}\right)\;.$$
One would naively identify ${\left[Df_{k}\right]}^{\left(0\right)}$ in Equation (Equation 96) as $Df_{k}^{\left(0\right)}=Df_{0k}$ (as was carried out by Hilbert, see [88]). However, Chapman and Enskog argued that the conservation laws introduce all-order contributions in ϵ to $Df_{0k}$. This can be seen using that
(98)$$Df_{0k}=\left(D\alpha_{0}-E_{k}D\beta_{0}-\beta_{0}k^{\langle \mu \rangle }Du_{\mu }\right)f_{0k}{\tilde{f}}_{0k}\;.$$
Then, making use of the conservation laws ([Disp-formula FD4a-entropy-26-00189]) to substitute $D\alpha_{0}$, $D\beta_{0}$, and $Du_{\mu }$ (cf. Equations ([Disp-formula FD256c-entropy-26-00189]), (Equation 257), and ([Disp-formula FD258a-entropy-26-00189])) we re-express this term as
(99)$$\begin{array}{cc} Df_{k}^{\left(0\right)}=Df_{0k}\; & =\left[\left(-\beta_{0}{\left.\frac{\partial p_{0}}{\partial n_{0}}\right|}_{\varepsilon_{0}}-\beta_{0}E_{k}{\left.\frac{\partial p_{0}}{\partial \varepsilon_{0}}\right|}_{n_{0}}\right)\theta -\frac{\beta_{0}}{\varepsilon_{0}+p_{0}}k\cdot F\right]f_{0k}{\tilde{f}}_{0k}\\ \; & \;\;+O(\delta n,\delta \varepsilon ,\Pi ,n^{\mu },h^{\mu },\pi^{\mu \nu })\;, \end{array}$$
Note that the terms $∼O(\delta n,\delta \varepsilon ,\Pi ,n^{\mu },h^{\mu },\pi^{\mu \nu })$ are at least of $O(\epsilon^{2})$, since they contain the product of dissipative quantities and derivatives of primary fluid-dynamical variables, or derivatives of dissipative quantities. Hence, we identify ${\left[Df_{k}\right]}^{\left(0\right)}$, the resummed zeroth-order contribution of $Df_{k}$, as the first-order contribution in ϵ of $Df_{0k}$,
(100)$${\left[Df_{k}\right]}^{\left(0\right)}={\left[Df_{k}^{\left(0\right)}\right]}^{\left(1\right)}=\left[\left(-\beta_{0}{\left.\frac{\partial p_{0}}{\partial n_{0}}\right|}_{\varepsilon_{0}}-\beta_{0}E_{k}{\left.\frac{\partial p_{0}}{\partial \varepsilon_{0}}\right|}_{n_{0}}\right)\theta -\frac{\beta_{0}}{\varepsilon_{0}+p_{0}}k\cdot F\right]f_{0k}{\tilde{f}}_{0k}\;.$$
Thus, the above equation is a constraint that must be enforced when determining the first-order solution of the Chapman–Enskog expansion. In practice, it guarantees that any comoving (time-like) derivative of a fluid-dynamical field is always replaced by space-like gradients of these fields. It is this feature that renders the Chapman–Enskog series an expansion solely in terms of *space-like* gradients. This feature is also the origin of the acausality and instability problems of relativistic Navier–Stokes theory.

Thus, Equation (Equation 96) can be cast in the following simple form
(101)$$f_{0k}{\tilde{f}}_{0k}\left(A_{k}\theta +B_{k}k\cdot I-\beta_{0}k^{\langle \mu }k^{\nu \rangle }\sigma_{\mu \nu }\right)=f_{0k}\hat{L}\phi_{k}\;,$$
where
(102a)$$\begin{array}{cc} \; & A_{k}=-\beta_{0}E_{k}{\left.\frac{\partial p_{0}}{\partial n_{0}}\right|}_{\varepsilon_{0}}-\beta_{0}E_{k}^{2}{\left.\frac{\partial p_{0}}{\partial \varepsilon_{0}}\right|}_{n_{0}}-\frac{\beta_{0}}{3}{∆}^{\lambda \sigma }k_{\lambda }k_{\sigma }\;, \end{array}$$
(102b)$$\begin{array}{cc} \; & B_{k}=1-\frac{n_{0}E_{k}}{\varepsilon_{0}+p_{0}}\;. \end{array}$$
The linear operator $\hat{L}$ is self-adjoint
(103)$$\int dK\;f_{0k}A_{k}\hat{L}B_{k}=\int dK\;f_{0k}B_{k}\hat{L}A_{k}\;,$$
and has five degenerate eigenfunctions with vanishing eigenvalue, corresponding to the five microscopic quantities that are conserved in binary elastic collisions: $\hat{L}1=0,\;\;\;\hat{L}k^{\lambda }=0$.

Equation (Equation 101) is an inhomogeneous linear integral equation for $\phi_{k}$ and, since $\hat{L}$ is a linear operator with a non-trivial kernel, the inversion procedure can only be performed in the linear subspace orthogonal to the kernel. The self-consistency aspect of this inversion may be demonstrated by multiplying Equation (Equation 101) by 1 or $k^{\lambda }$ and verifying whether these compatibility conditions are indeed satisfied. Using the above-mentioned properties of the linear collision operator, one finds the conditions
(104a)$$\begin{array}{cc} \; & \int dK\;\left(A_{k}\theta +B_{k}k\cdot I-\beta_{0}k^{\langle \mu }k^{\nu \rangle }\sigma_{\mu \nu }\right)f_{0k}{\tilde{f}}_{0k}=0\;, \end{array}$$
(104b)$$\begin{array}{cc} \; & \int dK\;k^{\lambda }\left(A_{k}\theta +B_{k}k\cdot I-\beta_{0}k^{\langle \mu }k^{\nu \rangle }\sigma_{\mu \nu }\right)f_{0k}{\tilde{f}}_{0k}=0\;, \end{array}$$
which are indeed automatically satisfied by $A_{k}$ and $B_{k}$ [48].

The general solution of Equation (Equation 101) is written as
(105)$$\phi_{k}=\phi_{k}^{\mathrm{hom}}+\phi_{k}^{\mathrm{part}}\;,$$
where $\phi_{k}^{\mathrm{hom}}$ is the solution of the homogeneous equation, $\hat{L}\phi_{k}=0$, and $\phi_{k}^{\mathrm{part}}$ is a particular solution of the inhomogeneous equation. Given the zero modes of the collision operator, the homogeneous solution is
(106)$$\phi_{k}^{\mathrm{hom}}=a+b\cdot k\;,$$
where *a* and $b^{\mu }$ are arbitrary real-valued constants, which will be determined by imposing the matching conditions (Equation 88). The arbitrariness in the choice of these constants is reflected by the fact that the choice of matching conditions is also arbitrary in kinetic theory.

Since $\hat{L}$ is a linear operator, the particular solution $\phi_{k}^{\mathrm{part}}$ must have the general form
(107)$$\phi_{k}^{\mathrm{part}}=S_{k}\theta +V_{k}k\cdot I+T_{k}k^{\langle \mu }k^{\nu \rangle }\sigma_{\mu \nu }\;,$$
where, at this point, $S_{k}$, $V_{k}$, and $T_{k}$ are unknown functions of $E_{k}$. The next step is to insert the particular solution (Equation 107) into Equation (Equation 101),
(108)$$(A_{k}\theta +B_{k}k\cdot I-\beta_{0}k^{\langle \mu }k^{\nu \rangle }\sigma_{\mu \nu })f_{0k}{\tilde{f}}_{0k}=\theta f_{0k}\hat{L}[S_{k}]+I_{\mu }f_{0k}\hat{L}[V_{k}k^{\langle \mu \rangle }]+\sigma_{\mu \nu }f_{0k}\hat{L}[T_{k}k^{\langle \mu }k^{\nu \rangle }]\;.$$
This results in coupled integral equations for $S_{k}$, $V_{k}$, and $T_{k}$. We now expand these functions using a complete basis of functions of $E_{k}$, $P_{nk}^{\left(\ell \right)}=P_{n}^{\left(\ell \right)}\left(\beta_{0}E_{k}\right)$, n,ℓ=0,1,…,
(109)$$S_{k}=\sum_{n=0}^{\infty }s_{n}P_{nk}^{\left(0\right)}\;,\;V_{k}=\sum_{n=0}^{\infty }v_{n}P_{nk}^{\left(1\right)}\;,\;T_{k}=\sum_{n=0}^{\infty }t_{n}P_{nk}^{\left(2\right)}\;.$$
Equation (Equation 108) can be decoupled by multiplying it with the basis elements $P_{nk}^{\left(\ell \right)}k^{\langle \mu_{1}}\cdots k^{\mu_{\ell }\rangle }$ (the functions $P_{nk}^{\left(\ell \right)}$ need not necessarily be orthogonal, and not even be polynomials, as in [121,139]), then integrating over momentum and using property (Equation 280). This leads to the following systems of equations:
(110a)$$\begin{array}{cc} \; & \sum_{n}S_{rn}s_{n}=\int dK\;P_{rk}^{\left(0\right)}A_{k}f_{0k}{\tilde{f}}_{0k}\equiv A_{r}\;, \end{array}$$
(110b)$$\begin{array}{cc} \; & \sum_{n}V_{rn}v_{n}=\int dK\;{∆}^{\mu \nu }k_{\mu }k_{\nu }P_{rk}^{\left(1\right)}B_{k}f_{0k}{\tilde{f}}_{0k}\equiv B_{r}\;, \end{array}$$
(110c)$$\begin{array}{cc} \; & \sum_{n}T_{rn}t_{n}=-\beta_{0}\int dK\;{\left({∆}^{\mu \nu }k_{\mu }k_{\nu }\right)}^{2}P_{rk}^{\left(2\right)}f_{0k}{\tilde{f}}_{0k}\equiv C_{r}\;, \end{array}$$
where we defined the following integrals of the linearized collision term
(111a)$$\begin{array}{cc} \; & S_{rn}\equiv \int dK\;P_{rk}^{\left(0\right)}\hat{L}\left[P_{nk}^{\left(0\right)}\right]f_{0k}\;, \end{array}$$
(111b)$$\begin{array}{cc} \; & V_{rn}\equiv \int dK\;P_{rk}^{\left(1\right)}k^{\langle \mu \rangle }\hat{L}\left[P_{nk}^{\left(1\right)}k_{\langle \mu \rangle }\right]f_{0k}\;, \end{array}$$
(111c)$$\begin{array}{cc} \; & T_{rn}\equiv \int dK\;P_{rk}^{\left(2\right)}k^{\langle \mu }k^{\nu \rangle }\hat{L}\left[P_{nk}^{\left(2\right)}k_{\langle \mu }k_{\nu \rangle }\right]f_{0k}\;. \end{array}$$
Equations ([Disp-formula FD110a-entropy-26-00189]) can be inverted as
(112)$$s_{n}=\sum_{m}{\left[S^{-1}\right]}_{nm}A_{m}\;,\;v_{n}=\sum_{m}{\left[V^{-1}\right]}_{nm}B_{m}\;,\;t_{n}=\sum_{m}{\left[T^{-1}\right]}_{nm}C_{m}\;.$$
We note that, if the basis contains parts of the homogeneous solution, the corresponding terms must be removed from the inversion procedure.

As discussed at the beginning of the present section, a common feature shared by all methods for computing transport coefficients is an *ad hoc* definition of a local-equilibrium state. Now, these conditions are used to obtain the coefficients *a* and $b^{\mu }$ of the homogeneous solution. Using the matching conditions (Equation 88) with Equation (Equation 89), we obtain [88]
(113a)$$\begin{array}{cc} a & =\frac{J_{q+1,0}{\langle E_{k}^{s}S_{k}\rangle }_{\tilde{\mathrm{eq}}}-{\langle E_{k}^{q}S_{k}\rangle }_{\tilde{\mathrm{eq}}}J_{s+1,0}}{G_{s+1,q}}\;\theta \;, \end{array}$$
(113b)$$\begin{array}{cc} b\cdot u & =\frac{{\langle E_{k}^{q}S_{k}\rangle }_{\tilde{\mathrm{eq}}}J_{s0}-J_{q0}{\langle E_{k}^{s}S_{k}\rangle }_{\tilde{\mathrm{eq}}}}{G_{s+1,q}}\;\theta \;, \end{array}$$
(113c)$$\begin{array}{cc} b^{\langle \lambda \rangle }\; & =\frac{1}{3}\frac{{\langle {∆}^{\mu \nu }k_{\mu }k_{\nu }E_{k}^{z}V_{k}\rangle }_{\tilde{\mathrm{eq}}}}{J_{z+2,1}}\;I^{\lambda }\;, \end{array}$$
where
(114a)$$\begin{array}{cc} {\langle \cdots \rangle }_{\tilde{\mathrm{eq}}} & \equiv \int dK\;\left(\cdots \right)f_{0k}{\tilde{f}}_{0k}\;, \end{array}$$
(114b)$$\begin{array}{cc} J_{nq} & \equiv \frac{1}{(2q+1)!!}{\langle {\left(-{∆}^{\mu \nu }k_{\mu }k_{\nu }\right)}^{\ell }E_{k}^{n-2q}\rangle }_{\tilde{\mathrm{eq}}}\;, \end{array}$$
(114c)$$\begin{array}{cc} G_{nm} & \equiv J_{n0}J_{m0}-J_{n-1,0}J_{m+1,0}\;. \end{array}$$

Finally, combining the results displayed above, we obtain the solution for the first-order Chapman–Enskog deviation function
(115)$$\phi_{k}={\tilde{S}}_{k}\theta +{\tilde{V}}_{k}k\cdot I+T_{k}k^{\langle \mu }k^{\nu \rangle }\sigma_{\mu \nu }\;,$$
where we defined the following scalar functions of $E_{k}$:
(116a)$$\begin{array}{cc} \tilde{S_{k}} & =\sum_{n\geq 0}\sum_{m\geq 0}{\left[S^{-1}\right]}_{nm}A_{m}\left(P_{nk}^{\left(0\right)}+\frac{J_{q+1,0}J_{sn}^{\left(0\right)}-J_{qn}^{\left(0\right)}J_{s+1,0}}{G_{s+1,q}}+\frac{J_{qn}^{\left(0\right)}J_{s0}-J_{q0}J_{sn}^{\left(0\right)}}{G_{s+1,q}}E_{k}\right)\;, \end{array}$$
(116b)$$\begin{array}{cc} {\tilde{V}}_{k}\; & =\sum_{n\geq 0}\sum_{m\geq 0}{\left[V^{-1}\right]}_{nm}B_{m}\left(P_{nk}^{\left(1\right)}-\frac{J_{zn}^{\left(1\right)}}{J_{z+2,1}}\right)\;, \end{array}$$
(116c)$$\begin{array}{cc} T_{k} & =\sum_{n\geq 0}\sum_{m\geq 0}{\left[T^{-1}\right]}_{nm}C_{m}P_{nk}^{\left(2\right)}\;. \end{array}$$
Here, we also made use of the thermodynamic integrals
(117)$$J_{mn}^{\left(\ell \right)}=\frac{1}{(2\ell +1)!!}{\langle {\left(-{∆}^{\mu \nu }k_{\mu }k_{\nu }\right)}^{\ell }E_{k}^{m}P_{nk}^{\left(\ell \right)}\rangle }_{\tilde{\mathrm{eq}}}\;.$$
The solution can then be employed to obtain the constitutive relations satisfied by the non-equilibrium corrections under general matching conditions. Indeed, the definitions ([Disp-formula FD81a-entropy-26-00189]) yield
(118)$$\Pi =-\zeta \theta \;,\;\delta n=\xi \theta \;,\;\delta \varepsilon =\chi \theta \;,\;n^{\mu }=\varkappa I^{\mu }\;,\;h^{\mu }=-\lambda I^{\mu }\;,\;\pi^{\mu \nu }=2\eta \sigma^{\mu \nu }\;,$$
with the transport coefficients given by
(119a)$$\begin{array}{cc} \; & \zeta =\sum_{n\geq 2}\sum_{m\geq 2}{\left[S^{-1}\right]}_{nm}A_{m}H_{n}^{\left(\zeta \right)}\;,\;\;\xi =\sum_{n\geq 2}\sum_{m\geq 2}{\left[S^{-1}\right]}_{nm}A_{m}H_{n}^{\left(\xi \right)}\;, \end{array}$$
(119b)$$\begin{array}{cc} \chi & =\sum_{n\geq 2}\sum_{m\geq 2}{\left[S^{-1}\right]}_{nm}A_{m}H_{n}^{\left(\chi \right)}\;,\;\;\varkappa =\sum_{n\geq 1}\sum_{m\geq 1}{\left[V^{-1}\right]}_{nm}B_{m}K_{n}^{\left(\varkappa \right)}\;, \end{array}$$
(119c)$$\begin{array}{cc} \lambda & =\sum_{n\geq 1}\sum_{m\geq 1}{\left[V^{-1}\right]}_{nm}B_{m}K_{n}^{\left(\lambda \right)}\;,\;\;\eta =\sum_{n\geq 0}\sum_{m\geq 0}{\left[T^{-1}\right]}_{nm}C_{m}J_{0n}^{\left(2\right)}\;, \end{array}$$
where
(120a)$$\begin{array}{cc} \; & H_{n}^{\left(\zeta \right)}=\frac{1}{3}\left(m^{2}J_{0n}^{\left(0\right)}-J_{2n}^{\left(0\right)}\right)+\frac{1}{3}\frac{m^{2}G_{q+1,0}-G_{q+1,2}}{G_{s+1,q}}J_{sn}^{\left(0\right)}-\frac{1}{3}\frac{m^{2}G_{s+1,0}-G_{s+1,2}}{G_{s+1,q}}J_{qn}^{\left(0\right)}\;, \end{array}$$
(120b)$$\begin{array}{cc} \; & H_{n}^{\left(\xi \right)}=J_{1n}^{\left(0\right)}+\frac{G_{q+1,1}}{G_{s+1,q}}J_{sn}^{\left(0\right)}-\frac{G_{s+1,1}}{G_{s+1,q}}J_{qn}^{\left(0\right)}\;, \end{array}$$
(120c)$$\begin{array}{cc} \; & H_{n}^{\left(\chi \right)}=J_{2n}^{\left(0\right)}+\frac{G_{q+1,2}}{G_{s+1,q}}J_{sn}^{\left(0\right)}-\frac{G_{s+1,2}}{G_{s+1,q}}J_{qn}^{\left(0\right)}, \end{array}$$
(120d)$$\begin{array}{cc} \; & K_{n}^{\left(\varkappa \right)}=-J_{0n}^{\left(1\right)}+\frac{J_{21}}{J_{z+2,1}}J_{zn}^{\left(1\right)}\;, \end{array}$$
(120e)$$\begin{array}{cc} \; & K_{n}^{\left(\lambda \right)}=J_{1n}^{\left(1\right)}-\frac{J_{31}}{J_{z+2,1}}J_{zn}^{\left(1\right)}\;. \end{array}$$
One observes that the transport coefficients are, in general, quite complicated functions of temperature and chemical potential. Some simplification can be made using phenomenological approximations for the collision term, such as the relaxation-time approximation [140,141]. It is also relevant to point out that the choice of matching conditions affects significantly some of the transport coefficients, which explicitly depend on the parameters *q*, *s*, and *z* necessary to define the matching conditions, see Equation (Equation 89). Indeed, if we use the Landau prescription, (q,s,z)=(1,2,1), we have ξ=χ=λ=0. If we use the Eckart prescription instead, then ξ=χ=ϰ=0. Alternatively, in a matching condition defined such that q=0 and s=2, we would have ζ=0. It should also be noted that, in the massless limit, because ${∆}^{\mu \nu }k_{\mu }k_{\nu }=-E_{k}^{2}$, we have that $A_{k}=0$, from which follows $S_{k}=0$, implying that all transport coefficients related to scalar non-equilibrium fields must vanish, i.e., ζ=ξ=χ=0. Moreover, we note that certain combinations of the transport coefficients can be shown to be independent of the matching conditions imposed. We shall refer to them as $\zeta_{s}$ and $\varkappa_{s}$ and they are given by [88],
(121)$$\zeta_{s}=\zeta +{\left.\frac{\partial p_{0}}{\partial n_{0}}\right|}_{\varepsilon_{0}}\xi +{\left.\frac{\partial p_{0}}{\partial \varepsilon_{0}}\right|}_{n_{0}}\chi \;,\;\varkappa_{s}=\varkappa +\frac{n_{0}}{\varepsilon_{0}+p_{0}}\lambda \;.$$

The equations of motion of relativistic Navier–Stokes theory are obtained by inserting the constitutive relations (Equation 118) into the exact conservation laws ([Disp-formula FD3a-entropy-26-00189]). Nevertheless, it is possible to demonstrate that the resulting equations of motion are acausal for any choice of matching condition [52] and, therefore, cannot be considered as a viable formulation of relativistic dissipative fluid dynamics. We finally note that the procedure outlined in this section can, in principle, be carried out to higher orders, even though this is, in practice, extremely challenging.

### 5.3. Modified Chapman–Enskog Expansion and BDNK Theory

In this subsection, we discuss the microscopic derivation of BDNK theory [43,52,73,74] from kinetic theory [88,116]. The crucial difference with respect to the traditional Navier–Stokes formulation is that the constitutive relations also include comoving, or time-like, derivatives of the fluid-dynamical fields. We shall now discuss how the Chapman–Enskog formulation can be altered to include this feature.

The reason why the Chapman–Enskog expansion leads to Navier–Stokes and not to BDNK theory is that time-like derivatives are replaced by space-like ones when obtaining the perturbative solution for the time-like derivatives of the distribution function (cf., for instance, Equation (Equation 100)). As already discussed, this replacement is essential to guarantee the validity of the compatibility conditions ([Disp-formula FD104a-entropy-26-00189]) in Chapman–Enskog theory. Hilbert’s expansion also does not lead to BDNK theory [88] because the time-like derivatives are *exactly* substituted by space-like ones due to the fact that the equations of motion include the Euler equations explicitly. In both cases, the zero modes of the linearized collision operator lead to conditions that enforce the replacement of time-like derivatives of the fluid-dynamical variables by space-like ones.

In order to address this, a new perturbative scheme was proposed in [88]: first the Boltzmann equation is integrated over a complete and irreducible basis $P_{nk}^{\left(\ell \right)}k_{\langle \mu_{1}}\cdots k_{\mu_{\ell }\rangle }$, thus forming an infinite tower of equations for its moments. Afterwards, the perturbative book-keeping parameter ϵ is inserted on the left-hand side of all moment equations,
(122)$$\epsilon \int dK\;P_{nk}^{\left(\ell \right)}k_{\langle \mu_{1}}\cdots k_{\mu_{\ell }\rangle }\;k\cdot \partial f_{k}=\int dK\;P_{nk}^{\left(\ell \right)}k_{\langle \mu_{1}}\cdots k_{\mu_{\ell }\rangle }\;C\left[f_{k}\right]\;,\;\ell =0,1,\ldots \;,\;\;n=0,1,\ldots \;.$$
Thus, if the basis elements include $1,k^{\nu }$ we obtain the usual conservation laws and the perturbative parameter simply disappears
(123)$$\int dK\;k\cdot \partial f_{k}=0\;,\;\int dK\;k^{\mu }k^{\nu }\partial_{\mu }f_{k}=0\;.$$
These will be treated non-perturbatively, as was the case in Chapman–Enskog theory. In this procedure, integrals of the linearized collision operator over $1,k^{\nu }$ simply do not appear and, hence, are not required to vanish (constraints ([Disp-formula FD104a-entropy-26-00189]) no longer exist). Thus, from now on, only the basis elements in the linear subspace orthogonal to the zero modes shall be considered in our analysis or, equivalently, we assume that the basis $P_{nk}^{\left(\ell \right)}k_{\langle \mu_{1}}\cdots k_{\mu_{\ell }\rangle }$ does not include 1 and $k^{\nu }$.

Applying a perturbative procedure, one assumes, as usual, an asymptotic-series solution for $f_{k}$, as in Equation (Equation 91), and Equation (Equation 122) are solved order-by-order in the perturbative parameter ϵ. Indeed, at $O(\epsilon^{0})$, we have
(124)$$0=\int dK\;P_{nk}^{\left(\ell \right)}k_{\langle \mu_{1}}\cdots k_{\mu_{\ell }\rangle }\;C\left[f_{k}^{\left(0\right)}\right]\;.$$
The fact that integrals over arbitrary basis elements all vanish implies that $C[f_{k}^{\left(0\right)}]=0$ and, thus, $f_{k}^{\left(0\right)}=f_{0k}$. This happens, because, even though the basis elements corresponding to the microscopically conserved quantities are excluded, we have the exact conditions $\int dK\;C\left[f_{0k}\right]=0\;,\;\int dK\;E_{k}\;C\left[f_{0k}\right]=0$, and $\int dK\;k^{\langle \mu \rangle }\;C\left[f_{0k}\right]=0$, which provide us with the complementary information at this order.

Next, collecting all terms of first order in ϵ, we obtain
(125)$$\begin{array}{cc} \; & \int dK\;P_{nk}^{\left(\ell \right)}k_{\langle \mu_{1}}\cdots k_{\mu_{\ell }\rangle }\;k\cdot \partial f_{0k}=\int dK\;P_{nk}^{\left(\ell \right)}k_{\langle \mu_{1}}\cdots k_{\mu_{\ell }\rangle }\;f_{0k}\hat{L}\phi_{k}^{\left(1\right)}\;, \end{array}$$
where we emphasize that the zero modes of the linearized collision operator do not enter in this set of equations, i.e., the basis elements 1 and $k^{\nu }$ are not present in this equation. This implies that the compatibility conditions that require the exchange of time-like derivatives of $f_{k}$ by space-like ones in Chapman–Enskog theory, see Equations ([Disp-formula FD104a-entropy-26-00189]), do not appear in this case. This is a consequence of performing the perturbative procedure on moments of the Boltzmann equation and not on the Boltzmann equation itself. The term inside each integral on the left-hand side can be irreducibly written as
(126)$$\begin{array}{cc} k\cdot \partial f_{0k} & =\left[E_{k}D\alpha_{0}-E_{k}^{2}D\beta_{0}-\frac{\beta_{0}}{3}{∆}^{\lambda \sigma }k_{\lambda }k_{\sigma }\theta +k\cdot I\right.\\ \; & \;\;\left.-E_{k}k^{\langle \mu \rangle }\left(\beta_{0}Du_{\mu }+\nabla_{\mu }\beta_{0}\right)-\beta_{0}k^{\langle \mu }k^{\nu \rangle }\sigma_{\mu \nu }\right]f_{0k}{\tilde{f}}_{0k}\;. \end{array}$$
Since $\hat{L}$ is a linear operator, Equation (Equation 125) implies that the solution for $\phi_{k}$ can be expressed as the sum of a homogeneous and a particular solution, as in Equation (Equation 105), where the homogeneous part has the form given in Equation (Equation 106). Since we do not have any self-consistency or compatibility conditions that impose the replacement of time-like derivatives of fluid-dynamical variables by space-like ones, the particular solution has the general form
(127)$$\phi_{k}^{\mathrm{part}}=S_{k}^{\left(\alpha \right)}D\alpha_{0}+S_{k}^{\left(\beta \right)}D\beta_{0}+S_{k}^{\left(\theta \right)}\theta +V_{k}^{\left(\alpha \right)}k\cdot I+V_{k}^{\left(\beta \right)}k^{\langle \mu \rangle }\left(\nabla_{\mu }\beta_{0}+\beta_{0}Du_{\mu }\right)+T_{k}\;k^{\langle \mu }k^{\nu \rangle }\sigma_{\mu \nu }\;.$$
The following steps are essentially the same as those applied in the Chapman–Enskog and Hilbert procedures and involve the inversion of the linearized collision operator (in the subspace excluding its zero modes). We assume that the functions $S_{k}^{\left(\alpha ,\beta ,\theta \right)}$, $V_{k}^{\left(\alpha ,\beta \right)}$, and $T_{k}$ can be expanded in terms of the complete basis $P_{n}^{\left(\ell \right)}$,
(128)$$S_{k}^{\left(\alpha ,\beta ,\theta \right)}=\sum_{n\geq 0}s_{n}^{\left(\alpha ,\beta ,\theta \right)}P_{n}^{\left(0\right)}\;,\;V_{k}^{\left(\alpha ,\beta \right)}=\sum_{n\geq 0}v_{n}^{\left(\alpha ,\beta \right)}P_{kn}^{\left(1\right)}\;,\;T_{k}=\sum_{n\geq 0}t_{n}P_{kn}^{\left(2\right)}\;,$$
which leads to the following system of linear equations:(129)$$\sum_{n}S_{rn}s_{n}^{\left(\alpha ,\beta ,\theta \right)}=A_{r}^{\left(\alpha ,\beta ,\theta \right)}\;,\;\sum_{n}V_{rn}v_{n}^{\left(\alpha ,\beta \right)}=B_{r}^{\left(\alpha ,\beta \right)}\;,\;\sum_{n}T_{rn}t_{n}=C_{r}\;,$$
to be solved for the coefficients $s_{n}^{\left(\alpha ,\beta ,\theta \right)}$, $v_{n}^{\left(\alpha ,\beta \right)}$, and $t_{n}$. The matrices *S*, *V*, and *T* were already defined in Equations ([Disp-formula FD111a-entropy-26-00189]). We further defined
(130a)$$\begin{array}{cc} A_{r}^{\left(\alpha \right)} & =J_{1r}^{\left(0\right)}\;,\;A_{r}^{\left(\beta \right)}=-J_{2r}^{\left(0\right)}\;,\;A_{r}^{\left(\theta \right)}=\frac{\beta_{0}}{3}\left(J_{2r}^{\left(0\right)}-m^{2}J_{0r}^{\left(0\right)}\right)\;, \end{array}$$
(130b)$$\begin{array}{cc} B_{r}^{\left(\alpha \right)} & =-3J_{0r}^{\left(1\right)}\;,\;B_{r}^{\left(\beta \right)}=3J_{1r}^{\left(1\right)}\;, \end{array}$$
(130c)$$\begin{array}{cc} C_{r} & =-15\beta_{0}J_{0r}^{\left(2\right)}\;, \end{array}$$
where the thermodynamic integrals $J_{rn}^{\left(\ell \right)}$ are given in Equation (Equation 117). Equation (Equation 129) can be inverted as
(131)$$s_{n}^{\left(\alpha ,\beta ,\theta \right)}=\sum_{m}{\left[S^{-1}\right]}_{nm}A_{m}^{\left(\alpha ,\beta ,\theta \right)}\;,\;v_{n}^{\left(\alpha ,\beta \right)}=\sum_{m}{\left[V^{-1}\right]}_{nm}B_{m}^{\left(\alpha ,\beta \right)}\;,\;t_{n}=\sum_{m}{\left[T^{-1}\right]}_{nm}C_{m}\;.$$

Then, we proceed to obtain the homogeneous solution $\phi_{k}^{\mathrm{hom}}$. To this end, we substitute Equations (Equation 105) and (Equation 106) into the general matching conditions (Equation 88) and (Equation 89), in complete analogy with Equations ([Disp-formula FD113a-entropy-26-00189]) in the Chapman–Enskog expansion. In the present case, this procedure yields
(132a)$$\begin{array}{cc} a & =\frac{J_{q+1,0}{\langle E_{k}^{s}S_{k}^{\left(\alpha \right)}\rangle }_{\tilde{\mathrm{eq}}}-{\langle E_{k}^{q}S_{k}^{\left(\alpha \right)}\rangle }_{\tilde{\mathrm{eq}}}J_{s+1,0}}{G_{s+1,q}}\;D\alpha_{0}+\frac{J_{q+1,0}{\langle E_{k}^{s}S_{k}^{\left(\beta \right)}\rangle }_{\tilde{\mathrm{eq}}}-{\langle E_{k}^{q}S_{k}^{\left(\beta \right)}\rangle }_{\tilde{\mathrm{eq}}}J_{s+1,0}}{G_{s+1,q}}\;D\beta_{0}\\ \; & \;\;+\frac{J_{q+1,0}{\langle E_{k}^{s}S_{k}^{\left(\theta \right)}\rangle }_{\tilde{\mathrm{eq}}}-{\langle E_{k}^{q}S_{k}^{\left(\theta \right)}\rangle }_{\tilde{\mathrm{eq}}}J_{s+1,0}}{G_{s+1,q}}\;\theta \;, \end{array}$$
(132b)$$\begin{array}{c} b\cdot u=\frac{{\langle E_{k}^{q}S_{k}^{\left(\alpha \right)}\rangle }_{\tilde{\mathrm{eq}}}J_{s0}-J_{q0}{\langle E_{k}^{s}S_{k}^{\left(\alpha \right)}\rangle }_{\tilde{\mathrm{eq}}}}{G_{s+1,q}}\;D\alpha_{0}+\frac{{\langle E_{k}^{q}S_{k}^{\left(\beta \right)}\rangle }_{0}J_{s0}-J_{q0}{\langle E_{k}^{s}S_{k}^{\left(\beta \right)}\rangle }_{\tilde{\mathrm{eq}}}}{G_{s+1,q}}\;D\beta_{0}\\ \;\;\;+\frac{{\langle E_{k}^{q}S_{k}^{\left(\theta \right)}\rangle }_{\tilde{\mathrm{eq}}}J_{s0}-J_{q0}{\langle E_{k}^{s}S_{k}^{\left(\theta \right)}\rangle }_{\tilde{\mathrm{eq}}}}{G_{s+1,q}}\;\theta \;, \end{array}$$
(132c)$$\begin{array}{c} b^{\langle \lambda \rangle }=\frac{1}{3}\frac{{\langle {∆}^{\mu \nu }k_{\mu }k_{\nu }E_{k}^{z}V_{k}^{\left(\alpha \right)}\rangle }_{\tilde{\mathrm{eq}}}}{J_{z+2,1}}I^{\lambda }+\frac{1}{3}\frac{{\langle {∆}^{\mu \nu }k_{\mu }k_{\nu }E_{k}^{z}V_{k}^{\left(\beta \right)}\rangle }_{\tilde{\mathrm{eq}}}}{J_{z+2,1}}\left(\nabla^{\lambda }\beta_{0}+\beta_{0}Du^{\lambda }\right)\;, \end{array}$$
where we used the notation ([Disp-formula FD114a-entropy-26-00189]). Finally, combining the homogeneous solution found above with the particular solution derived in Equation (Equation 131), we obtain the complete first-order solution of the modified perturbative procedure introduced in this section. The solution can be expressed as
(133)$$\phi_{k}={\tilde{S}}_{k}^{\left(\alpha \right)}D\alpha_{0}+{\tilde{S}}_{k}^{\left(\beta \right)}D\beta_{0}+{\tilde{S}}_{k}^{\left(\theta \right)}\theta +{\tilde{V}}_{k}^{\left(\alpha \right)}k\cdot I+{\tilde{V}}_{k}^{\left(\beta \right)}k^{\langle \mu \rangle }\left(\nabla_{\mu }\beta_{0}+\beta_{0}Du_{\mu }\right)+T_{k}k^{\langle \mu }k^{\nu \rangle }\sigma_{\mu \nu }\;,$$
where the momentum-dependent functions are defined as
(134a)$$\begin{array}{cc} {\tilde{S}}_{k}^{\left(\alpha ,\beta ,\theta \right)} & =\sum_{n}\sum_{m}{\left[S^{-1}\right]}_{nm}A_{m}^{\left(\alpha ,\beta ,\theta \right)}\left(P_{nk}^{\left(0\right)}+\frac{J_{q+1,0}J_{sn}^{\left(0\right)}-J_{qn}^{\left(0\right)}J_{s+1,0}}{G_{s+1,q}}+\frac{J_{qn}^{\left(0\right)}J_{s0}-J_{q0}J_{sn}^{\left(0\right)}}{G_{s+1,q}}E_{k}\right)\;, \end{array}$$
(134b)$$\begin{array}{cc} {\tilde{V}}_{k}^{\left(\alpha ,\beta \right)}\; & =\sum_{n}\sum_{m}{\left[V^{-1}\right]}_{nm}B_{m}^{\left(\alpha ,\beta \right)}\left(P_{nk}^{\left(1\right)}-\frac{J_{zn}^{\left(1\right)}}{J_{z+2,1}}\right)\;, \end{array}$$
(134c)$$\begin{array}{cc} T_{k} & =\sum_{n}\sum_{m}{\left[T^{-1}\right]}_{nm}C_{m}P_{nk}^{\left(2\right)}\;. \end{array}$$
Substituting the solution (Equation 133) into the definition of the dissipative currents ([Disp-formula FD81a-entropy-26-00189]), we obtain the following constitutive relations
(135a)$$\begin{array}{cc} \; & \Pi =\zeta^{\left(\alpha \right)}D\alpha_{0}-\zeta^{\left(\beta \right)}\frac{D\beta_{0}}{\beta_{0}}-\zeta^{\left(\theta \right)}\theta \;,\;\delta n=\xi^{\left(\alpha \right)}D\alpha_{0}-\xi^{\left(\beta \right)}\frac{D\beta_{0}}{\beta_{0}}-\xi^{\left(\theta \right)}\theta \;, \end{array}$$
(135b)$$\begin{array}{cc} \delta \varepsilon \; & =\chi^{\left(\alpha \right)}D\alpha_{0}-\chi^{\left(\beta \right)}\frac{D\beta_{0}}{\beta_{0}}-\chi^{\left(\theta \right)}\theta \;,\;n^{\mu }=\varkappa^{\left(\alpha \right)}I^{\mu }-\varkappa^{\left(\beta \right)}\left(\frac{1}{\beta_{0}}\nabla^{\mu }\beta_{0}+Du^{\mu }\right)\;, \end{array}$$
(135c)$$\begin{array}{cc} h^{\mu }\; & =\lambda^{\left(\alpha \right)}I^{\mu }-\lambda^{\left(\beta \right)}\left(\frac{1}{\beta_{0}}\nabla^{\mu }\beta_{0}+Du^{\mu }\right)\;,\;\pi^{\mu \nu }=2\eta \sigma^{\mu \nu }\;, \end{array}$$
where the microscopic expressions for the 14 transport coefficients introduced above are given by
(136a)$$\begin{array}{cc} \zeta^{\left(\alpha \right)} & =-\sum_{n,m}{\left[S^{-1}\right]}_{nm}A_{m}^{\left(\alpha \right)}H_{n}^{\left(\zeta \right)}\;,\;\;\zeta^{\left(\beta \right)}=\beta_{0}\sum_{n,m}{\left[S^{-1}\right]}_{nm}A_{m}^{\left(\beta \right)}H_{n}^{\left(\zeta \right)}\;, \end{array}$$
(136b)$$\begin{array}{cc} \zeta^{\left(\theta \right)} & =\sum_{n,m}{\left[S^{-1}\right]}_{nm}A_{m}^{\left(\theta \right)}H_{n}^{\left(\zeta \right)}\;,\;\;\xi^{\left(\alpha \right)}=\sum_{n,m}{\left[S^{-1}\right]}_{nm}A_{m}^{\left(\alpha \right)}H_{n}^{\left(\xi \right)}\;, \end{array}$$
(136c)$$\begin{array}{cc} \xi^{\left(\beta \right)} & =-\beta_{0}\sum_{n,m}{\left[S^{-1}\right]}_{nm}A_{m}^{\left(\beta \right)}H_{n}^{\left(\xi \right)}\;,\;\;\xi^{\left(\theta \right)}=-\sum_{n,m}{\left[S^{-1}\right]}_{nm}A_{m}^{\left(\theta \right)}H_{n}^{\left(\xi \right)}\;, \end{array}$$
(136d)$$\begin{array}{cc} \chi^{\left(\alpha \right)} & =\sum_{n,m}{\left[S^{-1}\right]}_{nm}A_{m}^{\left(\alpha \right)}H_{n}^{\left(\chi \right)}\;,\;\;\chi^{\left(\beta \right)}=-\beta_{0}\sum_{n,m}{\left[S^{-1}\right]}_{nm}A_{m}^{\left(\beta \right)}H_{n}^{\left(\chi \right)}\;, \end{array}$$
(136e)$$\begin{array}{cc} \chi^{\left(\theta \right)} & =-\sum_{n,m}{\left[S^{-1}\right]}_{nm}A_{m}^{\left(\theta \right)}H_{n}^{\left(\chi \right)}\;,\;\;\varkappa^{\left(\alpha \right)}=\sum_{n,m}{\left[V^{-1}\right]}_{nm}B_{m}^{\left(\alpha \right)}K_{n}^{\left(\varkappa \right)}\;, \end{array}$$
(136f)$$\begin{array}{cc} \varkappa^{\left(\beta \right)} & =-\beta_{0}\sum_{n,m}{\left[V^{-1}\right]}_{nm}B_{m}^{\left(\beta \right)}K_{n}^{\left(\varkappa \right)}\;,\;\;\lambda^{\left(\alpha \right)}=-\sum_{n,m}{\left[V^{-1}\right]}_{nm}B_{m}^{\left(\alpha \right)}K_{n}^{\left(\lambda \right)}\;, \end{array}$$
(136g)$$\begin{array}{cc} \lambda^{\left(\beta \right)} & =\beta_{0}\sum_{n,m}{\left[V^{-1}\right]}_{nm}B_{m}^{\left(\beta \right)}K_{n}^{\left(\lambda \right)}\;,\;\;\eta =\sum_{n,m}{\left[T^{-1}\right]}_{nm}C_{m}J_{0n}^{\left(2\right)}\;. \end{array}$$
The functions $H^{\left(\zeta ,\xi ,\chi \right)},K^{\left(\varkappa ,\lambda \right)}$ were already defined in Equations ([Disp-formula FD120a-entropy-26-00189]). We further notice that the shear-viscosity coefficient η has the same expression as in the Chapman–Enskog and Hilbert expansions. In the massless limit, since $\delta T_{\;\mu }^{\mu }=\delta \varepsilon -3\Pi =0$, we have that $3\zeta^{\left(\alpha \right)}=\chi^{\left(\alpha \right)}$, $3\zeta^{\left(\beta \right)}=\chi^{\left(\beta \right)}$, and $3\zeta^{\left(\theta \right)}=\chi^{\left(\theta \right)}$. Also in this limit, since ${∆}^{\lambda \sigma }k_{\lambda }k_{\sigma }=-E_{k}^{2}$, we have that $3\xi^{\left(\theta \right)}=\xi^{\left(\beta \right)}$ and $3\chi^{\left(\theta \right)}=\chi^{\left(\beta \right)}$, even though they are in general not zero. This is in contrast to what happened in the traditional Chapman–Enskog expansion where ζ, ξ, and χ vanish identically in the m→0 limit. The constitutive relations ([Disp-formula FD135a-entropy-26-00189]), combined with the conservation laws, lead to the BDNK equations [43,52,73,74,75]. It is also noted that, as in Navier–Stokes theory (see Equations ([Disp-formula FD119a-entropy-26-00189])), the majority of the coefficients strongly depend on the parameters *q*, *s*, and *z* that determine the matching conditions employed. In fact, for Landau matching conditions, (q,s,z)=(1,2,1), we have $\xi^{\left(\alpha ,\beta ,\theta \right)}=\chi^{\left(\alpha ,\beta ,\theta \right)}=\lambda^{\left(\alpha ,\beta \right)}=0$, while for Eckart matching conditions, (q,s,z)=(1,2,0), one finds $\xi^{\left(\alpha ,\beta ,\theta \right)}=\chi^{\left(\alpha ,\beta ,\theta \right)}=\lambda^{\left(\alpha ,\beta \right)}=0$. Furthermore, for matching conditions that respect (q,s)=(0,2) we have that $\zeta^{\left(\alpha ,\beta ,\theta \right)}=0$. Expressions for the various BDNK transport coefficients have been computed in the relaxation-time approximation [140,141] for the collision term in [88] and for a system of ultra-relativistic scalar particles with quartic self-interactions in [116]. In the latter reference, it was seen that BDNK theory and Navier–Stokes theory lead to unphysical solutions if sufficiently large initial gradients are present. This bears evidence that the content of these theories is similar since these solutions arise for initial Knudsen numbers of the same order of magnitude.

Furthermore, we point out that Navier–Stokes theory can be obtained from the BDNK equations by replacing the time-like derivatives of $\alpha_{0}$, $\beta_{0}$, and $u^{\mu }$ in the constitutive relations ([Disp-formula FD135a-entropy-26-00189]) using a first-order truncation of the conservation laws ([Disp-formula FD4a-entropy-26-00189]). Performing this substitution, we find relations between the transport coefficients appearing in BDNK theory and those of Navier–Stokes theory,
(137a)$$\begin{array}{cc} \zeta & ={\left.\frac{\partial p_{0}}{\partial n_{0}}\right|}_{\varepsilon_{0}}\beta_{0}\zeta^{\left(\alpha \right)}+{\left.\frac{\partial p_{0}}{\partial \varepsilon_{0}}\right|}_{n_{0}}\zeta^{\left(\beta \right)}+\zeta^{\left(\theta \right)}\;, \end{array}$$
(137b)$$\begin{array}{cc} -\xi & ={\left.\frac{\partial p_{0}}{\partial n_{0}}\right|}_{\varepsilon_{0}}\beta_{0}\xi^{\left(\alpha \right)}+{\left.\frac{\partial p_{0}}{\partial \varepsilon_{0}}\right|}_{n_{0}}\xi^{\left(\beta \right)}+\xi^{\left(\theta \right)}\;, \end{array}$$
(137c)$$\begin{array}{cc} -\chi & ={\left.\frac{\partial p_{0}}{\partial n_{0}}\right|}_{\varepsilon_{0}}\beta_{0}\chi^{\left(\alpha \right)}+{\left.\frac{\partial p_{0}}{\partial \varepsilon_{0}}\right|}_{n_{0}}\chi^{\left(\beta \right)}+\chi^{\left(\theta \right)}\;, \end{array}$$
(137d)$$\begin{array}{cc} \varkappa & =\varkappa^{\left(\alpha \right)}-\frac{p_{0}}{\varepsilon_{0}+p_{0}}\varkappa^{\left(\beta \right)}\;, \end{array}$$
(137e)$$\begin{array}{cc} -\lambda & =\lambda^{\left(\alpha \right)}-\frac{p_{0}}{\varepsilon_{0}+p_{0}}\lambda^{\left(\beta \right)}\;. \end{array}$$
This implies that, in general, $\zeta \neq \zeta^{\left(\theta \right)}$ and $\varkappa \neq \varkappa^{\left(\alpha \right)}$, for instance. This mapping between the coefficients was first derived via fluid-dynamical frame transformations in [73]. We also note the connection between the coefficients in the present section and the ones of Section 3.4, namely
(138a)$$\begin{array}{cc} \; & \nu_{1}=-\xi^{\left(\theta \right)},\;\nu_{2}=-\xi^{\left(\beta \right)},\;\nu_{3}=\xi^{\left(\alpha \right)}, \end{array}$$
(138b)$$\begin{array}{cc} \; & \varepsilon_{1}=-\chi^{\left(\theta \right)},\;\varepsilon_{2}=-\chi^{\left(\beta \right)},\;\varepsilon_{3}=\chi^{\left(\alpha \right)}, \end{array}$$
(138c)$$\begin{array}{cc} \; & \pi_{1}=\zeta^{\left(\theta \right)},\;\pi_{2}=\zeta^{\left(\beta \right)},\;\pi_{3}=-\zeta^{\left(\alpha \right)}, \end{array}$$
(138d)$$\begin{array}{cc} \; & \gamma_{1}=\gamma_{2}=-\varkappa^{\left(\beta \right)},\;\gamma_{3}=\varkappa^{\left(\alpha \right)}, \end{array}$$
(138e)$$\begin{array}{cc} \; & \theta_{1}=\theta_{2}=-\lambda^{\left(\beta \right)},\;\theta_{3}=\lambda^{\left(\alpha \right)}. \end{array}$$
Thus, the fact that the constitutive relations ([Disp-formula FD135a-entropy-26-00189]) arise from derivatives of the equilibrium distribution $f_{0k}$ yields the constraints $\gamma_{1}=\gamma_{2}$ and $\theta_{1}=\theta_{2}$.

Other approaches for the derivation of BDNK theory are possible. Indeed, in [89], it is discussed how it may be derived from holography, using the fluid/gravity correspondence [142]. The interesting properties of BDNK theory have also motivated some numerical studies [81,82,84]. In particular, in [82] after studying various numerical configurations such as the one-dimensional shock tube and the two-dimensional Kelvin-Helmholtz instability, the authors conclude that BDNK theory is “sufficiently stable and accurate to be applied to a variety of relativistic fluid-dynamics problems, where first-order dissipation might be relevant”. In [143], consequences of BDNK theory in the non-relativistic limit are explored. Recently, in [144], it was seen how causality and stability constrain the set of matching parameters *q*, *s*, and *z* that define the local-equilibrium state for a particular interaction.

### 5.4. Israel–Stewart Theory and 14-Moments Approximation

In the last decades, the most widely used variants of relativistic dissipative fluid dynamics in heavy-ion physics are transient fluid-dynamical theories. The main feature of this class of theories is that the dissipative currents are promoted to independent dynamical variables obeying nonlinear coupled relaxation-type equations of motion. These equations evolve asymptotically to the Navier–Stokes constitutive relations. The main motivation for the development of these theories in the context of special and general relativity are the causality and stability problems stated in Section 2.7. The derivation of these theories is based on ideas first developed by Grad [145,146,147] and Israel and Stewart [8,148]. The former were developed in the non-relativistic regime after the recognition that Burnett and super-Burnett equations of motion [69] (obtained in higher orders of the Chapman–Enskog expansion) feature instabilities [71,149,150,151].

As is the case with Hilbert and Navier–Stokes theories, the equations of motion for Israel–Stewart theory emerge as a result of a procedure that reduces the degrees of freedom of the microscopic theory. In the present case, the dynamics of the Boltzmann equation is translated into the dynamics of the momentum-space moments of the phase-space distribution function. They are the expansion coefficients of $f_{k}$ in terms of a complete set of functions. In a non-relativistic scenario, the expansion is made in terms of Hermite polynomials [146,147]. For relativistic systems, however, finding a suitable set of orthogonal polynomials is a very challenging task [115,152][note 13]. To circumvent this, Israel and Stewart expanded the single-particle distribution in a Taylor series in 4-momentum $k^{\mu }$. A major limitation of the procedure outlined in the last section appears when obtaining the expansion coefficients, which cannot be easily done due to the non-orthogonality of the basis. This led them to develop another method which consists of truncating the Taylor series at a given order and matching the remaining expansion coefficients in terms of the components of $N^{\mu }$ and $T^{\mu \nu }$, the 14-moments approximation. Procedures analogous to this are used for the derivation of transient fluid-dynamical theories in various instances [47,128,154,155,156]. However, the lack of a systematic power-counting procedure, as in the Chapman–Enskog case, has led to the development of further truncation procedures, which we shall discuss in Section 5.5.

For the sake of brevity, we shall discuss the 14-moment approximation exclusively for Landau matching conditions. A more general discussion is contained in [128]. We start by separating $f_{k}$ into equilibrium and non-equilibrium parts,
(139)$$f_{k}=f_{0k}+\delta f_{k}\equiv f_{0k}\left(1+G_{k}\phi_{k}\right)\;,$$
where $G_{k}$ is a function of $E_{k}=u\cdot k$, whose choice will be discussed below. If $G_{k}={\tilde{f}}_{0k}$, this coincides with the usual definition, $\phi_{k}=\delta f_{k}/f_{0k}{\tilde{f}}_{0k}$. Next, $\phi_{k}$ is expanded in terms of a complete tensor basis formed by $k^{\mu }$. Israel and Stewart [148] chose
(140)$$\phi_{k}=\epsilon +\epsilon_{\mu }k^{\mu }+\epsilon_{\mu \nu }k^{\mu }k^{\nu }+\ldots \;,$$
where the $\epsilon_{\mu_{1}\cdots \mu_{\ell }}$-coefficients inherit the space-time dependence of $\phi_{k}$. However, the main disadvantage of this expansion is that, since the tensor basis $\left\{1,k^{\mu },k^{\mu }k^{\nu },\ldots \right\}$ is not irreducible, it is not possible to derive individual equations of motion for the $\epsilon_{\mu_{1}\cdots \mu_{\ell }}$ coefficients, i.e., one is not able to separate each tensor order by, e.g., integrating over momentum space. Then, these coefficients are obtained only in approximate form by truncating the expansion at second order and matching them to moments of $f_{k}$. Moreover, this procedure is not unique and may lead to changes in the transport coefficients when the expansion is truncated at different orders [48].

Instead, in [42], the basis chosen for the expansion is formed by the irreducible tensors $\{1,k^{\langle \mu \rangle },k^{\langle \mu }k^{\nu \rangle },\ldots ,$$k^{\langle \mu_{1}}\cdots k^{\mu_{\ell }\rangle },\ldots \}$ and orthogonal polynomials in $E_{k}$,
(141)$$P_{kn}^{\left(\ell \right)}=\sum_{r=0}^{n}a_{nr}^{\left(\ell \right)}E_{k}^{r}\;.$$
The irreducible tensors obey the property (Equation 268). In turn, the polynomials $P_{kn}^{\left(\ell \right)}$ are constructed to obey the orthogonality condition,
(142)$$\frac{\ell !}{(2\ell +1)!!}\int dK\;{\left({∆}^{\mu \nu }k_{\mu }k_{\nu }\right)}^{\ell }P_{km}^{\left(\ell \right)}P_{kn}^{\left(\ell \right)}G_{k}f_{0k}=A_{n}^{\left(\ell \right)}\delta_{mn}\;.$$
Using the above mentioned basis, the function $\phi_{k}$ can be expanded as
(143)$$\phi_{k}=\sum_{\ell =0}^{\infty }\sum_{n=0}^{N_{\ell }}\Phi_{n}^{\mu_{1}\cdots \mu_{\ell }}P_{kn}^{\left(\ell \right)}k_{\langle \mu_{1}}\cdots k_{\mu_{\ell }\rangle }\;,$$
where the coefficients $\Phi_{n}^{\mu_{1}\cdots \mu_{\ell }}$ are obtained by multiplying the above equation by $G_{k}P_{km}^{\left(\ell \right)}$$\times k_{\langle \mu_{1}}\cdots k_{\mu_{\ell }\rangle }$ and integrating over momentum space, leading to
(144)$$\Phi_{n}^{\mu_{1}\cdots \mu_{\ell }}=\frac{1}{A_{n}^{\left(\ell \right)}}\int dK\;P_{km}^{\left(\ell \right)}k^{\langle \mu_{1}}\cdots k^{\mu_{\ell }\rangle }\delta f_{k}\;,$$
where the orthogonality relations (Equation 268) and (Equation 142) have been employed. Substituting the above result into Equation (Equation 143) and using Equation (Equation 141), we expand the non-equilibrium single-particle distribution function as
(145)$$f_{k}=f_{0k}+f_{0k}\sum_{\ell =0}^{\infty }\sum_{n=0}^{N_{\ell }}G_{k}H_{kn}^{\left(\ell \right)}\rho_{n}^{\mu_{1}\cdots \mu_{\ell }}k_{\langle \mu_{1}}\cdots k_{\mu_{\ell }\rangle }\;,$$
where we defined the following functions of $E_{k}$:(146)$$H_{kn}^{\left(\ell \right)}\equiv \frac{1}{A_{n}^{\left(\ell \right)}}\sum_{m=n}^{N_{\ell }}a_{mn}^{\left(\ell \right)}P_{km}^{\left(\ell \right)}\;,$$
and the irreducible moments of $\delta f_{k}$
(147)$$\rho_{r}^{\mu_{1}\cdots \mu_{\ell }}\equiv \int dK\;E_{k}^{r}k^{\langle \mu_{1}}\cdots k^{\mu_{\ell }\rangle }\delta f_{k}\;.$$
From Equations ([Disp-formula FD81a-entropy-26-00189]) and ([Disp-formula FD85a-entropy-26-00189]), defining the dissipative currents in terms of moments of the single-particle distribution function $f_{k}$, we identify
(148a)$$\begin{array}{cc} \; & \delta n=\rho_{1}\equiv 0\;,\;\delta \varepsilon =\rho_{2}\equiv 0\;,\;\Pi =\frac{1}{3}\left(\rho_{2}-m^{2}\rho_{0}\right)\;, \end{array}$$
(148b)$$\begin{array}{cc} \; & n^{\mu }=\rho_{0}^{\mu }\;,\;h^{\mu }=\rho_{1}^{\mu }\equiv 0\;,\;n^{\mu }=\rho_{0}^{\mu }\;,\;\pi^{\mu \nu }=\rho_{0}^{\mu \nu }\;, \end{array}$$
where we note that the matching conditions (Equation 88), responsible for defining the variables of the local-equilibrium state, imply that not all moments $\rho_{r}^{\mu_{1}\cdots \mu_{\ell }}$ are linearly independent.

The choice of $G_{k}$ changes the behavior of the polynomial basis used to expand $\phi_{k}$. This can be exploited to achieve better convergence in regimes of small or large values of $\beta_{0}E_{k}$. Indeed, for the choice
(149)$$G_{k}={\tilde{f}}_{0k}\;,$$
which was employed in [42], the truncated series (Equation 145) is a good approximation of $f_{k}$ for small $\beta_{0}E_{k}$. The choice (Equation 149) shall be employed in the remainder of the present section. An alternative is to consider that the weight $G_{k}$ might depend on the tensor rank *ℓ*. For instance, the choice $G_{k}^{\left(\ell \right)}={\tilde{f}}_{0k}/{\left(1+\beta_{0}E_{k}\right)}^{\ell }$ is expected to also work for large values of $\beta_{0}E_{k}$, because, in this regime, it is equivalent to the basis $1/(\beta_{0}E_{k})$, $1/{\left(\beta_{0}E_{k}\right)}^{2}$, $1/{\left(\beta_{0}E_{k}\right)}^{3},\ldots$, while in the regime of small $\beta_{0}E_{k}$, this basis is equivalent to $\beta_{0}E_{k}$, ${\left(\beta_{0}E_{k}\right)}^{2}$, ${\left(\beta_{0}E_{k}\right)}^{3},\ldots$ [48]. Summarily, this choice is important for the convergence of the transport coefficients for transient theories of fluid dynamics (see Section 5.5).

The equations of motion for the moments (Equation 147) can be obtained from the Boltzmann Equation (Equation 76), by multiplying both sides with $E_{k}^{r-1}k^{\langle \mu_{1}}\cdots k^{\mu_{\ell }\rangle }$ and integrating over momentum space. After some algebraic manipulations, which involve an integration by parts and some identities involving the irreducible tensors, one can show that the scalar, vector, and rank-2 tensor moments obey, respectively [42,48],
(150a)$$\begin{array}{cc} {\dot{\rho }}_{r}-C_{r-1}= & \;\alpha_{r}^{\left(0\right)}\theta -\frac{G_{2r}}{D_{20}}\Pi \theta +\frac{G_{2r}}{D_{20}}\pi^{\mu \nu }\sigma_{\mu \nu }+\frac{G_{3r}}{D_{20}}\partial \cdot n+\left(r-1\right)\rho_{r-2}^{\mu \nu }\sigma_{\mu \nu }\\ \; & +r\rho_{r-1}^{\mu }{\dot{u}}_{\mu }-\nabla_{\mu }\rho_{r-1}^{\mu }-\frac{1}{3}\left[\left(r+2\right)\rho_{r}-\left(r-1\right)m^{2}\rho_{r-2}\right]\theta \;, \end{array}$$
(150b)$$\begin{array}{cc} {\dot{\rho }}_{r}^{\langle \mu \rangle }-C_{r-1}^{\langle \mu \rangle }= & \;\alpha_{r}^{\left(1\right)}I^{\mu }+\omega^{\mu \nu }\rho_{r\nu }+\frac{1}{3}\left[\left(r-1\right)m^{2}\rho_{r-2}^{\mu }-\left(r+3\right)\rho_{r}^{\mu }\right]\theta -{∆}_{\lambda }^{\mu }\nabla_{\nu }\rho_{r-1}^{\lambda \nu }+r\rho_{r-1}^{\mu \nu }{\dot{u}}_{\nu }\\ \; & +\frac{1}{5}\left[2\left(2r-1\right)m^{2}\rho_{r-2}^{\nu }-\left(2r+3\right)\rho_{r}^{\nu }\right]\sigma_{\nu }^{\mu }+\frac{1}{3}\left[m^{2}r\rho_{r-1}-\left(r+3\right)\rho_{r+1}\right]{\dot{u}}^{\mu }\\ \; & +\frac{\beta_{0}J_{r+2,1}}{\varepsilon_{0}+p_{0}}\left(\Pi {\dot{u}}^{\mu }-\nabla^{\mu }\Pi +{∆}_{\nu }^{\mu }\partial_{\lambda }\pi^{\lambda \nu }\right)-\frac{1}{3}\nabla^{\mu }\left(m^{2}\rho_{r-1}-\rho_{r+1}\right)\\ \; & +\left(r-1\right)\rho_{r-2}^{\mu \nu \lambda }\sigma_{\lambda \nu }\;, \end{array}$$
(150c)$$\begin{array}{cc} {\dot{\rho }}_{r}^{\langle \mu \nu \rangle }-C_{r-1}^{\langle \mu \nu \rangle }= & \;2\alpha_{r}^{\left(2\right)}\sigma^{\mu \nu }-\frac{2}{7}\left[\left(2r+5\right)\rho_{r}^{\lambda \langle \mu }-2m^{2}\left(r-1\right)\rho_{r-2}^{\lambda \langle \mu }\right]\sigma_{\lambda }^{\nu \rangle }+2\rho_{r}^{\lambda \langle \mu }{\omega^{\nu \rangle }}_{\lambda }\\ \; & +\frac{2}{15}\left[\left(r+4\right)\rho_{r+2}-\left(2r+3\right)m^{2}\rho_{r}+\left(r-1\right)m^{4}\rho_{r-2}\right]\sigma^{\mu \nu }\\ \; & +\frac{2}{5}\nabla^{\langle \mu }\left(\rho_{r+1}^{\nu \rangle }-m^{2}\rho_{r-1}^{\nu \rangle }\right)-\frac{2}{5}\left[\left(r+5\right)\rho_{r+1}^{\langle \mu }-rm^{2}\rho_{r-1}^{\langle \mu }\right]{\dot{u}}^{\nu \rangle }\\ \; & -\frac{1}{3}\left[\left(r+4\right)\rho_{r}^{\mu \nu }-m^{2}\left(r-1\right)\rho_{r-2}^{\mu \nu }\right]\theta \\ \; & +\left(r-1\right)\rho_{r-2}^{\mu \nu \lambda \rho }\sigma_{\lambda \rho }-{∆}_{\alpha \beta }^{\mu \nu }\nabla_{\lambda }\rho_{r-1}^{\alpha \beta \lambda }+r\rho_{r-1}^{\mu \nu \lambda }{\dot{u}}_{\lambda }\;, \end{array}$$
In Equations ([Disp-formula FD150a-entropy-26-00189]), we defined
(151a)$$\begin{array}{cc} \alpha_{r}^{\left(0\right)} & \equiv \frac{G_{r+1,2}}{G_{31}}n_{0}+\frac{G_{r2}}{G_{31}}\left(\varepsilon_{0}+p_{0}\right)+\beta_{0}J_{r+1,1}\;, \end{array}$$
(151b)$$\begin{array}{cc} \alpha_{r}^{\left(1\right)} & \equiv J_{r+1,1}-\frac{n_{0}}{\varepsilon_{0}+p_{0}}J_{r+2,1}\;, \end{array}$$
(151c)$$\begin{array}{cc} \alpha_{r}^{\left(2\right)} & \equiv \beta_{0}J_{r+3,2}\;, \end{array}$$
(151d)$$\begin{array}{cc} D_{nm} & \equiv J_{n+1,m}J_{n-1,m}-J_{nm}^{2}\;, \end{array}$$
while $G_{nm}$ is defined in Equation ([Disp-formula FD114c-entropy-26-00189]). It is readily seen that the most non-trivial part in the computation of transport coefficients in fluid-dynamical theories is how to express the collisional moments $C_{r-1}^{\mu_{1}\cdots \mu_{\ell }}$ in terms of the moments $\rho_{r}^{\mu_{1}\cdots \mu_{\ell }}$. As we shall see, this is a crucial step since the relaxation times which can render the theory causal and stable arise from this term. We formally separate $C[f_{k}]$ into terms that are linear and nonlinear with respect to the deviation function $\phi_{k}$, namely,
(152)$$C_{r-1}^{\mu_{1}\cdots \mu_{\ell }}=L_{r-1}^{\mu_{1}\cdots \mu_{\ell }}+N_{r-1}^{\mu_{1}\cdots \mu_{\ell }}\;.$$
By construction, $L_{r-1}^{\mu_{1}\cdots \mu_{\ell }}$ contains only contributions that are linear in $\phi_{k}$, while $N_{r-1}^{\mu_{1}\cdots \mu_{\ell }}$ collects all contributions of higher order in $\phi_{k}$. In the present discussion, we shall neglect the nonlinear terms $N_{r-1}^{\mu_{1}\cdots \mu_{\ell }}$, but they have been considered, for instance, in [48,157]. Then, the linear term can be written as [42],
(153a)$$\begin{array}{cc} L_{r-1}^{\mu_{1}\cdots \mu_{\ell }}\; & =\int dQ\;dQ^{\prime }\;dK^{\prime }\;dK\;W_{kk^{\prime }↔qq^{\prime }}f_{0k}f_{0k^{\prime }}{\tilde{f}}_{0q}{\tilde{f}}_{0q^{\prime }}E_{k}^{r-1}k^{\langle \mu_{1}}\cdots k^{\mu_{\ell }\rangle }\left(\phi_{q}+\phi_{q^{\prime }}-\phi_{k}-\phi_{k^{\prime }}\right)\\ \; & =-\sum_{n=0}^{\infty }L_{rn}^{\left(\ell \right)}\rho_{n}^{\mu_{1}\cdots \mu_{\ell }}\;, \end{array}$$
(153b)$$\begin{array}{cc} L_{rn}^{\left(\ell \right)}\; & =\frac{1}{2\ell +1}\int dQ\;dQ^{\prime }\;dK^{\prime }\;dK\;W_{kk^{\prime }↔qq^{\prime }}f_{0k}f_{0k^{\prime }}{\tilde{f}}_{0q}{\tilde{f}}_{0q^{\prime }}E_{k}^{r-1}k^{\langle \mu_{1}}\cdots k^{\mu_{\ell }\rangle }\\ \; & \times \left(H_{kn}^{\left(\ell \right)}k_{\langle \mu_{1}}\cdots k_{\mu_{\ell }\rangle }+H_{k^{\prime }n}^{\left(\ell \right)}k_{\langle \mu_{1}}^{\prime }\cdots k_{\mu_{\ell }\rangle }^{\prime }-H_{qn}^{\left(\ell \right)}q_{\langle \mu_{1}}\cdots q_{\mu_{\ell }\rangle }-H_{q^{\prime }n}^{\left(\ell \right)}q_{\langle \mu_{1}}^{\prime }\cdots q_{\mu_{\ell }\rangle }^{\prime }\right)\;. \end{array}$$
In the derivation of the fluid-dynamical equations of motion, an essential step will be to invert the matrices $L^{\left(\ell =0,1,2\right)}$. The collisional invariants in binary elastic scattering imply that the following rows of these matrices vanish: $L_{1n}^{\left(0\right)}=L_{2n}^{\left(0\right)}=L_{1n}^{\left(1\right)}=0$. For the inversion, one therefore has to exclude these rows. The matrices nevertheless remain square matrices because the terms in the series ([Disp-formula FD153a-entropy-26-00189]) which are multiplied with moments $\rho_{n}^{\mu_{1}\cdots \mu_{\ell }}$ that vanish due to the choice of matching conditions are absent, and thus the corresponding columns in the matrices $L^{\left(\ell =0,1\right)}$ can be excluded as well. For instance, for Landau matching conditions, one can omit the columns $L_{r1}^{\left(0\right)}$, $L_{r2}^{\left(0\right)}$, and $L_{r1}^{\left(1\right)}$.

The 14-moment approximation starts with the truncation of the irreducible expansion (Equation 143),
(154)$$\phi_{k}≃\sum_{n=0}^{N_{0}}\Phi_{n}{\hat{P}}_{kn}^{\left(0\right)}+\sum_{n=0}^{N_{1}}\Phi_{n}^{\mu }{\hat{P}}_{kn}^{\left(1\right)}k_{\langle \mu \rangle }+\Phi_{0}^{\mu \nu }k_{\langle \mu }k_{\nu \rangle }\;,$$
where $N_{0}=2$ and $N_{1}=1$ (these values may change for other matching conditions [128]). By construction, in this truncation scheme, all irreducible moments $\rho_{r}^{\mu_{1}\cdots \mu_{\ell }}$ of rank higher than 2 vanish. The irreducible moments of rank 0, 1, and 2 will always be expressed in terms of the expansion coefficients $\Phi_{n}$, $\Phi_{n}^{\mu }$, and $\Phi_{0}^{\mu \nu }$.

The next step is to determine the expansion coefficients using the definitions of the dissipative currents, given in Equations ([Disp-formula FD81a-entropy-26-00189]), and the general matching conditions provided in Equations (Equation 88) and (Equation 89). This leads to a system of algebraic equations that can be solved separately for the scalar, vector, and tensor expansion coefficients. The expression for the rank-2 expansion coefficient, $\Phi_{0}^{\mu \nu }$, is obtained by inserting the truncated moment expansion (Equation 154) into the definition of the shear-stress tensor given in Equation ([Disp-formula FD81b-entropy-26-00189]). This leads to the simple relation
(155)$$\Phi_{0}^{\mu \nu }=\frac{\pi^{\mu \nu }}{{\left(1,1\right)}_{2}}=\frac{\pi^{\mu \nu }}{2J_{42}},$$
where we used the notation
(156)$${\left(\phi ,\psi \right)}_{\ell }=\frac{\ell !}{(2\ell +1)!!}{\langle {\left({∆}^{\mu \nu }k_{\mu }k_{\nu }\right)}^{\ell }\phi_{k}\psi_{k}\rangle }_{\tilde{\mathrm{eq}}}\;,$$
in terms of which the orthogonality relation (Equation 142), taking into account Equation (Equation 149), reads
(157)$${\left({\hat{P}}_{km}^{\left(\ell \right)},{\hat{P}}_{kn}^{\left(\ell \right)}\right)}_{\ell }=A_{m}^{\left(\ell \right)}\delta_{mn}.$$

Equations for the vector expansion coefficients, $\Phi_{0}^{\mu }$ and $\Phi_{1}^{\mu }$, are obtained by inserting the truncated moment expansion (Equation 154) into the definition of $n^{\mu }$, given in Equation ([Disp-formula FD81b-entropy-26-00189]), and also into the matching condition $\rho_{1}^{\mu }\equiv 0$. This will lead to two distinct equations that can be cast in the following matrix form:(158)$$\left(\begin{array}{cc} {\left(1,1\right)}_{1} & 0\\ {\left({\hat{P}}_{k0}^{\left(1\right)},E_{k}\right)}_{1} & {\left({\hat{P}}_{k1}^{\left(1\right)},E_{k}\right)}_{1} \end{array}\right)\left(\begin{array}{c} \Phi_{0}^{\mu }\\ \Phi_{1}^{\mu } \end{array}\right)=\left(\begin{array}{c} n^{\mu }\\ 0 \end{array}\right)\;.$$
The solution of this equation gives $\Phi_{0}^{\mu }$ and $\Phi_{1}^{\mu }$ in terms of $n^{\mu }$.

Equations for the scalar expansion coefficients, $\Phi_{0}$, $\Phi_{1}$, and $\Phi_{2}$ are obtained by inserting the truncated moment expansion (Equation 154) into the definitions of δn, δε, and Π, given in Equations ([Disp-formula FD81a-entropy-26-00189]), and also into the matching condition $\rho_{1}^{\mu }\equiv 0$. This will lead to three distinct equations that can be cast in the following matrix form:(159)$$M\cdot \vec{\Phi }=\vec{\Pi }\;,$$
where
(160a)$$\begin{array}{ccc} M & =\left(\begin{array}{ccc} {\left({\hat{P}}_{k0}^{\left(0\right)},E_{k}\right)}_{0} & {\left({\hat{P}}_{k1}^{\left(0\right)},E_{k}\right)}_{0} & {\left({\hat{P}}_{k2}^{\left(0\right)},E_{k}\right)}_{0}\\ {\left({\hat{P}}_{k0}^{\left(0\right)},E_{k}^{2}\right)}_{0} & {\left({\hat{P}}_{k1}^{\left(0\right)},E_{k}^{2}\right)}_{0} & {\left({\hat{P}}_{k2}^{\left(0\right)},E_{k}^{2}\right)}_{0}\\ {\left({\hat{P}}_{k0}^{\left(0\right)},E_{k}^{2}-m^{2}\right)}_{0} & {\left({\hat{P}}_{k1}^{\left(0\right)},E_{k}^{2}-m^{2}\right)}_{0} & {\left({\hat{P}}_{k2}^{\left(0\right)},E_{k}^{2}-m^{2}\right)}_{0} \end{array}\right) & \;, \end{array}$$
(160b)$$\begin{array}{cc} \vec{\Phi }\; & =\left(\begin{array}{c} \Phi_{0}\\ \Phi_{1}\\ \Phi_{2} \end{array}\right)\;,\;\vec{\Pi }=\left(\begin{array}{c} 0\\ 0\\ 3\Pi \end{array}\right)\;. \end{array}$$
In the limit of massless particles, m→0, the energy-momentum tensor becomes traceless, $T_{\;\mu }^{\mu }=\varepsilon_{0}-3p_{0}-3\Pi =0$, thus effectively removing one degree of freedom (δε or Π). This affects the above discussion, as the second and third rows of M become identical, and in practice, we have a linear homogeneous system defined by a non-singular (2×2) matrix. Thus, the Landau matching conditions imply that $\Phi_{0}=\Phi_{1}=0$ in the massless limit.

Now that the expansion coefficients Φ have been computed, we can express all moments of rank 0, 1, and 2 in terms of the dissipative fields $\left(\Pi ,n^{\mu },\pi^{\mu \nu }\right)$,
(161)$$\rho_{r}^{\mu \nu }≃Q_{\pi }^{\left(r\right)}\pi^{\mu \nu }\;,\;\rho_{r}^{\mu }≃Q_{n}^{\left(r\right)}n^{\mu }\;,\;\rho_{r}≃Q_{\Pi }^{\left(r\right)}\Pi \;,$$
where the coefficients Q are expressed in terms of thermodynamic integrals,
(162a)$$\begin{array}{cc} \; & Q_{n}^{\left(r\right)}=\frac{1}{{\left(1,1\right)}_{1}}\left[{\left(E_{k}^{r},1\right)}_{1}-\frac{{\left({\hat{P}}_{k0}^{\left(1\right)},E_{k}\right)}_{1}}{{\left({\hat{P}}_{k1}^{\left(1\right)},E_{k}\right)}_{1}}{\left(E_{k}^{r},{\hat{P}}_{k1}^{\left(1\right)}\right)}_{1}\right]\;, \end{array}$$
(162b)$$\begin{array}{cc} \; & Q_{\Pi }^{\left(r\right)}=\frac{3}{\det M}\left[{\left(E_{k}^{r},1\right)}_{0}{\left[M\right]}_{11}-{\left(E_{k}^{r},{\hat{P}}_{k1}^{\left(0\right)}\right)}_{0}{\left[M\right]}_{22}+{\left(E_{k}^{r},{\hat{P}}_{k2}^{\left(0\right)}\right)}_{0}{\left[M\right]}_{33}\right]\;, \end{array}$$
(162c)$$\begin{array}{cc} \; & Q_{\pi }^{\left(r\right)}=\frac{J_{r+4,2}}{J_{42}}\;. \end{array}$$
Here, ${\left[M\right]}_{IJ}$ denotes the (IJ) co-factor of M, i.e., the determinant of the sub-matrix obtained from M by removing the *I*-th row and the *J*-th column, multiplied by ${\left(-1\right)}^{I+J}$. As mentioned, moments of rank 3 or higher vanish in this approximation. With these prescriptions, it is possible to derive a closed system of partial differential equations in terms of the dissipative currents. Indeed, we obtain from Equations ([Disp-formula FD150a-entropy-26-00189]) for r=0 precisely Equations ([Disp-formula FD23a-entropy-26-00189]), but where all K- and R-terms are zero,
(163a)$$\begin{array}{cc} \tau_{\Pi }\dot{\Pi }+\Pi = & -\zeta \theta -\delta_{\Pi \Pi }\Pi \theta -\ell_{\Pi n}\nabla \cdot n-\tau_{\Pi n}n\cdot F-\lambda_{\Pi n}n\cdot I+\lambda_{\Pi \pi }\pi^{\mu \nu }\sigma_{\mu \nu }\;, \end{array}$$
(163b)$$\begin{array}{cc} \tau_{n}{\dot{n}}^{\langle \mu \rangle }+n^{\mu }= & \;\varkappa I^{\mu }-\delta_{nn}n^{\mu }\theta -\ell_{n\Pi }\nabla^{\mu }\Pi +\ell_{n\pi }\nabla_{\lambda }\pi^{\lambda \langle \mu \rangle }+\tau_{n\Pi }\Pi F^{\mu }\\ \; & -\tau_{n\pi }\pi^{\mu \nu }F_{\nu }-\lambda_{nn}\sigma^{\mu \nu }n_{\nu }+\lambda_{n\Pi }\Pi I^{\mu }-\lambda_{n\pi }\pi^{\mu \nu }I_{\nu }+\tau_{n}\omega^{\mu \nu }n_{\nu }\;, \end{array}$$
(163c)$$\begin{array}{cc} \tau_{\pi }{\dot{\pi }}^{\langle \mu \nu \rangle }+\pi^{\mu \nu }= & \;2\eta \sigma^{\mu \nu }-\delta_{\pi \pi }\pi^{\mu \nu }\theta +\ell_{\pi n}\nabla^{\langle \mu }n^{\nu \rangle }-\tau_{\pi n}n^{\langle \mu }F^{\nu \rangle }+\lambda_{\pi \Pi }\Pi \sigma^{\mu \nu }\\ \; & -\lambda_{\pi \pi }\pi_{\lambda }^{\langle \mu }\sigma^{\nu \rangle \lambda }+\lambda_{\pi n}n^{\langle \mu }I^{\nu \rangle }+2\tau_{\pi }\pi_{\lambda }^{\langle \mu }\omega^{\nu \rangle \lambda }\;. \end{array}$$
Note that the transport coefficients coupling the dissipative currents to the fluid vorticity tensor, $\tau_{n\omega }$ and $\tau_{\pi \omega }$ in Equations ([Disp-formula FD24b-entropy-26-00189]), are identical to the corresponding relaxation times, $\tau_{n\omega }\equiv \tau_{n}$ and $\tau_{\pi \omega }\equiv \tau_{\pi }$. The expressions for the bulk viscosity, particle diffusion, shear viscosity, and the corresponding relaxation times read,
(164a)$$\begin{array}{cc} \; & \zeta =m^{2}\tau_{\Pi }\alpha_{0}^{\left(0\right)}\;,\;\tau_{\Pi }={\left(-m^{2}\sum_{j=0}^{\infty }L_{0j}^{\left(2\right)}Q_{\Pi }^{\left(j\right)}\right)}^{-1}\;, \end{array}$$
(164b)$$\begin{array}{cc} \; & \varkappa =\tau_{n}\alpha_{0}^{\left(1\right)}\;,\;\tau_{n}={\left(\sum_{j=0}^{\infty }L_{0j}^{\left(1\right)}Q_{n}^{\left(j\right)}\right)}^{-1}\;, \end{array}$$
(164c)$$\begin{array}{cc} \; & \eta =\tau_{\pi }\alpha_{0}^{\left(2\right)}\;\;\tau_{\pi }={\left(\sum_{j=0}^{\infty }L_{0j}^{\left(2\right)}Q_{\pi }^{\left(j\right)}\right)}^{-1}\;. \end{array}$$
The remaining transport coefficients in Equation ([Disp-formula FD163a-entropy-26-00189]) for the bulk viscous pressure read
(165a)$$\begin{array}{cc} \; & \frac{\delta_{\Pi \Pi }}{\tau_{\Pi }}=\frac{2}{3}+\frac{m^{4}}{3}Q_{\Pi }^{\left(-2\right)}-\frac{G_{20}}{D_{20}}m^{2}\;,\;\frac{\lambda_{\Pi \pi }}{\tau_{\Pi }}=Q_{\Pi }^{\left(-2\right)}m^{2}-\frac{G_{20}}{D_{20}}m^{2}\;, \end{array}$$
(165b)$$\begin{array}{cc} \; & \frac{\ell_{\Pi n}}{\tau_{\Pi }}=\frac{G_{30}}{D_{20}}m^{2}-Q_{n}^{\left(-1\right)}m^{2}\;,\;\frac{\tau_{\Pi n}}{\tau_{\Pi }}=\frac{\beta_{0}}{\varepsilon_{0}+p_{0}}\frac{\partial Q_{n}^{\left(-1\right)}}{\partial \beta_{0}}-\frac{m^{2}}{\varepsilon_{0}+p_{0}}\frac{G_{30}}{D_{20}}\;, \end{array}$$
(165c)$$\begin{array}{cc} \; & \frac{\lambda_{\Pi n}}{\tau_{\Pi }}=-m^{2}\frac{\partial Q_{n}^{\left(-1\right)}}{\partial \alpha_{0}}-\frac{m^{2}n_{0}}{\varepsilon_{0}+p_{0}}\frac{\partial Q_{n}^{\left(-1\right)}}{\partial \beta_{0}}\;. \end{array}$$
The transport coefficients in the equation of motion ([Disp-formula FD163b-entropy-26-00189]) for the particle diffusion current read
(166a)$$\begin{array}{cc} \; & \frac{\delta_{nn}}{\tau_{n}}=1+\frac{m^{2}}{3}Q_{n}^{\left(-2\right)}\;,\;\frac{\ell_{n\Pi }}{\tau_{n}}=\frac{\beta_{0}J_{r+2,1}}{\varepsilon_{0}+p_{0}}+\frac{m^{2}}{3}Q_{\Pi }^{\left(-1\right)}\;, \end{array}$$
(166b)$$\begin{array}{cc} \; & \frac{\tau_{n\Pi }}{\tau_{n}}=\frac{\beta_{0}J_{r+2,1}}{{\left(\varepsilon_{0}+p_{0}\right)}^{2}}+\frac{m^{2}\beta_{0}}{3(\varepsilon_{0}+p_{0})}\frac{\partial Q_{\Pi }^{\left(-1\right)}}{\partial \beta_{0}}\;,\;\frac{\lambda_{n\Pi }}{\tau_{n}}=-\frac{m^{2}}{3}\frac{\partial Q_{\Pi }^{\left(-1\right)}}{\partial \alpha_{0}}-\frac{m^{2}}{3}\frac{\partial Q_{\Pi }^{\left(-1\right)}}{\partial \beta_{0}}\frac{n_{0}}{\left(\varepsilon_{0}+p_{0}\right)}\;, \end{array}$$
(166c)$$\begin{array}{cc} \; & \frac{\ell_{n\pi }}{\tau_{n}}=\frac{\beta_{0}J_{r+2,1}}{\varepsilon_{0}+p_{0}}-Q_{\pi }^{\left(-1\right)}\;,\;\frac{\lambda_{nn}}{\tau_{n}}=\frac{3}{5}-\frac{2m^{2}}{5}Q_{n}^{\left(-2\right)}\;, \end{array}$$
(166d)$$\begin{array}{cc} \; & \frac{\tau_{n\pi }}{\tau_{n}}=-\frac{\beta_{0}}{\left(\varepsilon_{0}+p_{0}\right)}\frac{\partial Q_{\Pi }^{\left(-1\right)}}{\partial \beta_{0}}\;,\;\frac{\lambda_{n\pi }}{\tau_{n}}=\frac{\partial Q_{\Pi }^{\left(-1\right)}}{\partial \alpha_{0}}+\frac{n_{0}}{\left(\varepsilon_{0}+p_{0}\right)}\frac{\partial Q_{\Pi }^{\left(-1\right)}}{\partial \beta_{0}}\;. \end{array}$$
Finally, the transport coefficients in the equation of motion ([Disp-formula FD163c-entropy-26-00189]) for the shear-stress tensor read
(167a)$$\begin{array}{cc} \; & \frac{\delta_{\pi \pi }}{\tau_{\pi }}=\frac{4}{3}+m^{2}Q_{\pi }^{\left(-2\right)}\;,\;\frac{\ell_{\pi n}}{\tau_{\pi }}=-\frac{2}{5}m^{2}Q_{n}^{\left(-1\right)}\;, \end{array}$$
(167b)$$\begin{array}{cc} \; & \frac{\lambda_{\pi n}}{\tau_{\pi }}=-\frac{2m^{2}}{5}\left(\frac{\partial Q_{n}^{\left(-1\right)}}{\partial \alpha_{0}}+\frac{n_{0}}{\varepsilon_{0}+p_{0}}\frac{\partial Q_{n}^{\left(-1\right)}}{\partial \beta_{0}}\right)\;,\;\frac{\tau_{\pi n}}{\tau_{\pi }}=-\frac{2}{5}\frac{\beta_{0}m^{2}}{\varepsilon_{0}+p_{0}}\frac{\partial Q_{n}^{\left(-1\right)}}{\partial \beta_{0}}\;, \end{array}$$
(167c)$$\begin{array}{cc} \; & \frac{\lambda_{\pi \Pi }}{\tau_{\pi }}=\frac{6}{5}-\frac{2}{15}m^{4}Q_{\Pi }^{\left(-2\right)},\;\frac{\lambda_{\pi \pi }}{\tau_{\pi }}=\frac{10}{7}+\frac{4}{7}m^{2}Q_{\pi }^{\left(-2\right)}\;. \end{array}$$

Employing the present truncation procedure, the various transport coefficients in the equations of motion ([Disp-formula FD163a-entropy-26-00189]) are usually expressed in terms of Q–coefficients defined in Equation ([Disp-formula FD162a-entropy-26-00189]) and linearized collision-term matrix elements, $L_{ij}^{\left(\ell \right)}$. In [128], the scheme constructed in this section has been generalized to derive transient fluid-dynamical equations of motion with alternative matching conditions.

The procedure of reducing the degrees of freedom by explicitly matching the coefficients of the expansion of the single-particle distribution with the dissipative degrees of freedom has been employed in several instances. Usually, this kind of procedure emerges as an initial approach for the derivation of many formulations of both non-relativistic [145] and relativistic dissipative fluid-dynamical theories [47,128,154,156,158]. However, one aspect that is to be contrasted with the methods discussed in Section 5.2 and Section 5.3 is that the 14-moments truncation lacks a perturbative parameter such as the Knudsen number, with which systematic improvements can be performed via a suitable power-counting. Moreover, it has been recognized that the equations of motion do not perform well in comparison with numerical solutions of the Boltzmann equation [42,48]. These problems have led to the development of other truncations for transient fluid-dynamical theories, termed Resummed Transient Fluid Dynamics [42], which will be discussed in the next subsection.

### 5.5. Resummed Transient Fluid Dynamics

In the last subsection, we have discussed procedures for obtaining transient fluid-dynamical theories from kinetic theory by explicitly truncating an expansion of the single-particle distribution function. After recognizing that these methods are not systematically improvable, other approaches have been developed, which culminated in power-counting methods that properly take into account how large the gradients are and how far from local equilibrium the system is. These methods, in general, include contributions of all moments $\rho_{r}^{\mu_{1}\cdots \mu_{\ell }}$ at a given tensor rank *ℓ*, and have thus been termed Resummed Transient Fluid Dynamics. In particular, two methods will be discussed in this subsection: the Denicol–Niemi–Molnár–Rischke (DNMR) approach [42,48] and the order-of-magnitude or Inverse-Reynolds Dominance (IReD) method [149,159,160]. For the sake of simplicity, we restrict the discussion to Landau matching conditions.

#### 5.5.1. DNMR Approach

As discussed in Section 2.4, fluid dynamics is valid when there is a large separation between a typical microscopic scale λ characterizing the interaction of particles and a typical macroscopic scale *L*, e.g., the scale over which fluid-dynamical quantities vary. The ratio of these scales, the Knudsen number, is then a small quantity, Kn≪1. Moreover, fluid dynamics is valid near local equilibrium, which requires that the inverse Reynolds numbers (Equation 13) are also small quantities, Rei−1≪1,i=ε,n,Π,n¯,h,π. These two measures of the validity of fluid dynamics are, in principle, independent and should not be considered to be similar since a system can have initial conditions for which Kn ≪Rei−1 (i.e., the system is far from equilibrium but the gradients of thermodynamic variables are small) or vice versa. Only for times much larger than the microscopic time scales, which are ∼λ, one has Kn ∼Rei−1. Then, the dissipative currents already had sufficient time to relax to their respective Navier–Stokes values.

The DNMR truncation procedure starts by introducing the matrix $\Omega^{\left(\ell \right)}$, which diagonalizes $L^{\left(\ell \right)}$ (see Equations (Equation 152) and ([Disp-formula FD153a-entropy-26-00189])),
(168)$${\left(\Omega^{\left(\ell \right)}\right)}^{-1}L^{\left(\ell \right)}\Omega^{\left(\ell \right)}=\mathrm{diag}\left(\chi_{0}^{\left(\ell \right)},\ldots ,\chi_{j}^{\left(\ell \right)},\ldots \right)\;,$$
where $\chi_{j}^{\left(\ell \right)}$ are the eigenvalues of $L^{\left(\ell \right)}$, which without loss of generality we assume to be ordered such that
(169)$$\chi_{j}^{\left(\ell \right)}<\chi_{j+1}^{\left(\ell \right)}\;\forall \;\ell =0,1,2,\ldots \;.$$
We also introduce the tensors
(170)$$X_{i}^{\mu_{1}\cdots \mu_{\ell }}=\sum_{j=0}^{N_{\ell }}{\left[{\left(\Omega^{\left(\ell \right)}\right)}^{-1}\right]}_{ij}\rho_{j}^{\mu_{1}\cdots \mu_{\ell }}\;,$$
which are the eigenmodes of the linearized Boltzmann equation. Taking the collisional moments $C_{r-1}^{\mu_{1}\cdots \mu_{\ell }}$ as the components of a vector, and multiplying from the left with the matrix ${\left(\Omega^{\left(\ell \right)}\right)}^{-1}$, we have
(171)$$\sum_{j=0}^{N_{\ell }}{\left[{\left(\Omega^{\left(\ell \right)}\right)}^{-1}\right]}_{ij}C_{j-1}^{\mu_{1}\cdots \mu_{\ell }}=-\chi_{i}^{\left(\ell \right)}X_{i}^{\mu_{1}\cdots \mu_{\ell }}+\ldots \;,$$
where there is no sum over *i* on the right-hand side. In the above expression, the ellipsis denotes nonlinear terms in the eigenmodes. Multiplying Equations ([Disp-formula FD150a-entropy-26-00189]) with ${\left[{\left(\Omega^{\left(\ell \right)}\right)}^{-1}\right]}_{ir}$ and summing over *r* we obtain a diagonal form in terms of the eigenmodes,
(172a)$$\begin{array}{cc} {\dot{X}}_{i}+\chi_{i}^{\left(0\right)}X_{i} & =\beta_{i}^{\left(0\right)}\theta +\ldots \;, \end{array}$$
(172b)$$\begin{array}{cc} {\dot{X}}_{i}^{\mu }+\chi_{i}^{\left(1\right)}X_{i}^{\mu } & =\beta_{i}^{\left(1\right)}I^{\mu }+\ldots \;, \end{array}$$
(172c)$$\begin{array}{cc} {\dot{X}}_{i}^{\mu \nu }+\chi_{i}^{\left(2\right)}X_{i}^{\mu \nu } & =\beta_{i}^{\left(2\right)}\sigma^{\mu \nu }+\ldots \;, \end{array}$$
where now the ellipsis contains the above-mentioned nonlinear moments of the collision term and products between eigenmodes and gradients. In the above equations, we also defined
(173)$$\beta_{i}^{\left(0\right)}=\sum_{j=0\neq 1,2}^{N_{0}}{\left[{\left(\Omega^{\left(0\right)}\right)}^{-1}\right]}_{ij}\alpha_{j}^{\left(0\right)},\;\beta_{i}^{\left(1\right)}=\sum_{j=0\neq 1}^{N_{1}}{\left[{\left(\Omega^{\left(1\right)}\right)}^{-1}\right]}_{ij}\alpha_{j}^{\left(1\right)},\;\beta_{i}^{\left(2\right)}=2\sum_{j=0}^{N_{2}}{\left[{\left(\Omega^{\left(2\right)}\right)}^{-1}\right]}_{ij}\alpha_{j}^{\left(2\right)}.$$
Equations ([Disp-formula FD172a-entropy-26-00189]) tell us that at sufficiently late times, where the nonlinear terms in dissipative quantities are assumed to be small, the eigenmodes relax exponentially to their respective Navier–Stokes values. This happens on a timescale given by $1/\chi_{i}^{\left(\ell \right)}$. Hence, the ordering implied by Equation (Equation 169) is also a hierarchy of modes with respect to how fast they approach the Navier–Stokes regime. In particular, we identify the slowest mode for each *ℓ* as $X_{0}^{\mu_{1}\cdots \mu_{\ell }}$. At this point, one should emphasize that the tensors $X_{r}^{\mu_{1}\cdots \mu_{\ell }}$ for ℓ≥3 have been neglected because their asymptotic values are at least O(Kn2,KnRe−1) since it is not possible to construct a tensor with rank 3 or larger with single powers of $I^{\mu }$, $F^{\mu }$, or $\nabla_{\mu }u_{\nu }$.

In order to implement the truncation and derive the transient fluid-dynamical equations, we assume that only the modes $X_{0}$, $X_{0}^{\mu }$, and $X_{0}^{\mu \nu }$ (the slowest ones for tensor rank less or equal than 2) remain in the transient regime,
(174a)$$\begin{array}{cc} {\dot{X}}_{0}+\chi_{0}^{\left(\ell \right)}X_{0} & =\beta_{0}^{\left(0\right)}\theta +\ldots \;, \end{array}$$
(174b)$$\begin{array}{cc} {\dot{X}}_{0}^{\mu }+\chi_{0}^{\left(\ell \right)}X_{0}^{\mu } & =\beta_{0}^{\left(1\right)}I^{\mu }+\ldots \;, \end{array}$$
(174c)$$\begin{array}{cc} {\dot{X}}_{0}^{\mu \nu }+\chi_{0}^{\left(\ell \right)}X_{0}^{\mu \nu } & =\beta_{0}^{\left(2\right)}\sigma^{\mu \nu }+\ldots \;, \end{array}$$
whereas the remaining modes have already assumed their asymptotic Navier–Stokes values,
(175)$$X_{r>0}≃\frac{\beta_{r}^{\left(0\right)}}{\chi_{r}^{\left(0\right)}}\theta +\ldots \;,\;X_{r>0}^{\mu }≃\frac{\beta_{r}^{\left(1\right)}}{\chi_{r}^{\left(1\right)}}I^{\mu }+\ldots \;,\;X_{r>0}^{\mu \nu }≃\frac{\beta_{r}^{\left(2\right)}}{\chi_{r}^{\left(2\right)}}\sigma^{\mu \nu }+\ldots \;,$$
where we note that $X_{r>0}$, $X_{r>0}^{\mu }$, and $X_{r>0}^{\mu \nu }$ are of order O(Kn) since $\theta ,I^{\mu },\sigma^{\mu \nu }∼1/L$, while ${\left(\chi_{r}^{\left(\ell \right)}\right)}^{-1}∼\lambda_{\mathrm{mfp}}$. The above equations allow us to express irreducible moments not appearing in the conserved currents $N^{\mu }$, $T^{\mu \nu }$ in terms of moments that do appear. Indeed, we make use of the inverse of Equation (Equation 170),
(176)$$\rho_{r}^{\mu_{1}\cdots \mu_{\ell }}=\sum_{j=0}^{N_{\ell }}{\left(\Omega^{\left(\ell \right)}\right)}_{rn}X_{n}^{\mu_{1}\cdots \mu_{\ell }}\;,$$
which, when one inserts Equation (Equation 175) yields
(177a)$$\begin{array}{cc} \rho_{r} & ≃\Omega_{r0}^{\left(0\right)}X_{0}+\sum_{n=3}^{N_{0}}\Omega_{rn}^{\left(\ell \right)}\frac{\beta_{n}^{\left(0\right)}}{\chi_{n}^{\left(0\right)}}\theta =\Omega_{r0}^{\left(0\right)}X_{0}+O\left(\mathrm{Kn}\right)\;, \end{array}$$
(177b)$$\begin{array}{cc} \rho_{r}^{\mu } & ≃\Omega_{r0}^{\left(1\right)}X_{0}^{\mu }+\sum_{n=2}^{N_{1}}\Omega_{rn}^{\left(1\right)}\frac{\beta_{n}^{\left(1\right)}}{\chi_{n}^{\left(1\right)}}I^{\mu }=\Omega_{r0}^{\left(1\right)}X_{0}^{\mu }+O\left(\mathrm{Kn}\right)\;, \end{array}$$
(177c)$$\begin{array}{cc} \rho_{r}^{\mu \nu } & ≃\Omega_{r0}^{\left(2\right)}X_{0}^{\mu \nu }+\sum_{n=1}^{N_{2}}\Omega_{rn}^{\left(2\right)}\frac{\beta_{n}^{\left(2\right)}}{\chi_{n}^{\left(2\right)}}\sigma^{\mu \nu }=\Omega_{r0}^{\left(2\right)}X_{0}^{\mu \nu }+O\left(\mathrm{Kn}\right)\;. \end{array}$$
Since we consider Landau matching conditions in this section, all relevant fluid-dynamical fields can be related to the moments $\rho_{0}\equiv -3\Pi /m^{2}$, $\rho_{0}^{\mu }\equiv n^{\mu }$, and $\rho_{0}^{\mu \nu }\equiv \pi^{\mu \nu }$. Thus, in the particular case r=0 Equations ([Disp-formula FD177a-entropy-26-00189]) reduce to
(178a)$$\begin{array}{cc} -\frac{3}{m^{2}}\Pi & ≃X_{0}+\sum_{n=3}^{N_{0}}\Omega_{0n}^{\left(\ell \right)}\frac{\beta_{n}^{\left(0\right)}}{\chi_{n}^{\left(0\right)}}\theta =X_{0}+O\left(\mathrm{Kn}\right)\;, \end{array}$$
(178b)$$\begin{array}{cc} n^{\mu } & ≃X_{0}^{\mu }+O\left(\mathrm{Kn}\right)+\sum_{n=2}^{N_{1}}\Omega_{rn}^{\left(1\right)}\frac{\beta_{n}^{\left(1\right)}}{\chi_{n}^{\left(1\right)}}I^{\mu }=X_{0}^{\mu }+O\left(\mathrm{Kn}\right)\;, \end{array}$$
(178c)$$\begin{array}{cc} \pi^{\mu \nu } & ≃X_{0}^{\mu \nu }+\sum_{n=1}^{N_{2}}\Omega_{rn}^{\left(2\right)}\frac{\beta_{n}^{\left(2\right)}}{\chi_{n}^{\left(2\right)}}\sigma^{\mu \nu }=X_{0}^{\mu \nu }+O\left(\mathrm{Kn}\right)\;. \end{array}$$
Solving for $X_{0}$, $X_{0}^{\mu }$, and $X_{0}^{\mu \nu }$, and substituting the result into Equations ([Disp-formula FD177a-entropy-26-00189]), as well as making use of Equation (Equation 173), we derive
(179a)$$\begin{array}{cc} \frac{m^{2}}{3}\rho_{r} & ≃-\Omega_{r0}^{\left(0\right)}\Pi +\left(\zeta_{r}-\Omega_{r0}^{\left(0\right)}\zeta_{0}\right)\theta \;, \end{array}$$
(179b)$$\begin{array}{cc} \rho_{r}^{\mu } & ≃\Omega_{r0}^{\left(1\right)}n^{\mu }+\left(\varkappa_{r}-\Omega_{r0}^{\left(1\right)}\varkappa_{0}\right)I^{\mu }\;, \end{array}$$
(179c)$$\begin{array}{cc} \rho_{r}^{\mu \nu } & ≃\Omega_{r0}^{\left(2\right)}\pi^{\mu \nu }+\left(\eta_{r}-\Omega_{r0}^{\left(2\right)}\eta_{0}\right)\sigma^{\mu \nu }\;, \end{array}$$
(179d)ρrμ1⋯μℓ≃O(Kn2,Rei−1),ℓ≥3,
where the coefficients $\zeta_{r}$, $\varkappa_{r}$, and $\eta_{r}$ are defined as
(180)$$\zeta_{r}=\frac{m^{2}}{3}\sum_{n=0,\neq 1,2}^{N_{0}}\tau_{rn}^{\left(0\right)}\alpha_{n}^{\left(0\right)}\;,\;\varkappa_{r}=\sum_{n=0,\neq 1}^{N_{1}}\tau_{rn}^{\left(1\right)}\alpha_{n}^{\left(1\right)}\;,\;\eta_{r}=\sum_{n=0}^{N_{2}}\tau_{rn}^{\left(2\right)}\alpha_{n}^{\left(2\right)}\;.$$
Here, we have employed
(181)$$\tau_{rn}^{\left(\ell \right)}\equiv \sum_{m=0}^{N_{\ell }}\Omega_{rm}^{\left(\ell \right)}\frac{1}{\chi_{m}^{\left(\ell \right)}}{\left[\Omega^{\left(\ell \right)}\right]}_{mn}^{-1}\;,$$
i.e., the matrix $\tau^{\left(\ell \right)}\equiv {\left(L^{\left(\ell \right)}\right)}^{-1}$. Equations ([Disp-formula FD179a-entropy-26-00189]) affect the reduction of dynamical variables from the generic moments $\rho_{r}^{\mu_{1}\cdots \mu_{\ell }}$ to the fluid-dynamical ones. It is emphasized that these relations are valid up to first order in Knudsen number. Increasing $N_{0}$, $N_{1}$, and $N_{2}$ takes into account an increasing number of degrees of freedom for the computation of the transport coefficients, which should converge for sufficiently large values of $N_{0}$, $N_{1}$, and $N_{2}$, respectively.

At this point, we note that the expressions ([Disp-formula FD179a-entropy-26-00189]) are only valid for non-negative values of *r*. However, irreducible moments with negative *r* do appear in Equations ([Disp-formula FD150a-entropy-26-00189]), e.g., consider the term $\rho_{r-2}^{\mu \nu }\sigma_{\mu \nu }$ appearing in Equation ([Disp-formula FD150a-entropy-26-00189]), which features $\rho_{-2}^{\mu \nu }$ (for r=0) and $\rho_{-1}^{\mu \nu }$ (for r=1). The moments $\rho_{r<0}^{\mu_{1}\cdots \mu_{\ell }}$ can be related to the moments $\rho_{r\geq 0}^{\mu_{1}\cdots \mu_{\ell }}$ by making use of the fact that the moment expansion was made in terms of a complete basis. Thus, all moments that do not appear in the expansion must be related to the ones that do appear [42,157]. Indeed,
(182)$$\rho_{-r}^{\mu_{1}\cdots \mu_{\ell }}=\sum_{n=0}^{N_{\ell }}F_{rn}^{\left(\ell \right)}\rho_{n}^{\mu_{1}\cdots \mu_{\ell }}\;,$$
where we have defined the thermodynamic integral
(183)$$F_{rn}^{\left(\ell \right)}=\frac{\ell !}{(2\ell +1)!!}\int dK\;{\left({∆}_{\alpha \beta }k^{\alpha }k^{\beta }\right)}^{\ell }E_{k}^{-r}H_{kn}^{\left(\ell \right)}f_{0k}{\tilde{f}}_{0k}\;.$$
Thus, the expressions analogous to Equations ([Disp-formula FD179a-entropy-26-00189]) for negative moments are
(184a)$$\begin{array}{cc} \frac{m^{2}}{3}\rho_{-r} & ≃-\gamma_{r}^{\left(0\right)}\Pi +\sum_{n=0,\neq 1,2}^{N_{0}}F_{rn}^{\left(0\right)}\left(\zeta_{n}-\Omega_{n0}^{\left(0\right)}\zeta_{0}\right)\theta =-\gamma_{r}^{\left(0\right)}\Pi +\left(\Gamma_{r}^{\left(0\right)}-\gamma_{r}^{\left(0\right)}\right)\zeta_{0}\theta \;, \end{array}$$
(184b)$$\begin{array}{cc} \rho_{-r}^{\mu } & ≃\gamma_{r}^{\left(1\right)}n^{\mu }+\sum_{n=0,\neq 1}^{N_{1}}F_{rn}^{\left(1\right)}\left(\varkappa_{n}-\Omega_{n0}^{\left(1\right)}\varkappa_{0}\right)I^{\mu }=\gamma_{r}^{\left(1\right)}n^{\mu }+\left(\Gamma_{r}^{\left(1\right)}-\gamma_{r}^{\left(1\right)}\right)\varkappa_{0}I^{\mu }\;, \end{array}$$
(184c)$$\begin{array}{cc} \rho_{-r}^{\mu \nu } & ≃\gamma_{r}^{\left(2\right)}\pi^{\mu \nu }+\sum_{n=0}^{N_{2}}F_{rn}^{\left(2\right)}\left(\eta_{n}-\Omega_{n0}^{\left(2\right)}\eta_{0}\right)\sigma^{\mu \nu }=\gamma_{r}^{\left(2\right)}\pi^{\mu \nu }+\left(\Gamma_{r}^{\left(2\right)}-\gamma_{r}^{\left(2\right)}\right)\eta_{0}\sigma^{\mu \nu }\;, \end{array}$$
(184d)ρ−rμ1⋯μℓ≃O(Kn2,Rei−1),ℓ≥3,
where
(185a)$$\begin{array}{cccccc} \gamma_{r}^{\left(0\right)} & =\sum_{n=0,\neq 1,2}^{N_{0}}F_{rn}^{\left(0\right)}\Omega_{n0}^{\left(0\right)}\;,\;\gamma_{r}^{\left(1\right)} & & =\sum_{n=0,\neq 1}^{N_{1}}F_{rn}^{\left(1\right)}\Omega_{n0}^{\left(1\right)}\;,\;\gamma_{r}^{\left(2\right)} & & =\sum_{n=0}^{N_{2}}F_{rn}^{\left(2\right)}\Omega_{n0}^{\left(2\right)}\;, \end{array}$$
(185b)$$\begin{array}{cccccc} \Gamma_{r}^{\left(0\right)} & =\sum_{n=0,\neq 1,2}^{N_{0}}F_{rn}^{\left(0\right)}\frac{\zeta_{n}}{\zeta_{0}}\;,\;\Gamma_{r}^{\left(1\right)} & & =\sum_{n=0,\neq 1}^{N_{1}}F_{rn}^{\left(1\right)}\frac{\varkappa_{n}}{\varkappa_{0}}\;,\;\Gamma_{r}^{\left(2\right)} & & =\sum_{n=0}^{N_{2}}F_{rn}^{\left(2\right)}\frac{\eta_{n}}{\eta_{0}}\;. \end{array}$$

To finally derive the equations of motion complementing the conservation laws ([Disp-formula FD4a-entropy-26-00189]), it is convenient to first diagonalize the collision term by multiplying Equations ([Disp-formula FD150a-entropy-26-00189]) from the left with $\tau^{\left(\ell \right)}$, so that the linear component of the collision term is diagonal, i.e., $\sum_{j=0}\tau_{ij}^{\left(\ell \right)}L_{j-1}^{\mu_{1}\cdots \mu_{\ell }}=-\rho_{i}^{\mu_{1}\cdots \mu_{\ell }}$. One further step is to substitute all time- by space-like derivatives using Equation (Equation 257). Then, we obtain the equations of motion for the dissipative currents in the form ([Disp-formula FD23a-entropy-26-00189]), with the various terms on the right-hand sides given by Equations ([Disp-formula FD24a-entropy-26-00189]), ([Disp-formula FD25a-entropy-26-00189]), and ([Disp-formula FD26a-entropy-26-00189]), respectively. Again, we find that $\tau_{n\omega }\equiv \tau_{n}$ and $\tau_{\pi \omega }\equiv \tau_{\pi }$. We note that the terms of order O(Rei−2), i.e., R, $R^{\mu }$, and $R^{\mu \nu }$, respectively, arise from the nonlinear terms of the collisional moments. Equations ([Disp-formula FD24a-entropy-26-00189])–([Disp-formula FD26a-entropy-26-00189]) contain a plethora of transport coefficients. Indeed, the first-order coefficients related to the bulk viscous pressure, particle diffusion, and shear-stress tensor and their relaxation times are, respectively [42,48],
(186a)$$\begin{array}{cc} \; & \zeta \equiv \zeta_{0}=\frac{m^{2}}{3}\sum_{r=0,\neq 1,2}^{N_{0}}\tau_{0r}^{\left(0\right)}\alpha_{r}^{\left(0\right)}\;, \end{array}$$
(186b)$$\begin{array}{cc} \; & \varkappa =\varkappa_{0}=\sum_{r=0,\neq 1}^{N_{1}}\tau_{0r}^{\left(1\right)}\alpha_{r}^{\left(1\right)}\;, \end{array}$$
(186c)$$\begin{array}{cc} \; & \eta \equiv \eta_{0}=\sum_{r=0}^{N_{2}}\tau_{0r}^{\left(2\right)}\alpha_{r}^{\left(2\right)}\;, \end{array}$$
(186d)$$\begin{array}{cc} \; & \tau_{\Pi }\equiv \frac{1}{\chi_{0}^{\left(0\right)}}\;,\;\tau_{n}\equiv \frac{1}{\chi_{0}^{\left(1\right)}}\;,\;\tau_{\pi }\equiv \frac{1}{\chi_{0}^{\left(2\right)}}\;. \end{array}$$
The transport coefficients associated with J (see Equation ([Disp-formula FD24a-entropy-26-00189])), which are O(KnRe−1) in the equation of motion for the bulk viscous pressure, read [42,48],
(187a)$$\begin{array}{cc} \ell_{\Pi n} & =-\frac{m^{2}}{3}\left(\gamma_{1}^{\left(1\right)}\tau_{00}^{\left(0\right)}-\sum_{r=0,\neq 1,2}^{N_{0}}\tau_{0r}^{\left(0\right)}\frac{G_{3r}}{D_{20}}+\sum_{r=0}^{N_{0}-3}\tau_{0,r+3}^{\left(0\right)}\Omega_{r+2,0}^{\left(1\right)}\right)\;, \end{array}$$
(187b)$$\begin{array}{cc} \tau_{\Pi n} & =\frac{m^{2}}{3\left(\varepsilon_{0}+p_{0}\right)}\left[\tau_{00}^{\left(0\right)}\frac{\partial \gamma_{1}^{\left(1\right)}}{\partial \ln\beta_{0}}-\sum_{r=0,\neq 1,2}^{N_{0}}\tau_{0r}^{\left(0\right)}\frac{G_{3r}}{D_{20}}+\sum_{r=0}^{N_{0}-3}\tau_{0,r+3}^{\left(0\right)}\left(r+3+\frac{\partial }{\partial \ln\beta_{0}}\right)\Omega_{r+2,0}^{\left(1\right)}\right]\;, \end{array}$$
(187c)$$\begin{array}{cc} \delta_{\Pi \Pi } & =\frac{1}{3}\tau_{00}^{\left(0\right)}\left(2+m^{2}\gamma_{2}^{\left(0\right)}\right)-\frac{m^{2}}{3}\sum_{r=0,\neq 1,2}^{N_{0}}\tau_{0r}^{\left(0\right)}\frac{G_{2r}}{D_{20}}\\ \; & \;\;+\frac{1}{3}\sum_{r=0}^{N_{0}-3}\left(r+5\right)\tau_{0,r+3}^{\left(0\right)}\Omega_{r+3,0}^{\left(0\right)}-\frac{m^{2}}{3}\sum_{r=0}^{N_{0}-5}\left(r+4\right)\tau_{0,r+5}^{\left(0\right)}\Omega_{r+3,0}^{\left(0\right)}\\ \; & \;\;+\sum_{r=3}^{N_{0}}\tau_{0r}^{\left(0\right)}\left[\frac{\left(\varepsilon_{0}+p_{0}\right)J_{10}-n_{0}J_{20}}{D_{20}}\frac{\partial }{\partial \alpha_{0}}+\frac{\left(\varepsilon_{0}+p_{0}\right)J_{20}-n_{0}J_{30}}{D_{20}}\frac{\partial }{\partial \beta_{0}}\right]\Omega_{r0}^{\left(0\right)}\;, \end{array}$$
(187d)$$\begin{array}{cc} \lambda_{\Pi n} & =-\frac{m^{2}}{3}\left[\tau_{00}^{\left(0\right)}\left(\frac{\partial }{\partial \alpha_{0}}+\frac{n_{0}}{\varepsilon_{0}+p_{0}}\frac{\partial }{\partial \beta_{0}}\right)\gamma_{1}^{\left(1\right)}+\sum_{r=0}^{N_{0}-3}\tau_{0,r+3}^{\left(0\right)}\left(\frac{\partial }{\partial \alpha_{0}}+\frac{n_{0}}{\varepsilon_{0}+p_{0}}\frac{\partial }{\partial \beta_{0}}\right)\Omega_{r+2,0}^{\left(1\right)}\right]\;, \end{array}$$
(187e)$$\begin{array}{cc} \lambda_{\Pi \pi } & =\frac{m^{2}}{3}\left[\gamma_{2}^{\left(2\right)}\tau_{00}^{\left(0\right)}-\sum_{r=0,\neq 1,2}^{N_{0}}\tau_{0r}^{\left(0\right)}\frac{G_{2r}}{D_{20}}-\sum_{r=0}^{N_{0}-3}\left(r+2\right)\tau_{0,r+3}^{\left(0\right)}\Omega_{r+1,0}^{\left(2\right)}\right]\;. \end{array}$$
The transport coefficients associated with $J^{\mu }$ (see Equation ([Disp-formula FD24b-entropy-26-00189])), which are O(KnRe−1) in the equation of motion for the particle diffusion current, read
(188a)$$\begin{array}{cc} \delta_{nn} & =\tau_{00}^{\left(1\right)}\left(1+\frac{m^{2}}{3}\gamma_{2}^{\left(1\right)}\right)-\frac{m^{2}}{3}\sum_{r=0}^{N_{1}-2}\left(r+1\right)\tau_{0,r+2}^{\left(1\right)}\Omega_{r0}^{\left(1\right)}\\ \; & +\sum_{r=2}^{N_{1}}\tau_{0r}^{\left(1\right)}\left[1+\frac{r}{3}+\frac{\left(\varepsilon_{0}+p_{0}\right)J_{20}-n_{0}J_{30}}{D_{20}}\frac{\partial }{\partial \alpha_{0}}+\frac{\left(\varepsilon_{0}+p_{0}\right)J_{10}-n_{0}J_{20}}{D_{20}}\frac{\partial }{\partial \beta_{0}}\right]\Omega_{r0}^{\left(1\right)}\;, \end{array}$$
(188b)$$\begin{array}{cc} \ell_{n\Pi } & =\left(\frac{n_{0}}{\varepsilon_{0}+p_{0}}-\gamma_{1}^{\left(0\right)}\right)\tau_{00}^{\left(1\right)}+\sum_{r=0}^{N_{1}-2}\tau_{0,r+2}^{\left(1\right)}\left(\frac{\beta_{0}J_{r+4,1}}{\varepsilon_{0}+p_{0}}+\frac{\Omega_{r+3,0}^{\left(0\right)}}{m^{2}}\right)-\sum_{r=0}^{N_{1}-4}\tau_{0,r+4}^{\left(1\right)}\Omega_{r+3,0}^{\left(0\right)}\;, \end{array}$$
(188c)$$\begin{array}{cc} \tau_{n\Pi } & =\frac{1}{\varepsilon_{0}+p_{0}}\left\{\tau_{00}^{\left(1\right)}\left(\frac{n_{0}}{\varepsilon_{0}+p_{0}}-\frac{\partial \gamma_{1}^{\left(0\right)}}{\partial \ln\beta_{0}}\right)-\sum_{r=0}^{N_{1}-4}\tau_{0,r+4}^{\left(1\right)}\left(r+4+\frac{\partial }{\partial \ln\beta_{0}}\right)\Omega_{r+3,0}^{\left(0\right)}\right.\\ \; & \;+\left.\sum_{r=0}^{N_{1}-2}\tau_{0,r+2}^{\left(1\right)}\left[\frac{\beta_{0}J_{r+4,1}}{\varepsilon_{0}+p_{0}}+\frac{1}{m^{2}}\left(r+5+\frac{\partial }{\partial \ln\beta_{0}}\right)\Omega_{r+3,0}^{\left(0\right)}\right]\right\}\;, \end{array}$$
(188d)$$\begin{array}{cc} \ell_{n\pi } & =\tau_{00}^{\left(1\right)}\left(\frac{n_{0}}{\varepsilon_{0}+p_{0}}-\gamma_{1}^{\left(2\right)}\right)+\sum_{r=0}^{N_{1}-2}\tau_{0,r+2}^{\left(1\right)}\left(\frac{\beta_{0}J_{r+4,1}}{\varepsilon_{0}+p_{0}}-\Omega_{r+1,0}^{\left(2\right)}\right)\;, \end{array}$$
(188e)$$\begin{array}{cc} \tau_{n\pi } & =\frac{1}{\varepsilon_{0}+p_{0}}\left\{\tau_{00}^{\left(1\right)}\left(\frac{n_{0}}{\varepsilon_{0}+p_{0}}-\frac{\partial \gamma_{1}^{\left(2\right)}}{\partial \ln\beta_{0}}\right)\right.\\ \; & \;+\left.\sum_{r=0}^{N_{1}-2}\tau_{0,r+2}^{\left(1\right)}\left[\frac{\beta_{0}J_{r+4,1}}{\varepsilon_{0}+p_{0}}-\left(r+2+\frac{\partial }{\partial \ln\beta_{0}}\right)\Omega_{r+1,0}^{\left(2\right)}\right]\right\}\;, \end{array}$$
(188f)$$\begin{array}{cc} \lambda_{nn} & =\tau_{00}^{\left(1\right)}\frac{1}{5}\left(3+2m^{2}\gamma_{2}^{\left(1\right)}\right)-\frac{2}{5}m^{2}\sum_{r=0,r\neq 1}^{N_{1}-2}\left(r+1\right)\tau_{0,r+2}^{\left(1\right)}\Omega_{r0}^{\left(1\right)}+\frac{1}{5}\sum_{r=2}^{N_{1}}\left(2r+3\right)\tau_{0r}^{\left(1\right)}\Omega_{r0}^{\left(1\right)}\;, \end{array}$$
(188g)$$\begin{array}{cc} \lambda_{n\Pi } & =\tau_{00}^{\left(1\right)}\left(\frac{\partial }{\partial \alpha_{0}}+\frac{n_{0}}{\varepsilon_{0}+p_{0}}\frac{\partial }{\partial \beta_{0}}\right)\gamma_{1}^{\left(0\right)}-\frac{1}{m^{2}}\sum_{r=0}^{N_{1}-2}\tau_{0,r+2}^{\left(1\right)}\left(\frac{\partial }{\partial \alpha_{0}}+\frac{n_{0}}{\varepsilon_{0}+p_{0}}\frac{\partial }{\partial \beta_{0}}\right)\Omega_{r+3,0}^{\left(0\right)}\\ \; & +\sum_{r=0}^{N_{1}-4}\tau_{0,r+4}^{\left(1\right)}\left(\frac{\partial }{\partial \alpha_{0}}+\frac{n_{0}}{\varepsilon_{0}+p_{0}}\frac{\partial }{\partial \beta_{0}}\right)\Omega_{r+3,0}^{\left(0\right)}\;, \end{array}$$
(188h)$$\begin{array}{cc} \lambda_{n\pi } & =\tau_{00}^{\left(1\right)}\left(\frac{\partial }{\partial \alpha_{0}}+\frac{n_{0}}{\varepsilon_{0}+p_{0}}\frac{\partial }{\partial \beta_{0}}\right)\gamma_{1}^{\left(2\right)}+\sum_{r=0}^{N_{1}-2}\tau_{0,r+2}^{\left(1\right)}\left(\frac{\partial }{\partial \alpha_{0}}+\frac{n_{0}}{\varepsilon_{0}+p_{0}}\frac{\partial }{\partial \beta_{0}}\right)\Omega_{r+1,0}^{\left(2\right)}\;. \end{array}$$
Finally, the transport coefficients associated with $J^{\mu \nu }$ (see Equation ([Disp-formula FD24c-entropy-26-00189])), which are O(KnRe−1), in the shear-stress tensor equation of motion are
(189a)$$\begin{array}{cc} \delta_{\pi \pi } & =\frac{1}{3}m^{2}\gamma_{2}^{\left(2\right)}\tau_{00}^{\left(2\right)}+\frac{1}{3}\sum_{r=0}^{N_{2}}\left(r+4\right)\tau_{0r}^{\left(2\right)}\Omega_{r0}^{\left(2\right)}-\frac{1}{3}m^{2}\sum_{r=0}^{N_{2}-2}\left(r+1\right)\tau_{0,r+2}^{\left(2\right)}\Omega_{r0}^{\left(2\right)}\\ \; & \;\;+\sum_{r=0}^{N_{2}}\tau_{0r}^{\left(2\right)}\left[\frac{\left(\varepsilon_{0}+p_{0}\right)J_{20}-n_{0}J_{30}}{D_{20}}\frac{\partial }{\partial \alpha_{0}}+\frac{\left(\varepsilon_{0}+p_{0}\right)J_{10}-n_{0}J_{20}}{D_{20}}\frac{\partial }{\partial \beta_{0}}\right]\Omega_{r0}^{\left(2\right)}\;, \end{array}$$
(189b)$$\begin{array}{cc} \lambda_{\pi \pi } & =\frac{4}{7}m^{2}\gamma_{2}^{\left(2\right)}\tau_{00}^{\left(2\right)}+\frac{2}{7}\sum_{r=0}^{N_{2}}\left(2r+5\right)\tau_{0r}^{\left(2\right)}\Omega_{r0}^{\left(2\right)}-\frac{4}{7}m^{2}\sum_{r=0}^{N_{2}-2}\left(r+1\right)\tau_{0,r+2}^{\left(2\right)}\Omega_{r0}^{\left(2\right)}\;, \end{array}$$
(189c)$$\begin{array}{cc} \lambda_{\pi \Pi } & =\tau_{00}^{\left(2\right)}\frac{2}{5}\left(3+m^{2}\gamma_{2}^{\left(0\right)}\right)-\frac{2}{5m^{2}}\sum_{r=0}^{N_{2}-1}\left(r+5\right)\tau_{0,r+1}^{\left(2\right)}\Omega_{r+3,0}^{\left(0\right)}\\ \; & \;\;+\frac{2}{5}\sum_{r=3}^{N_{2}}\left(2r+3\right)\tau_{0r}^{\left(2\right)}\Omega_{r0}^{\left(0\right)}-\frac{2}{5}m^{2}\sum_{r=0,\neq 1,2}^{N_{2}-2}\left(r+1\right)\tau_{0,r+2}^{\left(2\right)}\Omega_{r0}^{\left(0\right)}\;, \end{array}$$
(189d)$$\begin{array}{cc} \;\tau_{\pi n} & =\frac{2}{5(\varepsilon_{0}+p_{0})}\left[-m^{2}\tau_{00}^{\left(2\right)}\frac{\partial \gamma_{1}^{\left(1\right)}}{\partial \ln\beta_{0}}+\sum_{r=0}^{N_{2}-1}\tau_{0,r+1}^{\left(2\right)}\left(r+6+\frac{\partial }{\partial \ln\beta_{0}}\right)\Omega_{r+2,0}^{\left(1\right)}\right.\\ \; & \;\left.-m^{2}\sum_{r=0,\neq 1}^{N_{2}-1}\left(r+1\right)\tau_{0,r+1}^{\left(2\right)}\Omega_{r0}^{\left(1\right)}-m^{2}\sum_{r=0}^{N_{2}-3}\tau_{0,r+3}^{\left(2\right)}\frac{\partial \Omega_{r+2,0}^{\left(1\right)}}{\partial \ln\beta_{0}}\right]\;, \end{array}$$
(189e)$$\begin{array}{cc} \ell_{\pi n} & =-\frac{2}{5}m^{2}\gamma_{1}^{\left(1\right)}\tau_{00}^{\left(2\right)}+\frac{2}{5}\sum_{r=0}^{N_{2}-1}\tau_{0,r+1}^{\left(2\right)}\Omega_{r+2,0}^{\left(1\right)}-\frac{2}{5}m^{2}\sum_{r=0,\neq 1}^{N_{2}-1}\tau_{0,r+1}^{\left(2\right)}\Omega_{r0}^{\left(1\right)}\;, \end{array}$$
(189f)$$\begin{array}{cc} \lambda_{\pi n} & =-\frac{2}{5}m^{2}\tau_{00}^{\left(2\right)}\left(\frac{\partial }{\partial \alpha_{0}}+\frac{n_{0}}{\varepsilon_{0}+p_{0}}\frac{\partial }{\partial \beta_{0}}\right)\gamma_{1}^{\left(1\right)}+\frac{2}{5}\sum_{r=0}^{N_{2}-1}\tau_{0,r+1}^{\left(2\right)}\left(\frac{\partial }{\partial \alpha_{0}}+\frac{n_{0}}{\varepsilon_{0}+p_{0}}\frac{\partial }{\partial \beta_{0}}\right)\Omega_{r+2,0}^{\left(1\right)}\\ \; & \;\;-\frac{2}{5}m^{2}\sum_{r=0}^{N_{2}-3}\tau_{0,r+3}^{\left(2\right)}\left(\frac{\partial }{\partial \alpha_{0}}+\frac{n_{0}}{\varepsilon_{0}+p_{0}}\frac{\partial }{\partial \beta_{0}}\right)\Omega_{r+2,0}^{\left(1\right)}\;. \end{array}$$
With increasing values of $N_{0}$, $N_{1}$, and $N_{2}$, an increasing number of non-hydrodynamic moments contribute to the transport coefficients. These should converge for sufficiently large values of $N_{0}$, $N_{1}$, and $N_{2}$. The simplest truncation, $N_{0}=2$, $N_{1}=1$ and $N_{2}=0$, implies that the distribution function is expanded in terms of 14 moments. Thus this is called the 14-moment approximation. Subsequent truncations involve 23, 32, 41, …, $\left(5N_{2}+3N_{1}+N_{0}+9\right)$ moments and receive the corresponding approximation name. In [42], tables containing results for a gas with a constant cross-section for binary elastic scattering are displayed for 14, 23, 32, and 41 moments. In Appendix C, we also display the $O({\mathrm{Kn}}^{2})$ transport coefficients associated with the K-terms in Equations ([Disp-formula FD25a-entropy-26-00189]). Moreover, there, expressions for the transport coefficients in the R-terms, which are O(Rei−1Rej−1) and arise from quadratic terms in the nonlinear collisional moments $N_{r-1}^{\mu_{1}\cdots \mu_{\ell }}$ (see Equation (Equation 152)), are shown as computed in [157].

#### 5.5.2. IReD Approach

While the DNMR method consists of diagonalizing the collision kernel and subsequently approximating all but the slowest eigenmodes by their respective Navier–Stokes values, the so-called *order-of-magnitude* approximation [149,160] or *Inverse-Reynolds Dominance* (IReD) approach [159] takes a different route by not assuming an ordering of the microscopic timescales.

Assuming that the Knudsen number and the inverse Reynolds numbers are of the same order, i.e., Kn∼Rei−1, with i∈{Π,n,π}, the moment equations can be written as
(190)ρr=3m2ζrθ+O(KnRe−1),ρrμ=ϰrIμ+O(KnRe−1),ρrμν=2ηrσμν+O(KnRe−1).

Here, Re−1 denotes any of the possible inverse Reynolds numbers. Since this relation holds for any *r*, we can use it to express a moment of tensor-rank ℓ≤2 and energy-rank *r* in terms of an arbitrary moment of the same tensor-, but different energy-rank *n*:(191)ρr=ζrζnρn+O(KnRe−1),ρrμ=ϰrϰnρnμ+O(KnRe−1),ρrμν=ηrηnρnμν+O(KnRe−1).
Choosing n=0, we find
(192)ρr=−3m2Cr(0)Π+O(KnRe−1),ρrμ=Cr(1)nμ+O(KnRe−1),ρrμν=Cr(2)πμν+O(KnRe−1),
where we defined
(193)$$C_{r}^{\left(0\right)}\equiv \frac{\zeta_{r}}{\zeta_{0}}\;,\;C_{r}^{\left(1\right)}\equiv \frac{\varkappa_{r}}{\varkappa_{0}}\;,\;C_{r}^{\left(2\right)}\equiv \frac{\eta_{r}}{\eta_{0}}\;.$$
Equation (Equation 192) denote the asymptotic matching scheme for moments with energy-rank r>0 in the IReD approach and have to be contrasted with the DNMR one, cf. Equations ([Disp-formula FD179a-entropy-26-00189]). Two main differences can be noticed: first, the DNMR-type asymptotic matching depends on the matrices $\Omega^{\left(\ell \right)}$, which diagonalize the linearized collision term, while in the IReD matching only the first-order transport coefficients $\zeta_{r},\;\varkappa_{r}$, and $\eta_{r}$ appear. Second, in Equations ([Disp-formula FD179a-entropy-26-00189]), the right-hand sides contain terms that are linear in inverse Reynolds numbers as well as terms linear in the Knudsen number. In contrast, all quantities in Equation (Equation 192) are of first order in inverse Reynolds numbers only.

The fact that terms of first order in the Knudsen number are absent in Equation (Equation 192) has profound consequences on the structure of the resulting fluid-dynamical equations. Inserting the asymptotic matching scheme (Equation 192) into the moment Equations ([Disp-formula FD150a-entropy-26-00189]) and neglecting terms of order O(Kn2Re−1) and higher, we obtain fluid-dynamical equations that are formally identical to Equations ([Disp-formula FD23a-entropy-26-00189]), with the important difference that all terms of second order in the Knudsen number vanish identically, i.e., $K=K^{\mu }=K^{\mu \nu }=0$. The reason for this lies in the fact that the origin of the K-terms in the DNMR approach are derivatives in Equations ([Disp-formula FD150a-entropy-26-00189]), which act on the terms of first order in the Knudsen number appearing in Equations ([Disp-formula FD179a-entropy-26-00189]), which are absent from Equation (Equation 192). In addition to this, the transport coefficients appearing in the fluid-dynamical equations differ from the ones obtained in the DNMR approach as soon as a truncation higher than $N_{0}=2,\;N_{1}=1,\;N_{2}=0$ is chosen [42,159,160,161].

In [159], it is shown that, as long as terms of third order in Knudsen and inverse Reynolds numbers are neglected consistently, the two approaches are perturbatively equivalent since Equation (Equation 190) can be used to move terms from $J^{\mu_{1}\cdots \mu_{\ell }}$ to $K^{\mu_{1}\cdots \mu_{\ell }}$ and vice versa. In order to see how this happens, consider, for instance, the term $\zeta_{2}\sigma^{\mu \nu }\sigma_{\mu \nu }$ that appears in K, Equation ([Disp-formula FD25a-entropy-26-00189]). Using Equation (Equation 190) and setting $\eta \equiv \eta_{0}$, we find
(194)ζ˜2σμνσμν=ζ˜22ησμνπμν+O(Kn2Re−1),
where the right-hand side can now be included in the respective term $∼\lambda_{\Pi \pi }\pi^{\mu \nu }\sigma_{\mu \nu }$ in J, Equation ([Disp-formula FD24a-entropy-26-00189]). This procedure is in fact the same as in resummed BRSSS theory, although here not all factors of $\sigma^{\mu \nu }$ are replaced, cf. Section 3.5. Thus, one can arrive at the fluid-dynamical equations of the IReD approach either by directly employing the asymptotic matching (Equation 192) or by first employing the DNMR approach and then performing steps similar to Equation (Equation 194) to shift the K-terms into the other second-order transport coefficients. In short, the procedure can be graphically described as
$$\begin{array}{cccccc} \; & \left(\mathrm{DNMR}\right) & \; & & \; & \left(\mathrm{IReD}\right) \end{array}$$
(195a)$$\begin{array}{cccccc} \; & \Omega_{r0}^{\left(\ell \right)}\; & ⟶ & \; & & C_{r}^{\left(\ell \right)}\;, \end{array}$$
(195b)$$\begin{array}{cccccc} \; & \gamma_{r}^{\left(\ell \right)}\; & ⟶ & \; & & C_{-r}^{\left(\ell \right)}\;, \end{array}$$
(195c)$$\begin{array}{cccccc} \; & K^{\mu_{1}\cdots \mu_{\ell }}\; & ⟶ & \; & & 0\;. \end{array}$$
In practical applications, the IReD approach has several advantages compared to the standard DNMR prescription. Since the terms of second order in the Knudsen number are expected to destroy the hyperbolic character of the equations of motion [42], they are usually discarded in a numerical implementation. The IReD approach resums the effect of these contributions into the terms $J^{\mu_{1}\cdots \mu_{\ell }}$, which do not necessarily break causality. In fact, as shown for a gas of massless particles in [96], the transport coefficients computed via the IReD prescription render the system symmetric hyperbolic in the linear regime. Furthermore, as demonstrated in [160], the approach readily applies to systems with multiple conserved charges. Recently, in [162], the performance of the fluid-dynamical equations obtained from the DNMR approach (where the terms of second order in the Knudsen number have been neglected) and the IReD approach has been compared in the context of an exactly solvable microscopic model of coupled relaxation-type equations. In this simplified setup, it could be seen that omitting the terms of order $O({\mathrm{Kn}}^{2})$ can significantly impact the accuracy of the DNMR prescription, while the IReD approach turned out to yield a good approximation of the exact dynamics, even if the slowest microscopic mode did not dominate the dynamics of the system.

Lastly, a remark has to be made on the interpretation of the relaxation times in the two approaches. While in the DNMR prescription the dissipative fluid-dynamical degrees of freedom relax by construction on a timescale given by the largest eigenvalue of the inverse collision matrix, the relaxation time in the IReD approach does not have this microscopic interpretation. The reason for this lies again in the asymptotic matching: The relations are
(196)$$\tau_{\Pi }={\tilde{\tau }}_{\Pi }+\frac{{\tilde{\zeta }}_{1}}{\zeta }\;,\;\tau_{n}={\tilde{\tau }}_{n}+\frac{{\tilde{\varkappa }}_{5}}{2\varkappa }\;,\;\tau_{\pi }={\tilde{\tau }}_{\pi }+\frac{{\tilde{\eta }}_{1}}{2\eta }\;,$$
where we denoted the relaxation times in the DNMR approach with a tilde. Thus, the relaxation times in the IReD approach contain additional contributions from the K-terms. More specifically, while in the DNMR method the relaxation time is given only by the largest eigenvalue of the respective inverse collision matrix, in the IReD method, it consists of a weighted sum of all eigenvalues, with the weights quantifying how relevant a certain eigenmode is to the fluid-dynamical quantity under consideration, cf. [162].

## 6. Zubarev’s Approach

Zubarev’s approach, also known as the method of the non-equilibrium statistical operator [163,164,165,166,167], to derive dissipative fluid dynamics is based on quantum field theory as the underlying microscopic theory. This method generalizes the Gibbs canonical ensemble to non-equilibrium states, i.e., the statistical operator is promoted to a non-local functional of the thermodynamic parameters and their space-time derivatives. Assuming that the thermodynamic parameters vary sufficiently smoothly over the correlation lengths characterizing the system, the statistical operator is then expanded in a series in gradients of these parameters. The equations of motion for the dissipative currents emerge after statistical averaging of the relevant quantum operators. An advantage of Zubarev’s formalism is that its applicability is, in principle, not limited to the weak-coupling regime: the transport coefficients of the system are obtained in the form of Kubo-type relations, i.e., they are related to certain correlation functions of the underlying field theory, which are valid also in the strong-coupling limit.

In recent years, there has been a renewed interest in applications of Zubarev’s approach to relativistic fluid dynamics. Novel developments include a formulation of anisotropic magnetohydrodynamics in strong magnetic fields [168], reformulations suitable for applications in heavy-ion collisions [169], a second-order expansion of the statistical operator, which leads to second-order fluid dynamics [170], and fluid dynamics with anomalies [171,172]. In the following, the discussion largely follows [173].

Let us assume a foliation of space-time in the form of hypersurfaces Σ at constant times, t=const., with time-like normal vectors $d\Sigma_{\mu }=g_{\mu 0}d^{3}r^{\prime }$. The starting point of Zubarev’s approach is the statistical operator ${\hat{\rho }}_{l}\left(t\right)$ of a local-equilibrium state,
(197a)$$\begin{array}{cc} {\hat{\rho }}_{l}\left(t\right) & =\exp\left\{\Omega_{l}\left(t\right)-\int_{\Sigma }d\Sigma_{\mu }\left[\beta_{\nu }\left(t,r^{\prime }\right){\hat{T}}^{\mu \nu }\left(t,r^{\prime }\right)-\alpha_{0}\left(t,r^{\prime }\right){\hat{N}}^{\mu }\left(t,r^{\prime }\right)\right]\right\}\;, \end{array}$$
(197b)$$\begin{array}{cc} e^{-\Omega_{l}\left(t\right)} & =\mathrm{Tr}\exp\left\{-\int_{\Sigma }d\Sigma_{\mu }\left[\beta_{\nu }\left(t,r^{\prime }\right){\hat{T}}^{\mu \nu }\left(t,r^{\prime }\right)-\alpha_{0}\left(t,r^{\prime }\right){\hat{N}}^{\mu }\left(t,r^{\prime }\right)\right]\right\}\;, \end{array}$$
where ${\hat{T}}^{\mu \nu }$ and ${\hat{N}}^{\mu }$ are the operators of the energy-momentum tensor and particle 4-current and
(198)$$\beta_{\nu }\left(t,r^{\prime }\right)=\beta_{0}\left(t,r^{\prime }\right)u_{\nu }\left(t,r^{\prime }\right)\;,\;\alpha_{0}\left(t,r^{\prime }\right)=\beta_{0}\left(t,r^{\prime }\right)\mu \left(t,r^{\prime }\right)\;.$$
Local equilibrium implies that each fluid element can be ascribed local values of the hydrodynamic parameters $\alpha_{0}\left(t,r^{\prime }\right)$ and $\beta_{0}\left(t,r^{\prime }\right)$, as well as a fluid 4-velocity $u^{\mu }\left(t,r^{\prime }\right)$. These are assumed to vary slowly in space and time.

Next, we define the operators of energy and particle-number densities in the comoving frame via $\hat{\varepsilon }=u_{\mu }u_{\nu }{\hat{T}}^{\mu \nu }$ and $\hat{n}=u\cdot \hat{N}$. The local values of the Lorentz-invariant thermodynamic parameters $\alpha_{0}$ and $\beta_{0}$ at any *given* point $x^{\mu }\equiv \left(t,r\right)$ on Σ are then fixed by given average values of the operators $\hat{\epsilon }$ and $\hat{n}$ via the following matching conditions [163,164,165]
(199)$$\varepsilon_{0}\left(x\right)\equiv \langle \hat{\varepsilon }\left(x\right)\rangle ={\langle \hat{\varepsilon }\left(x\right)\rangle }_{l}\;,\;n_{0}\left(x\right)\equiv \langle \hat{n}\left(x\right)\rangle ={\langle \hat{n}\left(x\right)\rangle }_{l}\;,$$
where we introduced the notation
(200)$$\langle \hat{F}\left(x\right)\rangle =\mathrm{Tr}\left[\hat{\rho }\left(t\right)\hat{F}\left(x\right)\right]\;,\;{\langle \hat{F}\left(x\right)\rangle }_{l}=\mathrm{Tr}\left[{\hat{\rho }}_{l}\left(t\right)\hat{F}\left(x\right)\right]\;.$$
Note that the conditions (Equation 199) define the energy density $\varepsilon_{0}\left(x\right)$ and particle-number density $n_{0}\left(x\right)$ as *non-local functionals* of $\alpha_{0}$ and $\beta_{0}$, $\varepsilon_{0}\left[\alpha_{0},\beta_{0}\right]$, $n_{0}\left[\alpha_{0},\beta_{0}\right]$, the reason being that one integrates over *all* values of $\alpha_{0}\left(t,r^{\prime }\right)$ and $\beta_{0}\left(t,r^{\prime }\right)$ on the *whole* space-time hypersurface in Equations (197a,b), while $x^{\mu }=\left(t,r\right)$ is a *given* point on that surface in Equation (Equation 199). Inverting this functional dependence defines temperature and chemical potential as *non-local functionals* of $\varepsilon_{0}$ and $n_{0}$, $\alpha_{0}\left[\varepsilon_{0},n_{0}\right]$, $\beta_{0}\left[\varepsilon_{0},n_{0}\right]$. However, in the fluid-dynamical description, one needs to define thermodynamical parameters as *local* functions of the energy and particle-number densities, just as in global thermodynamical equilibrium. This can be done by assuming that all fluid elements where local equilibrium is already established are statistically independent of each other [174]. In other words, the local-equilibrium values ${\langle \hat{\varepsilon }\rangle }_{l}$ and ${\langle \hat{n}\rangle }_{l}$ in Equation (Equation 199) should be evaluated formally at *constant values* of $\alpha_{0}$ and $\beta_{0}$, which are then determined by matching ${\langle \hat{\varepsilon }\rangle }_{l}$ and ${\langle \hat{n}\rangle }_{l}$ to the real values $\langle \hat{\varepsilon }\rangle$ and $\langle \hat{n}\rangle$ of these quantities at a given point $x^{\mu }$ in space-time. This assigns a *fictitious local-equilibrium state* to any given point, such that it reproduces the local values of the energy and particle-number densities.

While the local-equilibrium statistical operator ([Disp-formula FD197a-entropy-26-00189]) reproduces the local values of the macroscopic observables ε and *n*, it does not satisfy the Liouville equation (in the Heisenberg picture)
(201)$$\frac{d\hat{\rho }\left(t\right)}{dt}=0\;,$$
and, therefore, cannot describe non-equilibrium processes.

As shown in [173], by averaging over initial states, one can derive a statistical operator which fulfills the Liouville Equation (Equation 201) and incorporates irreversible thermodynamic processes,
(202)$$\hat{\rho }\left(t\right)=Q^{-1}e^{-\hat{A}+\hat{B}}\;,\;Q=\mathrm{Tr}\;e^{-\hat{A}+\hat{B}}\;,$$
with
(203a)$$\begin{array}{cc} \hat{A}\left(t\right) & \equiv \int d^{3}r^{\prime }\left[\beta_{\nu }\left(t,r^{\prime }\right){\hat{T}}^{0\nu }\left(t,r^{\prime }\right)-\alpha_{0}\left(t,r^{\prime }\right){\hat{N}}^{0}\left(t,r^{\prime }\right)\right]\;, \end{array}$$
(203b)$$\begin{array}{cc} \hat{B}\left(t\right) & \equiv \int d^{3}r^{\prime }\;\int_{-\infty }^{t}dt_{1}\;e^{\delta (t_{1}-t)}\hat{C}\left(t_{1},r^{\prime }\right)\;, \end{array}$$
(203c)$$\begin{array}{cc} \hat{C}\left(t,r\right) & \equiv {\hat{T}}^{\mu \nu }\left(t,r\right)\partial_{\mu }\beta_{\nu }\left(t,r\right)-{\hat{N}}^{\mu }\left(t,r\right)\partial_{\mu }\alpha_{0}\left(t,r\right)\;. \end{array}$$
The term $\hat{A}\left(t\right)$ corresponds to the local-equilibrium part of the statistical operator. The term $\hat{C}\left(t\right)$ is a thermodynamic “force” as it involves gradients of temperature, chemical potential, and the velocity field. Naturally, the term $\hat{B}\left(t\right)$ is then identified with the non-equilibrium part of the statistical operator. Here, the exponential function $e^{\delta (t_{1}-t)}$ is a convergence-enforcing factor, as δ is a small positive parameter that is sent to zero at the end of the calculation.

The statistical operator given by Equation (Equation 202) can now be used to derive the equations of motion for the dissipative currents. For this purpose, one treats the non-equilibrium part ([Disp-formula FD203b-entropy-26-00189]) as a perturbation. Keeping only the first-order terms in the Taylor expansion of $\hat{\rho }\left(t\right)$ with respect to the operator $\hat{B}\left(t\right)$ yields the usual first-order Navier–Stokes theory [166,168]. Including also all second-order terms in the Taylor expansion, one obtains second-order dissipative fluid dynamics within the Zubarev formalism, as detailed in [173].

The expansion of the statistical operator (Equation 202) up to second order in $\hat{B}$ reads [173]
(204)$$\hat{\rho }={\hat{\rho }}_{l}+{\hat{\rho }}_{1}+{\hat{\rho }}_{2}\;,$$
with
(205a)$$\begin{array}{c} {\hat{\rho }}_{1}\left(t\right)=\int d^{4}x_{1}\int_{0}^{1}d\tau \left[{\hat{C}}_{\tau }\left(x_{1}\right)-{\langle {\hat{C}}_{\tau }\left(x_{1}\right)\rangle }_{l}\right]{\hat{\rho }}_{l}\;, \end{array}$$
(205b)$$\begin{array}{cc} {\hat{\rho }}_{2}\left(t\right)\; & =\frac{1}{2}\int d^{4}x_{1}d^{4}x_{2}\int_{0}^{1}d\tau \int_{0}^{1}d\lambda [\tilde{T}\left\{{\hat{C}}_{\lambda }\left(x_{1}\right){\hat{C}}_{\tau }\left(x_{2}\right)\right\}-{\langle \tilde{T}\left\{{\hat{C}}_{\lambda }\left(x_{1}\right){\hat{C}}_{\tau }\left(x_{2}\right)\right\}\rangle }_{l}\\ \; & \;-{\langle {\hat{C}}_{\lambda }\left(x_{1}\right)\rangle }_{l}{\hat{C}}_{\tau }\left(x_{2}\right)-{\hat{C}}_{\lambda }\left(x_{1}\right){\langle {\hat{C}}_{\tau }\left(x_{2}\right)\rangle }_{l}+2{\langle {\hat{C}}_{\lambda }\left(x_{1}\right)\rangle }_{l}{\langle {\hat{C}}_{\tau }\left(x_{2}\right)\rangle }_{l}]{\hat{\rho }}_{l}\;, \end{array}$$
where $\tilde{T}$ is the anti-time-ordering operator and we introduced
(206)$${\hat{X}}_{\tau }=e^{-\tau \hat{A}}\hat{X}e^{\tau \hat{A}}\;,$$
for any operator $\hat{X}$, as well as
(207)$$\int d^{4}x_{1}\equiv \int d^{3}r_{1}\int_{-\infty }^{t}dt_{1}\;e^{\delta (t_{1}-t)}\;.$$

Given the generic expansions above, we can now write down the statistical average of an arbitrary operator $\hat{X}\left(x\right)$ with the help of Equations (Equation 200), (Equation 204), and ([Disp-formula FD205a-entropy-26-00189]) as
(208)$$\langle \hat{X}\left(x\right)\rangle ={\langle \hat{X}\left(x\right)\rangle }_{l}+\int d^{4}x_{1}\left(\hat{X}\left(x\right),\hat{C}\left(x_{1}\right)\right)+\int d^{4}x_{1}d^{4}x_{2}\left(\hat{X}\left(x\right),\hat{C}\left(x_{1}\right),\hat{C}\left(x_{2}\right)\right)\;,$$
where we defined the 2-point correlation function
(209a)$$\left(\hat{X}\left(x\right),\hat{Y}\left(x_{1}\right)\right)\equiv \int_{0}^{1}d\tau \;{\langle \hat{X}\left(x\right)\left[{\hat{Y}}_{\tau }\left(x_{1}\right)-{\langle {\hat{Y}}_{\tau }\left(x_{1}\right)\rangle }_{l}\right]\rangle }_{l}\;,$$
and the 3-point correlation function
(209b)$$\begin{array}{c} \left(\hat{X}\left(x\right),\hat{Y}\left(x_{1}\right),\hat{Z}\left(x_{2}\right)\right)\equiv \\ \equiv \frac{1}{2}\int_{0}^{1}d\tau \int_{0}^{1}d\lambda \;\langle \tilde{T}\left\{\hat{X}\left(x\right)[{\hat{Y}}_{\lambda }\left(x_{1}\right){\hat{Z}}_{\tau }\left(x_{2}\right)-{\langle {\hat{Y}}_{\lambda }\left(x_{1}\right)\rangle }_{l}{\hat{Z}}_{\tau }\left(x_{2}\right)-{\hat{Y}}_{\lambda }\left(x_{1}\right){\langle {\hat{Z}}_{\tau }\left(x_{2}\right)\rangle }_{l}\right.\\ \;-\left.{\langle \tilde{T}\;{\hat{Y}}_{\lambda }\left(x_{1}\right){\hat{Z}}_{\tau }\left(x_{2}\right)\rangle }_{l}+2{\langle {\hat{Y}}_{\lambda }\left(x_{1}\right)\rangle }_{l}{\langle {\hat{Z}}_{\tau }\left(x_{2}\right)\rangle }_{l}]\right\}\rangle_{l}\;. \end{array}$$

Using the conservation equations in the Landau frame (where the energy diffusion current vanishes), the operator $\hat{C}\left(t,r\right)$, Equation ([Disp-formula FD203c-entropy-26-00189]), can be written as [173]
(210)$$\hat{C}={\hat{C}}_{1}+{\hat{C}}_{2}\;,$$
where ${\hat{C}}_{1}$ and ${\hat{C}}_{2}$ are the first- and the second-order contributions in gradients and dissipative currents, respectively,
(211a)$$\begin{array}{cc} {\hat{C}}_{1} & =-\beta_{0}\theta {\hat{p}}^{*}+\beta_{0}{\hat{\pi }}_{\mu \nu }\sigma^{\mu \nu }-\hat{n}\cdot I\;, \end{array}$$
(211b)$$\begin{array}{cc} {\hat{C}}_{2} & =-{\hat{\beta }}^{*}\left(\Pi \theta -\pi^{\mu \nu }\sigma_{\mu \nu }\right)+{\hat{\alpha }}^{*}\partial \cdot n\;. \end{array}$$
Here, ${\hat{n}}^{\mu }$ is the operator of the particle diffusion current, $I^{\mu }\equiv \nabla^{\mu }\alpha_{0}$, and
(212a)$$\begin{array}{cc} {\hat{p}}^{*} & \equiv \hat{p}-\hat{\varepsilon }{\left.\frac{\partial p_{0}}{\partial \varepsilon_{0}}\right|}_{n_{0}}-\hat{n}{\left.\frac{\partial p_{0}}{\partial n_{0}}\right|}_{\varepsilon_{0}}\;, \end{array}$$
(212b)$$\begin{array}{cc} {\hat{\beta }}^{*} & \equiv \hat{\varepsilon }{\left.\frac{\partial \beta_{0}}{\partial \varepsilon_{0}}\right|}_{n_{0}}+\hat{n}{\left.\frac{\partial \beta_{0}}{\partial n_{0}}\right|}_{\varepsilon_{0}}\;, \end{array}$$
(212c)$$\begin{array}{cc} {\hat{\alpha }}^{*} & \equiv \hat{\varepsilon }{\left.\frac{\partial \alpha_{0}}{\partial \varepsilon_{0}}\right|}_{n_{0}}+\hat{n}{\left.\frac{\partial \alpha_{0}}{\partial n_{0}}\right|}_{\varepsilon_{0}}\;. \end{array}$$

The final step consists of consecutively inserting the operators $\hat{\Pi }$, ${\hat{n}}^{\mu }$, and ${\hat{\pi }}^{\mu \nu }$ for $\hat{X}$ into Equation (Equation 208), as well as $\hat{C}$ from Equation (Equation 210). Then, using Curie’s principle, one eventually derives the following constitutive relations for the dissipative currents [173] (note that we changed the notation of several transport coefficients with respect to [173], in order to adhere to the conventions used in other parts of the present work),
(213a)$$\begin{array}{cc} \Pi & =-\zeta \theta +\zeta \tau_{\Pi }\dot{\theta }+\lambda_{\Pi \pi }\left(\sigma_{\mu \nu }\pi^{\mu \nu }-\theta \Pi \right)-\ell_{\Pi n}\partial \cdot n+{\tilde{\zeta }}_{2}\sigma_{\mu \nu }\sigma^{\mu \nu }+{\tilde{\zeta }}_{3}\theta^{2}+{\tilde{\zeta }}_{4}\;I\cdot I\;, \end{array}$$
(213b)$$\begin{array}{cc} n^{\mu }\; & =\varkappa I^{\mu }-\varkappa \tau_{n}{\dot{I}}^{\langle \mu \rangle }+{\tilde{\varkappa }}_{3}I^{\mu }\theta -\lambda_{nn}\sigma^{\mu \nu }I_{\nu }\;, \end{array}$$
(213c)$$\begin{array}{cc} \pi^{\mu \nu }\; & =2\eta \sigma^{\mu \nu }-2\eta \tau_{\pi }{\dot{\sigma }}^{\langle \mu \nu \rangle }+{\tilde{\eta }}_{2}\theta \sigma^{\mu \nu }+{\tilde{\eta }}_{3}\sigma^{\lambda \langle \mu }\sigma_{\;\;\lambda }^{\nu \rangle }+{\tilde{\eta }}_{5}I^{\langle \mu }I^{\nu \rangle }\;. \end{array}$$
The transport coefficients in these expressions are given by Kubo relations. The first-order coefficients read
(214a)ζ=β0∫d4x1p^∗(x),p^∗(x1)=−ddωImGp^∗p^∗R(ω)|ω=0,
(214b)ϰ=−13∫d4x1n^λ(x),n^λ(x1)=T3ddωImGn^λn^λR(ω)|ω=0,
(214c)η=β010∫d4x1π^μν(x),π^μν(x1)=−110ddωImGπ^μνπ^μνR(ω)|ω=0,
where
(215)$$G_{\hat{X}\hat{Y}}^{R}\left(\omega \right)=-i\;\int_{0}^{\infty }dt\;e^{i\omega t}\;\;\int d^{3}r\;{\langle \left[\hat{X}\left(t,r\right),\hat{Y}\left(0,0\right)\right]\rangle }_{l}$$
is the Fourier transform of the retarded 2-point correlator taken in the zero-wavenumber limit and the square brackets denote the commutator. The second-order coefficients read
(216a)ζτΠ=12d2dω2ReGp^∗p^∗R(ω)|ω=0,ϰτn=−T6d2dω2ReGn^λn^λR(ω)|ω=0,
(216b)ητπ=120d2dω2ReGπ^ijπ^ijR(ω)|ω=0,
(216c)$$\begin{array}{c} {\tilde{\zeta }}_{2}=\frac{\beta_{0}^{2}}{5}\int d^{4}x_{1}d^{4}x_{2}\left({\hat{p}}^{*}\left(x\right),{\hat{\pi }}_{\gamma \delta }\left(x_{1}\right),{\hat{\pi }}^{\gamma \delta }\left(x_{2}\right)\right)\;, \end{array}$$
(216d)$$\begin{array}{c} {\tilde{\zeta }}_{4}=\frac{1}{3}\int d^{4}x_{1}d^{4}x_{2}\left({\hat{p}}^{*}\left(x\right),{\hat{n}}_{\gamma }\left(x_{1}\right),{\hat{n}}^{\gamma }\left(x_{2}\right)\right)\;, \end{array}$$
(216e)$$\begin{array}{c} \lambda_{\Pi \pi }=\int d^{4}x_{1}\left({\hat{p}}^{*}\left(x\right),{\hat{\beta }}^{*}\left(x_{1}\right)\right)\;,\;\ell_{\Pi n}=-\int d^{4}x_{1}\left({\hat{p}}^{*}\left(x\right),{\hat{\alpha }}^{*}\left(x_{1}\right)\right)\;, \end{array}$$
(216f)$$\begin{array}{c} {\tilde{\varkappa }}_{3}=\frac{2\beta_{0}}{3}\int d^{4}x_{1}d^{4}x_{2}\left({\hat{n}}_{\gamma }\left(x\right),{\hat{n}}^{\gamma }\left(x_{1}\right),{\hat{p}}^{*}\left(x_{2}\right)\right)\;, \end{array}$$
(216g)$$\begin{array}{c} \lambda_{nn}=\frac{2\beta_{0}}{5}\;\int \;d^{4}x_{1}d^{4}x_{2}\left({\hat{n}}^{\gamma }\left(x\right),{\hat{n}}^{\delta }\left(x_{1}\right),{\hat{\pi }}_{\gamma \delta }\left(x_{2}\right)\right)\;, \end{array}$$
(216h)$$\begin{array}{c} {\tilde{\eta }}_{2}=-\frac{2\beta_{0}^{2}}{5}\;\int d^{4}x_{1}d^{4}x_{2}\left({\hat{\pi }}_{\gamma \delta }\left(x\right),{\hat{\pi }}^{\gamma \delta }\left(x_{1}\right),{\hat{p}}^{*}\left(x_{2}\right)\right)-2\eta \tau_{\pi }{\left.\frac{\partial p_{0}}{\partial \varepsilon_{0}}\right|}_{n_{0}}\;, \end{array}$$
(216i)$$\begin{array}{c} {\tilde{\eta }}_{3}=\frac{12}{35}\beta_{0}^{2}\;\int d^{4}x_{1}d^{4}x_{2}\left({\hat{\pi }}_{\gamma }^{\delta }\left(x\right),{\hat{\pi }}_{\delta }^{\lambda }\left(x_{1}\right),{\hat{\pi }}_{\lambda }^{\gamma }\left(x_{2}\right)\right)\;, \end{array}$$
(216j)$$\begin{array}{c} {\tilde{\eta }}_{5}=\frac{1}{5}\int d^{4}x_{1}d^{4}x_{2}\left({\hat{\pi }}_{\gamma \delta }\left(x\right),{\hat{n}}^{\gamma }\left(x_{1}\right),{\hat{n}}^{\delta }\left(x_{2}\right)\right)\;. \end{array}$$
The most complicated coefficient is ${\tilde{\zeta }}_{3}$ in Equation ([Disp-formula FD213a-entropy-26-00189]),
(216k)$$\begin{array}{c} {\tilde{\zeta }}_{3}=\lambda_{\Pi }+\zeta^{*}+\psi_{\varepsilon_{0}\varepsilon_{0}}\zeta_{\varepsilon }^{2}+2\psi_{\varepsilon_{0}n_{0}}\zeta_{\varepsilon }\zeta_{n}+\psi_{n_{0}n_{0}}\zeta_{n}^{2}\;, \end{array}$$
with
(216l)$$\begin{array}{c} \lambda_{\Pi }=\beta_{0}^{2}\int d^{4}x_{1}d^{4}x_{2}\left({\hat{p}}^{*}\left(x\right),{\hat{p}}^{*}\left(x_{1}\right),{\hat{p}}^{*}\left(x_{2}\right)\right)\;, \end{array}$$
(216m)$$\begin{array}{c} \zeta^{*}={\left.\frac{\partial p_{0}}{\partial \varepsilon_{0}}\right|}_{n_{0}}\zeta \tau_{\Pi }+2\zeta_{\varepsilon }\tau_{\varepsilon }\left[\psi_{\varepsilon_{0}\varepsilon_{0}}\left(\varepsilon_{0}+p_{0}\right)+n_{0}\psi_{\varepsilon_{0}n_{0}}\right]+2\zeta_{n}\tau_{n}\left[\psi_{\varepsilon_{0}n_{0}}\left(\varepsilon_{0}+p_{0}\right)+n_{0}\psi_{n_{0}n_{0}}\right]\;, \end{array}$$
(216n)ζε=β0∫d4x1ε^(x),p^∗(x1)=−ddωImGε^p^∗R(ω)|ω=0,
(216o)ζn=β0∫d4x1n^(x),p^∗(x1)=−ddωImGn^p^∗R(ω)|ω=0,
(216p)ζετε=12d2dω2ReGp^∗ε^R(ω)|ω=0,ζnτn=12d2dω2ReGp^∗n^R(ω)|ω=0,
and $\psi_{ab}=\frac{1}{2}\partial^{2}p_{0}/\partial a\partial b$, with $a,b=\varepsilon_{0},n_{0}$. There are several things to note about Equations ([Disp-formula FD213a-entropy-26-00189])–([Disp-formula FD216a-entropy-26-00189]):(i)Equations ([Disp-formula FD213a-entropy-26-00189]) are the same type of constitutive relations for the dissipative currents as in BRSSS theory, cf. Section 3.5, and *not* relaxation-type equations, as in Equations ([Disp-formula FD23a-entropy-26-00189]). However, with the same manipulation as there, i.e., replacing, e.g., $\sigma^{\mu \nu }\to \pi^{\mu \nu }/\eta$, one can turn them into the latter type of equations, for details see [173].(ii)The transport coefficients ([Disp-formula FD214a-entropy-26-00189]), ([Disp-formula FD216a-entropy-26-00189]) are given by Kubo relations and are, thus, intrinsically *nonperturbative*.(iii)While the first-order transport coefficients ([Disp-formula FD214a-entropy-26-00189]) are given by 2-point correlation functions, some second-order coefficients are given by *3-point correlation functions*.(iv)Not all second-order coefficients that appear in Equations ([Disp-formula FD23a-entropy-26-00189]), i.e., that are allowed by Lorentz symmetry, actually show up in the Zubarev approach, i.e., their corresponding values are zero.(v)The relaxation times are given by second-order derivatives of retarded Green’s functions, cf. Equation ([Disp-formula FD216a-entropy-26-00189]), not by poles of the Green’s function in the complex frequency plane [175].

Finally, we note that transport coefficients can also be computed by embedding the theory in background fields, whose variations allow one to define the correlators of conserved currents and, consequently, transport coefficients via Kubo formulas (see [176,177] for a review). For example, by considering metric variations, one can define retarded Green’s functions of the energy-momentum tensor. This has been particularly useful for understanding the transport properties of strongly coupled theories in the context of the holographic correspondence [178], which allows one to compute retarded correlators and the respective transport coefficients via Kubo formulas for a vast class of gauge theories with gravity duals [179,180,181].

## 7. Further Developments

We conclude this review by briefly mentioning some further developments in the theory of relativistic fluid dynamics, but without attempting a self-contained presentation. We start the discussion with anisotropic fluid dynamics, in which an alternative, non-equilibrium reference state is employed instead of the local-equilibrium one to derive the fluid-dynamical equations. This is followed by a brief review of maximum-entropy fluid dynamics, where the form of the distribution function is determined from maximizing an entropy functional. What distinguishes both anisotropic and maximum-entropy fluid dynamics from the other approaches discussed in this paper is that they extend the fluid-dynamical approach further into the non-equilibrium domain, by assuming physically motivated parametrizations of the (possibly large) dissipative terms in $T^{\mu \nu }$ and $N^{\mu }$. Then, we discuss relativistic transient second-order dissipative magnetohydrodynamics, in which the dynamics of electromagnetic fields is coupled to that of the fluid, and show how to derive it from kinetic theory with an external force term. Closing the chapter, we consider the formulation of fluid dynamics where spin degrees of freedom are taken into account, so-called spin hydrodynamics, which has received recent attention in heavy-ion physics. It can also be derived via kinetic theory by enlarging the phase space to encompass spin degrees of freedom.

### 7.1. Anisotropic Fluid Dynamics

Here, we briefly comment on some pertinent developments concerning anisotropic fluid dynamics [182,183]. The latter has evolved into a very active field over the recent years, with many applications in heavy-ion collisions [184,185]. We refer the reader to the dedicated review [186] for a general discussion of anisotropic fluid dynamics and its applications.

In ordinary fluid dynamics, one usually assumes a local-equilibrium reference state, characterized by a single-particle distribution function $f_{0k}$, cf. Equation (Equation 83). In contrast, in anisotropic fluid dynamics, one generalizes this to a non-equilibrium reference state, characterized by a single-particle distribution function ${\hat{f}}_{0k}$ which is chosen to incorporate certain properties of the system. For instance, the initial stages of a heavy-ion collision feature a strong momentum anisotropy, which can be parametrized as [187,188]
(217)$${\hat{f}}_{0k}\equiv \frac{g}{\exp(\hat{\beta }\sqrt{k^{\mu }k^{\mu }\Omega_{\mu \nu }}-\hat{\alpha })-a}\;,$$
where
(218)$$\;\Omega_{\mu \nu }\equiv u_{\mu }u_{\mu }+\xi l_{\mu }l_{\nu }\;,$$
and $l^{\mu }$ is a space-like normalized vector orthogonal to $u^{\mu }$, l·l=−1, l·u=0, characterizing the direction of the anisotropy (in heavy-ion collisions, usually the beam direction). The constant ξ parametrizes the strength of the momentum anisotropy, while $\hat{\alpha }$ and $\hat{\beta }$ are parameters that allow to adjust the energy and the particle-number density in the system. For ξ→0, the distribution (Equation 217) becomes the usual local-equilibrium distribution function, and $\hat{\alpha }\to \alpha_{0}\equiv \beta_{0}\mu$, $\hat{\beta }\to \beta_{0}\equiv 1/T$.

The rationale to use a non-equilibrium reference state like that given by Equation (Equation 217) is the following: if one is far from local equilibrium, the deviation $\delta f_{k}\equiv f_{k}-f_{0k}$ of the actual single-particle distribution function $f_{k}$ from the local-equilibrium one $f_{0k}$ may be large, $|\delta f_{k}|∼f_{0k}$, such that an expansion around the local-equilibrium state has bad convergence properties. On the other hand, if the actual single-particle distribution $f_{k}$ is already reasonably well approximated by the non-equilibrium distribution ${\hat{f}}_{0k}$, then the deviation $\delta {\hat{f}}_{k}\equiv f_{k}-{\hat{f}}_{0k}$ may be small, $|\delta {\hat{f}}_{k}|\ll 1$, such that an expansion around the non-equilibrium state converges rapidly.

As compared to the case of isotropic local equilibrium, the mathematical difficulty lies in the fact that the three-dimensional subspace orthogonal to $u^{\mu }$ is now further subdivided into a one-dimensional subspace parallel to $l^{\mu }$ and a two-dimensional subspace orthogonal to both $u^{\mu }$ and $l^{\mu }$. One defines the energy of a particle with 4-momentum $k^{\mu }$ as $E_{ku}\equiv k\cdot u$, its momentum component in $l^{\mu }$-direction as $E_{kl}\equiv -k\cdot l$, as well as the two-dimensional projection operator
(219)$$\Xi^{\mu \nu }\equiv {∆}^{\mu \nu }+l^{\mu }l^{\nu }\;,$$
with ${∆}^{\mu \nu }=g^{\mu \nu }-u^{\mu }u^{\nu }$ being the 3-dimensional projection operator onto the subspace orthogonal to $u^{\mu }$. The tensor projections of fluid-dynamical quantities now become more involved: in addition to the usual terms projected onto the direction of $u^{\mu }$, they now feature terms projected onto the direction of $l^{\mu }$, as well as terms projected using the projector (Equation 219), e.g., for the particle-number 4-current and the energy-momentum tensor we now obtain
(220)$$\begin{array}{cc} N^{\mu } & =n\;u^{\mu }+n_{l}\;l^{\mu }+V_{⊥}^{\mu }\;, \end{array}$$
(221)$$\begin{array}{cc} T^{\mu \nu } & =\varepsilon \;u^{\mu }u^{\nu }+2\;M\;u^{\left(\mu \right.}l^{\left.\nu \right)}+P_{l}\;l^{\mu }l^{\nu }-P_{⊥}\;\Xi^{\mu \nu }+2\;W_{⊥u}^{\left(\mu \right.}u^{\left.\nu \right)}+2\;W_{⊥l}^{\left(\mu \right.}l^{\left.\nu \right)}+\pi_{⊥}^{\mu \nu }\;. \end{array}$$
Here, in addition to particle-number density *n* and energy density ε, we have denoted the part of the particle diffusion current in the $l^{\mu }$-direction by $n_{l}$, while the particle diffusion current orthogonal to both $u^{\mu }$ and $l^{\mu }$ is denoted by $V_{⊥}^{\mu }$. The pressure in the direction orthogonal to both $u^{\mu }$ and $l^{\mu }$ is denoted by $P_{⊥}$, while the pressure in the direction parallel to $l^{\mu }$ is $P_{l}$. The projection of the energy-momentum tensor in both $u^{\mu }$- and $l^{\nu }$-direction is denoted by *M*. The projection in either $u^{\mu }$- or $l^{\mu }$-direction and orthogonal to both directions is denoted by $W_{⊥u}^{\mu }$ or $W_{⊥l}^{\mu }$, respectively. The only rank-2 tensor in the subspace orthogonal to both $u^{\mu }$ and $l^{\mu }$ is given by the transverse shear-stress tensor $\pi_{⊥}^{\mu \nu }$.

Using the method of moments, one can derive equations of motion for second-order dissipative fluid dynamics with a reference state parametrized by the single-particle distribution function (Equation 217), for details, see [189]. The resulting conservation equations and equations of motion for the dissipative currents are too lengthy to quote here but can be found in [189]. Usually, one restricts the application of anisotropic fluid dynamics to the leading-order case, where $\delta {\hat{f}}_{k}=0$. Then, $n_{l}$, *M*, $V_{⊥}^{\mu }$, $W_{⊥u}^{\mu }$, $W_{⊥l}^{\mu }$, and $\pi_{⊥}^{\mu \nu }$ vanish, just as in ideal fluid dynamics, while the pressure remains anisotropic, $P_{l}\neq P_{⊥}$, see, e.g., [188]. This case is closest to ideal fluid dynamics and only requires an additional equation of motion to determine the anisotropy parameter ξ in the distribution function (Equation 217). As shown in [188], choosing the equation of motion for $P_{l}$ as derived from the system of moment equations for the Boltzmann equation gives the best agreement with the solution of the microscopic Boltzmann equation in relaxation-time approximation. The case where all dissipative corrections to the anisotropic non-equilibrium reference state pertaining to the energy-momentum tensor (but not to the particle current) are taken into account was studied in detail in [185,190,191,192], where it was also shown that this theory is highly successful in the description of data from heavy-ion collisions at high energies (where the baryon number density is small).

### 7.2. Maximum-Entropy Fluid Dynamics

In this section, we shortly introduce the key concepts behind *maximum-entropy (ME)* fluid dynamics [193]. As several other fluid-dynamical theories discussed in this review, this theory is based on kinetic theory and was inspired by the application to relativistic heavy-ion collisions, in which one usually concatenates methods of (macroscopic) fluid dynamics to (microscopic) hadron-resonance dynamics [194]. While the evolution of the strongly coupled medium is characterized by fluid dynamics, at some point the system becomes dilute, such that the description has to be switched to a kinetic approach. To achieve the transition from a fluid-dynamical model, whose dynamical fields are given by the conserved currents, i.e., $N^{\mu }$ and $T^{\mu \nu }$, to a kinetic formulation, which is determined by a phase-space distribution function f(x,k), some approximation has to be made about the structure of f(x,k) in momentum space. One possible Ansatz that is widely used consists of linearizing the distribution function around local equilibrium and truncating the expansion such that the fluid-dynamical degrees of freedom remain. Explicitly, this corresponds to restricting the sum in Equation (Equation 145) to ℓ∈{0,1,2}, and choosing $N_{0}=2$, $N_{1}=1$, and $N_{2}=0$.

While straightforward, the problem with such an approach is that it puts strong assumptions on the momentum-space structure of f(x,k). The more agnostic approach put forward in [194] constrains the distribution function by demanding that it maximizes the entropy density in the fluid rest frame, u·S, where the entropy four-current in kinetic theory has been introduced in Equation ([Disp-formula FD82a-entropy-26-00189]). This maximization is subject to the constraint that the kinetic description of the fluid-dynamical degrees of freedom (particle 4-current and energy-momentum tensor) matches the actual values of the latter. The result of this procedure is the so-called ME distribution function
(222)$$f_{\mathrm{ME}}\left(x,k\right)={\left[\exp\left(\Lambda E_{k}-\frac{\lambda_{\Pi }}{E_{k}}k^{\langle \mu \rangle }k_{\langle \mu \rangle }+\frac{\gamma_{\langle \mu \nu \rangle }}{E_{k}}k^{\langle \mu }k^{\nu \rangle }\right)+a\right]}^{-1}\;,$$
where Λ, $\lambda_{\Pi }$, and $\gamma_{\langle \mu \nu \rangle }$ are Lagrange multipliers defined by the relations
(223)$$\varepsilon_{0}=\int dK\;E_{k}^{2}f_{\mathrm{ME}}\;,\;p_{0}+\Pi =-\frac{1}{3}\int dK\;{∆}_{\mu \nu }k^{\mu }k^{\nu }f_{\mathrm{ME}}\;,\;\pi^{\mu \nu }=\int dK\;k^{\langle \mu }k^{\nu \rangle }f_{\mathrm{ME}}\;.$$
The distribution function (Equation 222) is similar to the Ansatz originally proposed by Israel and Stewart in [8], except that they expanded this function around equilibrium and then matched the expansion coefficients to the fluid-dynamical degrees of freedom. The latter can be done analytically, while the ME method requires a numerical evaluation of Equations (Equation 223).

The distribution function $f_{\mathrm{ME}}$ shares the feature of introducing a more complicated momentum structure in the exponential with the anisotropic single-particle distribution function ${\hat{f}}_{0k}$, cf. Equation (Equation 217), although their explicit forms differ. Furthermore, while ${\hat{f}}_{0k}$ is motivated by a concrete physical scenario, the ME distribution function is built to accommodate information from any given fluid configuration in the least biased way.

As demonstrated in [193] for highly symmetric geometries, this distribution function also provides a way to close the hierarchy of moment equations[note 14]. The main idea consists of replacing the actual distribution function by $f_{\mathrm{ME}}$ in those terms which are not directly related to the fluid-dynamical degrees of freedom. In this way, one is able to express the terms in question as functions of the Lagrange multipliers Λ, $\lambda_{\Pi }$, and $\gamma_{\langle \mu \nu \rangle }$, which in turn are related to fluid-dynamical quantities through Equation (Equation 223). This then closes the system of equations, and the resulting theory of fluid dynamics is referred to as ME fluid dynamics.

The procedure described above has been carried out in [193] for Bjorken and Gubser flow geometries, using an RTA collision kernel. Indeed, the resulting time evolution of the dissipative currents show good agreement with the underlying kinetic description, even in the regions which are far away from local equilibrium. The method has not yet been put to the test in fully (3+1)-dimensional flows, but work in this direction is ongoing.

### 7.3. Relativistic Second-Order Dissipative Magnetohydrodynamics

The derivation of second-order dissipative fluid dynamics via the method of moments, as discussed in Section 5.4, can be generalized to the case where electromagnetic fields are present, thus resulting in the equations of motion of relativistic second-order dissipative magnetohydrodynamics [155,158]. To this end, one starts with the Boltzmann equation (Equation 76) in the presence of electromagnetic fields, which exert a Lorentz force on the charge carriers,
(224)$$k\cdot \partial f_{k}+qF^{\mu \nu }k_{\nu }\frac{\partial }{\partial k^{\mu }}f_{k}=C\left[f_{k}\right]\;,$$
where q is the electric charge of the particles and $F^{\mu \nu }$ the field-strength tensor of the electromagnetic field. Here, we have assumed that there is only a single species of charged particles, which do not carry an electric or magnetic moment. The latter would result in an additional term on the left-hand side, the so-called Mathisson force [118].

The fluid-dynamical conservation laws look formally the same as in the case without electromagnetic fields, see Equation ([Disp-formula FD3a-entropy-26-00189]), where, in the Landau frame,
(225)$$N^{\mu }\equiv nu^{\mu }+n^{\mu }$$
is the particle number 4-current. However, now
(226)$$T^{\mu \nu }\equiv T_{\mathrm{em}}^{\mu \nu }+T_{f}^{\mu \nu }$$
is the *total* energy-momentum tensor of the system, where
(227)$$T_{\mathrm{em}}^{\mu \nu }\equiv -F^{\mu \lambda }F_{\;\;\lambda }^{\nu }+\frac{1}{4}g^{\mu \nu }F^{\alpha \beta }F_{\alpha \beta }$$
is the energy-momentum tensor of the electromagnetic field and
(228)$$T_{f}^{\mu \nu }\equiv \varepsilon u^{\mu }u^{\nu }-p{∆}^{\mu \nu }+\pi^{\mu \nu }$$
the energy-momentum tensor of the fluid.

The conservation laws ([Disp-formula FD3a-entropy-26-00189]) have to be supplemented by Maxwell’s equations,
(229)$$\partial_{\mu }F^{\mu \nu }=qN^{\nu }\;,\;\epsilon^{\mu \nu \alpha \beta }\partial_{\mu }F_{\alpha \beta }=0\;.$$
These equations imply that
(230)$$\partial_{\mu }T_{\mathrm{em}}^{\mu \nu }=-q\;F^{\nu \lambda }N_{\lambda }\;.$$
From this and Equation ([Disp-formula FD3a-entropy-26-00189]) follows that the energy-momentum tensor of the fluid satisfies [2]
(231)$$\partial_{\mu }T_{f}^{\mu \nu }=q\;F^{\nu \lambda }N_{\lambda }\;.$$

The field-strength tensor can be tensor-decomposed with respect to the fluid 4-velocity $u^{\mu }$ as
(232)$$F^{\mu \nu }\equiv E^{\mu }u^{\nu }-E^{\nu }u^{\mu }+\epsilon^{\mu \nu \alpha \beta }u_{\alpha }B_{\beta }\;,$$
where the electric and magnetic field 4-vectors $E^{\mu }\equiv F^{\mu \nu }u_{\nu }$ and $B^{\mu }\equiv \frac{1}{2}\epsilon^{\mu \nu \alpha \beta }F_{\alpha \beta }u_{\nu }$, respectively, with $\epsilon^{\mu \nu \alpha \beta }$ being the Levi-Civita tensor.

The Lorentz-force term in Equation (Equation 224) gives rise to additional terms in the equations of motion of the irreducible moments and, therefore, ultimately also in the equations of motion of the dissipative currents. Applying the method of moments and working in the Landau frame and in the 14-moment approximation, one arrives at the following set of equations; for details see [155,158]:
(233a)$$\begin{array}{cc} \tau_{\Pi }\dot{\Pi }+\Pi & =-\zeta \theta -\delta_{\Pi \Pi }\;\Pi \theta -\ell_{\Pi n}\;\nabla \cdot n-\tau_{\Pi n}\;n\cdot F-\lambda_{\Pi n}\;n\cdot I+\lambda_{\Pi \pi }\;\pi^{\mu \nu }\sigma_{\mu \nu }\\ \; & -\delta_{\Pi nE}\;q\;n\cdot E\;, \end{array}$$
(233b)$$\begin{array}{cc} \tau_{n}{\dot{n}}^{\langle \mu \rangle }+n^{\mu }\; & =\varkappa I^{\mu }-\delta_{nn}\;n^{\mu }\theta -\ell_{n\Pi }\nabla^{\mu }\Pi +\ell_{n\pi }\nabla_{\lambda }\pi^{\lambda \langle \mu \rangle }+\tau_{n\Pi }\;\Pi F^{\mu }\\ \; & -\tau_{n\pi }\;\pi^{\mu \nu }F_{\nu }-\lambda_{nn}\;\sigma^{\mu \nu }n_{\nu }+\lambda_{n\Pi }\;\Pi I^{\mu }-\lambda_{n\pi }\;\pi^{\mu \nu }I_{\nu }+\tau_{n}\omega^{\mu \nu }n_{\nu }\\ \; & -\delta_{nB}\;q\;Bb^{\mu \nu }n_{\nu }+\delta_{nE}\;q\;E^{\mu }+\delta_{n\Pi E}\;q\;\Pi E^{\mu }+\delta_{n\pi E}\;q\;\pi^{\mu \nu }E_{\nu }\;, \end{array}$$
(233c)$$\begin{array}{cc} \tau_{\pi }{\dot{\pi }}^{\langle \mu \nu \rangle }+\pi^{\mu \nu } & =2\eta \sigma^{\mu \nu }-\delta_{\pi \pi }\;\pi^{\mu \nu }\theta +\ell_{\pi n}\nabla^{\langle \mu }n^{\nu \rangle }-\tau_{\pi n}\;n^{\langle \mu }F^{\nu \rangle }+\lambda_{\pi \Pi }\;\Pi \sigma^{\mu \nu }\\ \; & -\lambda_{\pi \pi }\;\pi^{\lambda \langle \mu }\sigma_{\lambda }^{\nu \rangle }+\lambda_{\pi n}\;n^{\langle \mu }I^{\nu \rangle }+2\tau_{\pi }\pi_{\lambda }^{\langle \mu }\omega^{\nu \rangle \lambda }\\ \; & -\delta_{\pi B}\;qBb^{\alpha \beta }{∆}_{\alpha \kappa }^{\mu \nu }g_{\lambda \beta }\pi^{\kappa \lambda }+\delta_{\pi nE}\;q\;E^{\langle \mu }n^{\nu \rangle }\;, \end{array}$$
with $B\equiv \sqrt{-B\cdot B}$ and $b^{\mu \nu }\equiv -\epsilon^{\mu \nu \alpha \beta }u_{\alpha }b_{\beta }$, where $b^{\mu }\equiv B^{\mu }/B$. The last line in each equation contains the new terms due to the coupling to the electromagnetic fields.

Of particular interest is the Navier–Stokes limit of Equations ([Disp-formula FD233b-entropy-26-00189]), where all second-order terms are neglected. Counting electric fields as ∼O(Kn) and magnetic fields as ∼O(1) [196], one arrives at
(234a)$$\begin{array}{cc} n^{\mu }\; & =\varkappa I^{\mu }+\delta_{nE}\;q\;E^{\mu }-\delta_{nB}\;q\;Bb^{\mu \nu }n_{\nu }\;, \end{array}$$
(234b)$$\begin{array}{cc} \pi^{\mu \nu } & =2\eta \sigma^{\mu \nu }-\delta_{\pi B}\;qBb^{\alpha \beta }{∆}_{\alpha \kappa }^{\mu \nu }g_{\lambda \beta }\pi^{\kappa \lambda }\;. \end{array}$$
The Ohmic induction current is given by the second term of Equation ([Disp-formula FD234a-entropy-26-00189]) (after multiplying by q),
(235)$$J_{\mathrm{ind}}^{\mu }\equiv \sigma_{E}E^{\mu }\;,$$
with the electric conductivity
(236)$$\sigma_{E}\equiv q^{2}\delta_{nE}\;.$$
As originally noted by Einstein [139], the electric conductivity and the particle-diffusion coefficient must be related by
(237)$$\sigma_{E}=q^{2}\beta_{0}\varkappa \;,$$
which is the kinetic-theory version of the famous Wiedemann–Franz law. For the massless Boltzmann gas, the validity of this relation can be easily checked using the relation $\delta_{nE}=\frac{3}{16}n_{0}\beta_{0}\lambda_{\mathrm{mfp}}$ and the fact that $\varkappa =\frac{3}{16}n_{0}\lambda_{\mathrm{mfp}}$ [42]. As noted in [197], this relation must also hold for a different reason: in a state of constant *T* and $u^{\mu }$ and in the absence of dissipation, an electric field induces a charge-density gradient such that
(238)$$I^{\mu }\equiv \nabla^{\mu }\alpha_{0}=-q\beta_{0}E^{\mu }\;.$$
This relation can also be found from the second-order transport equation ([Disp-formula FD233b-entropy-26-00189]), setting all dissipative quantities to zero, which leads to the condition $\varkappa I^{\mu }=-\delta_{nE}\;q\;E^{\mu }$. This relation, together with Equation (Equation 238), then confirms the Einstein relation (Equation 237).

A nonzero magnetic field singles out a particular direction in space and breaks spatial isotropy. In this case, the Navier–Stokes relations become
(239)$$\Pi =-\zeta^{\mu \nu }\partial_{\mu }u_{\nu }\;,\;n^{\mu }=\varkappa^{\mu \nu }I_{\nu }\;,\;\pi^{\mu \nu }=\eta^{\mu \nu \alpha \beta }\sigma_{\alpha \beta }\;,$$
with tensor-valued transport coefficients [155],
(240a)$$\begin{array}{cc} \zeta^{\mu \nu } & =\zeta_{⊥}\Xi^{\mu \nu }-\zeta_{\|}b^{\mu }b^{\nu }-\zeta_{\times }b^{\mu \nu }\;, \end{array}$$
(240b)$$\begin{array}{cc} \varkappa^{\mu \nu } & =\varkappa_{⊥}\Xi^{\mu \nu }-\varkappa_{\|}b^{\mu }b^{\nu }-\varkappa_{\times }b^{\mu \nu }\;, \end{array}$$
(240c)$$\begin{array}{cc} \eta^{\mu \nu \alpha \beta } & =2{\bar{\eta }}_{0}\;{∆}^{\mu \nu \alpha \beta }+{\bar{\eta }}_{1}\left({∆}^{\mu \nu }-\frac{3}{2}\Xi^{\mu \nu }\right)\left({∆}^{\alpha \beta }-\frac{3}{2}\Xi^{\alpha \beta }\right)\\ \; & -2{\bar{\eta }}_{2}\left(\Xi^{\mu \alpha }b^{\nu }b^{\beta }+\Xi^{\nu \alpha }b^{\mu }b^{\beta }\right)-2{\bar{\eta }}_{3}\left(\Xi^{\mu \alpha }b^{\nu \beta }+\Xi^{\nu \alpha }b^{\mu \beta }\right)+2{\bar{\eta }}_{4}\left(b^{\mu \alpha }b^{\nu }b^{\beta }+b^{\nu \alpha }b^{\mu }b^{\beta }\right)\;, \end{array}$$
where $\Xi^{\mu \nu }\equiv {∆}^{\mu \nu }+b^{\mu }b^{\nu }$. In the 14-moment approximation, the scalar transport coefficients are given by [155]
(241a)$$\begin{array}{cc} \zeta_{\|} & =\zeta_{⊥}=\zeta \;,\;\zeta_{\times }=0\;, \end{array}$$
(241b)$$\begin{array}{cc} \varkappa_{\|} & =\varkappa \;,\;\varkappa_{⊥}=\frac{\varkappa }{1+{\left(\varkappa \;qBG_{n}\right)}^{2}}\;,\;\varkappa_{\times }=\varkappa \frac{\varkappa \;qBG_{n}}{1+{\left(\varkappa \;qBG_{n}\right)}^{2}}\;, \end{array}$$
(241c)$$\begin{array}{cc} {\bar{\eta }}_{0} & =\frac{\eta }{1+{\left(2\eta \;qBG_{\pi }\right)}^{2}}\;,\;{\bar{\eta }}_{1}=\frac{\eta }{3}\frac{{\left(4\eta \;qBG_{\pi }\right)}^{2}}{1+{\left(2\eta \;qBG_{\pi }\right)}^{2}}\;, \end{array}$$
(241d)$$\begin{array}{cc} {\bar{\eta }}_{2} & =3\eta \frac{{\left(\eta \;qBG_{\pi }\right)}^{2}}{1+{\left(\eta \;qBG_{\pi }\right)}^{2}}\frac{1}{1+{\left(2\eta \;qBG_{\pi }\right)}^{2}}\;, \end{array}$$
(241e)$$\begin{array}{cc} {\bar{\eta }}_{3} & =\eta \;\frac{\eta \;qBG_{\pi }}{1+{\left(2\eta \;qBG_{\pi }\right)}^{2}}\;,\;{\bar{\eta }}_{4}=\eta \;\frac{\eta \;qBG_{\pi }}{1+{\left(\eta \;qBG_{\pi }\right)}^{2}}\;, \end{array}$$
where ζ, ϰ, and η are the usual transport coefficients for vanishing magnetic field,
(242)$$\zeta =\frac{m^{2}}{3}\frac{\alpha_{0}^{\left(0\right)}}{L_{00}^{\left(0\right)}}\;,\;\varkappa =\frac{\alpha_{0}^{\left(1\right)}}{L_{00}^{\left(1\right)}}\;,\;\eta =\frac{\alpha_{0}^{\left(2\right)}}{L_{00}^{\left(2\right)}}\;,$$
with the $\alpha_{0}^{\left(\ell \right)}$ defined in Equation ([Disp-formula FD151a-entropy-26-00189]) and $L_{00}^{\left(\ell \right)}$ defined in Equation ([Disp-formula FD153b-entropy-26-00189]). Furthermore,
(243)$$G_{n}\equiv \frac{F_{10}^{\left(1\right)}+\alpha_{0}^{h}}{\alpha_{0}^{\left(1\right)}}\;,\;G_{\pi }\equiv \frac{F_{10}^{\left(2\right)}}{\alpha_{0}^{\left(2\right)}}$$
are thermodynamic functions depending only on $\alpha_{0}$ and $\beta_{0}$, with $F_{10}^{\left(\ell \right)}$ from Equation (Equation 183) and $\alpha_{0}^{h}\equiv -\beta_{0}J_{21}/\left(\varepsilon_{0}+p_{0}\right)$. Equation ([Disp-formula FD241a-entropy-26-00189]) implies that the Navier–Stokes limit for the bulk viscosity is the same as without magnetic field, Π=−ζθ. This holds at least when the magnetic field is not too strong, such that Landau-level quantization can be neglected and Equation (Equation 224) remains valid. For bulk viscosity in strong magnetic fields, see [198]. The transport coefficients for particle diffusion and shear stress, however, are different when the magnetic field is nonzero.

Finally, we note that including dissipation in relativistic magnetohydrodynamics is relevant for various fields besides heavy-ion collisions. For example, dissipative effects are expected to be non-negligible to describe the plasma surrounding supermassive black holes [199]. This follows from the fact that low-luminosity black holes cannot accrete matter at a rate that balances dissipative effects. This, in turn, heats the plasma and expands it into optically thin disks of hot, low-density charged particles. Such systems are thus expected to be nearly collisionless, as the collisional Coulomb mean free path is expected to be orders of magnitude larger than the black hole-horizon radius. Therefore, at least naively, the Knudsen number of the system is much larger than unity. This should serve as a solid motivation to investigate the properties of dissipative relativistic magnetohydrodynamics further [78,155,158,200,201,202,203,204] and its extension to the far-from-equilibrium regime.

### 7.4. Spin Hydrodynamics

The fundamental conservation laws of fluid dynamics, Equations ([Disp-formula FD3a-entropy-26-00189]), are, by Noether’s theorem, the consequence of continuous symmetries of the system. While the conservation of, e.g., the electric-charge current (which is equivalent to the particle-number current in a monatomic fluid) arises from the U(1) symmetry of electromagnetic interactions, the conservation of energy and momentum follows from the invariance under the group of space-time translations $R^{1,3}$. In general, however, a relativistic system should also be invariant under the homogeneous Lorentz group SO(1,3), which leads to the conservation of the total angular momentum,
(244)$$\partial_{\lambda }J^{\lambda \mu \nu }=0\;.$$
Since the total angular momentum can be decomposed as
(245)$$J^{\lambda \mu \nu }=2T^{\lambda [\nu }x^{\mu ]}+S^{\lambda \mu \nu }\;,$$
where $S^{\lambda \mu \nu }=-S^{\lambda \nu \mu }$ is the *spin tensor*, the conservation law (Equation 244) becomes
(246)$$\partial_{\lambda }S^{\lambda \mu \nu }=2T^{\left[\nu \mu \right]}\;,$$
where we used energy-momentum conservation. For a fluid consisting of particles without spin, $S^{\lambda \mu \nu }=0$, the conservation law (Equation 246) is automatically fulfilled as long as the energy-momentum tensor is symmetric. In the case that the spin tensor is nonzero, this situation changes[note 15].

In order to properly describe fluids whose spin tensor does not vanish, the relation (Equation 246) has to be included as a set of additional equations of motion describing the evolution of the spin tensor. As in the case of usual fluid dynamics, it is clear that, in the dissipative case, the conservation laws are insufficient to determine all independent degrees of freedom of the system. In particular, Equation (Equation 246) provides six equations of motion, whereas the spin tensor has 4×6=24 components. The conservation of the total angular momentum may be taken to describe the evolution of the so-called *spin potential* $\Omega_{0}^{\mu \nu }$, which is an antisymmetric second-rank tensor and thus has six independent components, similar to the electromagnetic field-strength tensor. This spin potential, which can be decomposed into electric- and magnetic-like parts,
(247)$$\Omega_{0}^{\mu \nu }=2u^{[\mu }\kappa_{0}^{\nu ]}+\epsilon^{\mu \nu \alpha \beta }u_{\alpha }\omega_{0,\beta }\;,$$
may be taken to be the intensive variable which is the thermodynamic conjugate of the spin density. In global equilibrium, it has to be constant and equal to the so-called *thermal vorticity* $\varpi^{\mu \nu }\equiv \partial^{[\nu }\beta^{\mu ]}$ [125,127], whereas in general, it is a spacetime-dependent field whose evolution is determined by Equation (Equation 246). It has been found that, when considering small perturbations around a static background, the components of the spin potential show a wave-like behavior [213].

To describe the full behavior of the spin tensor in the dissipative case, essentially all methods presented in the previous sections can be applied. There have been numerous works deriving the additional equations needed from the entropy principle [214,215,216], Zubarev’s non-equilibrium statistical operator [217,218], Lagrangian hydrodynamics [219,220], Chapman–Enskog theory [221], coupling to torsion [222,223,224], and kinetic theory [87,122,123,127,156,225,226,227,228]. The linear stability of spin hydrodynamics has been addressed in [87,229,230], and steps towards spin hydrodynamics in the presence of electromagnetic fields have been undertaken in [231,232,233]. While a full account of the various formulations of spin hydrodynamics as described above is outside the scope of this work, we will outline the most salient features when considering a formulation based on quantum kinetic theory, which is similar to the procedures discussed in Section 5.

One important difference to standard kinetic theory lies in the fact that the phase space consists not only of position and momentum, but acquires an additional spin degree of freedom, which is realized through a space-like 4-vector $s^{\mu }$ of length σ which is orthogonal to the 4-momentum $k^{\mu }$, with the associated measure $dS\left(k\right)\equiv \left(k/\sigma \pi \right)d^{4}s\delta \left(s^{2}+\sigma^{2}\right)\delta \left(k\cdot s\right)$[note 16]. Consequently, the single-particle distribution function will depend on position, momentum, and spin, f=f(x,k,s). Note that in the case of a distribution that does not depend on spin, the integral over spin space reduces to a simple factor, ∫dS=2s+1, with *s* being the spin of the particle.

The Boltzmann equation that determines the evolution of the single-particle distribution function also receives a few important modifications: In the case of spin-1/2 particles interacting via binary elastic scattering, it reads
(248)$$\begin{array}{cc} k\cdot \partial f(x,k,s) & =\frac{1}{4}\int d\Gamma_{1}\;d\Gamma_{2}\;d\Gamma^{\prime }\;d\bar{S}\;W\left[f\left(x+{∆}_{1}-∆,k_{1},s_{1}\right)f\left(x+{∆}_{2}-∆,k_{2},s_{2}\right)\right.\\ \; & \;\left.-f\left(x,k,\bar{s}\right)f\left(x+{∆}^{\prime }-∆,k^{\prime },s^{\prime }\right)\right]\;, \end{array}$$
which has to be contrasted with the standard Boltzmann equation (Equation 76). Note that we defined the measure $d\Gamma_{i}\equiv dK_{i}dS_{i}\left(k_{i}\right)$ and assumed classical statistics to obtain a simpler expression, although quantum statistics can be included straightforwardly [122].

There are two things that are noteworthy about Equation (Equation 248). First, one of the distribution functions in the loss term depends not on $s^{\mu }$, but another spin vector ${\bar{s}}^{\mu }$ that is integrated over[note 17]. The second, and arguably more important, modification lies in the appearance of the shifts in the position of the particles given by ${∆}_{1}^{\mu }-{∆}^{\mu }$, ${∆}_{2}^{\mu }-{∆}^{\mu }$, and ${∆}^{\prime \mu }-{∆}^{\mu }$. These shifts are uniquely determined by the kinematics of the collision, in the sense that they depend on the momenta and spins of the particles participating in the collision, for explicit expressions, see [122]. Furthermore, they induce a *nonlocal* part of the collision term, implying that the particles no longer scatter at the same space-time point. The reason for this is rather intuitive: if a collision is local, i.e., all particles collide at the same space-time point $x^{\mu }$, one can shift the coordinate system such that there is no orbital angular momentum at the time of the collision. Since the total angular momentum is conserved, this implies that there can also be no net spin exchange in such a collision. In a nonlocal collision, however, there is always a nonzero amount of orbital angular momentum, which thus can be converted into spin. This nonlocality of the collision term is what allows the spin degrees of freedom to equilibrate and hence determines the approach of the spin potential $\Omega_{0}^{\mu \nu }$ to its global-equilibrium value $\varpi^{\mu \nu }.$

We further remark that, at this point, a new length scale enters the theory beside the mean free path and the fluid-dynamical scale, namely the Compton wavelength $\lambda_{C}∼\hbar /m∼\left|∆\right|$, which is on the order of the interaction range and thus much smaller than $\lambda_{\mathrm{mfp}}$. The ordering of scales in the problem is then $\lambda_{C}\ll \lambda_{\mathrm{mfp}}\ll L$, and in addition to the usual Knudsen number $\mathrm{Kn}\equiv \lambda_{\mathrm{mfp}}/L$ one can introduce a *quantum Knudsen number*$\kappa \equiv \lambda_{C}/L\ll \mathrm{Kn}$, which characterizes how well the underlying wavepackets can be approximated as point particles.

The local-equilibrium distribution function $f_{\mathrm{eq}}$ is defined by the requirement that the *local* part of the collision term vanishes, leading to the solution
(249)$$f_{\mathrm{eq}}\left(x,k,s\right)=\exp\left(\alpha_{0}-\beta_{0}E_{k}-s\frac{\hbar }{2m}\epsilon^{\mu \nu \alpha \beta }\Omega_{\mu \nu }k_{\alpha }s_{\beta }\right)\;.$$
It is important to note that requiring the distribution function to make the *full* collision term vanish (and not just the local part) leads to the additional requirements $\partial_{(\mu }\beta_{\nu )}=0$ and $\Omega_{0}^{\mu \nu }=\varpi^{\mu \nu }$, which are indeed the conditions for *global* (and not just local) equilibrium [127,156].

When moving on to construct dissipative spin hydrodynamics from such a quantum kinetic theory, the method of moments introduced in Section 5.4 can be applied, albeit with a few modifications. After splitting the distribution function into equilibrium and dissipative parts, $f_{k}=f_{\mathrm{eq}}+\delta f_{ks}$, one has to account for the structure of $\delta f_{ks}$ both in momentum and spin space. In particular, in the case of spin-1/2 particles, one has to consider two sets of irreducible moments,
(250a)$$\begin{array}{cc} \rho_{r}^{\mu_{1}\cdots \mu_{\ell }} & \equiv \int d\Gamma E_{k}^{r}k^{\langle \mu_{1}}\cdots k^{\mu_{\ell }\rangle }\delta f_{ks}\;, \end{array}$$
(250b)$$\begin{array}{cc} \tau_{r}^{\mu ,\mu_{1}\cdots \mu_{\ell }} & \equiv \int d\Gamma E_{k}^{r}s^{\mu }k^{\langle \mu_{1}}\cdots k^{\mu_{\ell }\rangle }\delta f_{ks}\;, \end{array}$$
which differ by the number of spin vectors. It is, in this case, sufficient to only consider moments with zero or one power of $s^{\mu }$ since the distribution function is at most linear in the spin vector [156]. This situation changes in the case of particles of spin 1 (or higher), where additional moments with more spin vectors have to be included [127,234]. The equations of motion for the moments $\tau_{r}^{\mu ,\mu_{1}\cdots \mu_{\ell }}$ can be obtained along the same lines as Equations ([Disp-formula FD150a-entropy-26-00189]), but we do not list them explicitly for the sake of brevity.

In kinetic theory for spin-1/2 particles, a possible representation of the spin tensor is given by
(251)$$S^{\lambda \mu \nu }=-\frac{1}{2m}\int d\Gamma k^{\lambda }\left(\epsilon^{\mu \nu \alpha \beta }k_{\alpha }s_{\beta }+\frac{\hbar }{2m}k^{[\mu }\partial^{\nu ]}\right)f\left(x,k,s\right)\;.$$
Ignoring the second term in parentheses, which does not enter the equation of motion (Equation 246), we can express the dissipative part of the spin tensor as
(252)$$\begin{array}{cc} \delta S^{\lambda \mu \nu } & \equiv -\frac{1}{2m}\int d\Gamma k^{\lambda }\epsilon^{\mu \nu \alpha \beta }k_{\alpha }s_{\beta }\delta f_{ks}\\ \; & =-\frac{1}{2m}\epsilon^{\mu \nu \alpha \beta }{∆}^{\lambda \rho }\left[u_{\alpha }\tau_{1,(\langle \beta \rangle ,\rho )}+\frac{1}{3}{∆}_{\rho \alpha }\left(m^{2}\tau_{0,\beta }-\tau_{2,\beta }\right)+\tau_{0,\beta ,\alpha \rho }\right]\;, \end{array}$$
where we used the Landau-type matching condition $u_{\lambda }\delta S^{\lambda \mu \nu }=0$. From this expression, it is clear that the irreducible moments that are connected to the spin tensor are given by[note 18]
(253)$$p^{\mu }\equiv \tau_{0}^{\langle \mu \rangle }\;,\;z^{\mu \nu }\equiv 2\tau_{1}^{\left(\langle \mu \rangle ,\langle \nu \rangle \right)}\;,\;q^{\lambda \mu \nu }\equiv \tau_{0}^{\langle \lambda \rangle ,\mu \nu }\;.$$
Here, $\tau_{2}^{\mu }$ is not listed because it can be related to the other quantities through the matching condition [156]. After employing the equations of motion for the irreducible moments $\tau_{r}^{\mu ,\mu_{1}\cdots \mu_{\ell }}$, one finds
(254a)τpp˙〈μ〉+pμ=e(0)ϵμναβuνΩαβ−ϖαβ+O(KnRe−1),
(254b)τzz˙〈μ〉〈ν〉+zμν=O(KnRe−1),
(254c)τqq˙〈λ〉〈μν〉+qλμν=−d(2)β0σρ〈μϵν〉λαρuα+O(KnRe−1).
Here, $e^{\left(0\right)}$ and $d^{\left(2\right)}$ are transport coefficients which determine the Navier–Stokes limit of the dissipative components of the spin tensor, while $\tau_{p}$, $\tau_{z}$, and $\tau_{q}$ denote the respective relaxation times. We remark that the coefficients $e^{\left(0\right)}$ and $d^{\left(2\right)}$ are determined solely by the nonlocal part of the collision term, whereas the relaxation times arise solely from local collisions.

## Data Availability

Not applicable.

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
