# Peer review of "Theories of Relativistic Dissipative Fluid Dynamics"

_entropy, 2024, doi:10.3390/e26030189_

Round 1

Reviewer 1 Report

Comments and Suggestions for Authors

The authors present an extensive review of various approaches for constructing theories of relativistic dissipative fluid dynamics. The review is well structured, with a concise and didactical Sec. II on preliminaries to establish notation, conventions, and key ideas and principles that will be highly appreciated by readers. The review is exceedingly well written. Rather than just summarizing results from the literature from a 30,000 foot perspective (to which the authors resort only at the end of each Chapter, in order to point readers to specific recent developments), in the body of each Chapter the authors go into sufficient technical detail to allow practitioners among their readership to re-derive important results and check their own work. This makes this review very valuable and will ensure its long-term impact on the community.

I found surprisingly few typos and minor errors but would like to point out two significant omissions which the authors would to well to address. Let me start by listing a few suggested minor corrections:

On p. 6, 4 lines below Eq.(8), I believe that a full stop is missing and the word "Throughout" should start a new sentence and be capitalized.

On p. 8, at the end of the paragraph following Eq.(21), a full stop is missing.

On p. 12, after Eq.(31), can the authors biefly state the rationale for eliminating gradients of \beta_0 in favor of gradients of p_0 and \alpha_0?

On page 20, at the end of the paragraph below Eqs.(71), the authors may consider adding "(as it should be because the equilibrium state maximizes the entropy)".

On p. 24, the first sentence of the paragraph starting below Eq.(78) states that the relativistic BE also emerges from QFT. I suggest adding to the list of references supporting this statement the key papers by Arnold, Moore and Yaffe (https://arxiv.org/abs/hep-ph/0209353 (2003)) and by Mrowczynski and Heinz (Annals Phys. 229 (1994) 1).

The two significant omissions allude to above refer to the discussion of anisotropic fluid dynamics (Sec. VII.A) and the lack of any discussion of Maximum Entropy (ME) hydrodynamics (which deserves its own Section VII.A'). Let me first note that what sets both of these approaches apart from the rest of the paper is their ambition to extend the validity of the hydrodynamic approach deeper into domains of evolution far away from local equilibrium, by giving up on expansions in gradients, Knudsen or inverse Reynolds numbers and replacing them by physically motivated non-perturbative parametrizations of the (possibly large) dissipative terms in T^{\mun\nu} and N^\mu and their evolution equations, as well as testing their performance for systems with known microscopic kinetic dynamics. This should be pointed out explicitly. 

Second, I take issue with the following statement on p. 52, middle of the second paragraph: "To our knowledge, the case where all dissipative corrections to the non-equilibrium reference state are taken into account has so far not been studied, although some attempts to incorporate them have been made [173]." While technically correct it paints the real situation much too negatively: the only dissipative correction that is still missing in practical implementations is the diffusion current and its couplings to the remaining dissipative currents; all 6 dissipative corrections to T^{\mu\nu}, however, are fully included (https://arxiv.org/abs/2101.02827, https://arxiv.org/abs/1803.01810), and this theory, when imbedded into a hybrid model with a additional microscopic kinetic module for the late hadronic freeze-out stage, has been shown (in [168] which is already cited) to be phenomenologically highly successful at high energies where baryon diffusion can be ignored (see also https://arxiv.org/abs/2311.03306). In any case, the era where \delta\tilde f_k corrections are neglected (as stated by the authors) is long behind us. Correspondingly, the last paragraph of Sec. VII.A should be significantly rewritten.

Finally, a brief additional Sec. VII.A' should be dedicated to a description of "ME hydrodynamics" (https://arxiv.org/abs/2101.01130, https://arxiv.org/abs/2307.10769, https://arxiv.org/abs/2312.04760). It generalizes anisotropic hydrodynamics (which in practical applications treats only the large shear and bulk viscous pressures generated by rapid longitudinal expansion non-perturbatively, but all other dissipative flows perturbatively) to generic far-off-equlibrium dynamical states where all dissipative flows (including diffusion) must be treated non-perturbatively. It is based on a momentum-moment expansion of the Boltzmann equation where the moment hierarchy is truncated by evaluating the coupling terms between the hydrodynamic and non-hydrodynamic moments with a "maximum entropy" ansatz for the distribution function which maximizes the information entropy (i.e. which is the most likely one) given the (large) dissipative flows in the hydrodynamically evolved T^{\mu\nu} and N^\mu. While formally worked out in full generality, practical implementations still consider only a subset of the dissipative flows. Tests of the ME hydro scheme in practical situations where this restriction is justified by symmetries of the flow proved highly successful even in situations where the Knudsen and inverse Reynolds number are much larger than 1. This review should not end without a discussion of this exciting development.

Of course, this new Sec. VII.A' should then be added to the description of the content of Sec. VII in the last paragraph of the Introduction.

I hope the authors agree that these suggestions will contribute to the longevity of this otherwise outstanding review.

Reviewer 2 Report

Comments and Suggestions for Authors

This review offers an analysis of relativistic dissipative fluid dynamics, a topic that finds extensive use in the area of high-energy nuclear physics. Additionally, it presents a comparison of the current theories in this field.

The authors discuss various theories of dissipative fluid dynamics. They begin the review with easy concepts such as conserved quantities, conservation laws, ideal fluid dynamics, matching conditions, constitutive relations, and causality and stability, which is adequate for a review. Then they introduce Knudsen and Reynolds numbers, which are quantities used to define the validity of fluid dynamics and will later be used in various theories to describe the fluid, which range from Navier-Stokes, Chapman-Enskog, Israel-Stewart, BDNK, and others.

I understand that the work is well written and substantiated, making it a good review for experts in the description of fluid dynamics for systems in extreme energy conditions.

Hence, I am of the opinion that the review is appropriate for publishing in its current form.

Reviewer 3 Report

Comments and Suggestions for Authors

This is a competent review that serves as a good reference anyone
going into the subject of relativistic hydrodynamics can use to learn the
basics of the status of the field and find references with more detailed
discussion.    There are a couple of points I wish the authors revisit
before publication:

a)   Section IID:  The power counting the authors propose is actually
incomplete:  There is the Knudsen/Reynolds scale sensitive to gradients, but
there is also a "thermal/microscopic" scale that measures fluctuations.
The fact that these scales are different, through related, is evident from the
fact that thermodynamic fluctuations at equilibrium are non-zero even when
the mean free path vanishes.       

It is also worth mentioning that the kinetic
theory derivation in section V already assumes this scale vanishes.
To be fair the expansion around this scale is still in its infancy.  Some works
(Jain and Kovtun) seem to suggest totally new transport coefficients, unobtainable from the gradient expansion, arise in this scale.
They also suggest once this new scale is included hydrodynamics loses much of
its universality.   On the other hand, experimental data (small hadronic systems,
cold atoms, even everyday phenomena like the "Brazilian nut effect/granular
convection) seems to suggest hydrodynamics is much more robust and universal
than that, indeed more universal than the introduction to this work gives
it credit for.   This is still an outstanding puzzle.
This work should include these issues.

b)  Lagrangian hydrodynamics, and its
expansion as an EFT has not been mentioned at all in this review (except a brief mention within the spin section).   It would
be great to have some mention of it as a general approach

c)  Kubo formulae are actually not specific to Zubarev's approach but can
serve as a _definition_ of transport coefficients.   A lot of the results
the authors calculate with the Boltzmann equation can actually be shown to   
arise from Kubo's formula to leading order, see the papers by
Hosoya and later Jeon and Arnold,Moore and Yaffe.
(note that corrections from this leading order is still a matter of
controversy, and this is related to point (a) regarding thermal fluctuations).

It is also far from clear that this tree-level equivalence extends to the
modified Chapman-Enskog expansion described in section V.C.
Note that the compatibility condition is related to the difficulty of the
Boltzmann approach to mainain Lorentz invariance and number-changing processes.  This is a problem specific to the Boltzmann prescription, in
fact QFT was constructed explicitly to solve this issue.

I think the role of Kubo formulae in the definition of hydrodynamics
needs to be mentioned in the introduction together with
conservation laws and "long" wavelength behavior.   An explanation of how
fluctuation-dissipation fits here would also be useful

d)  A small point but it might be worth noting that the significance of
the Reynlods number in this work is very different from its usual definition
in terms of turbulence (though quantitatively they are similar), and it might
be worth specifying it for the reader.

Round 2

Reviewer 1 Report

Comments and Suggestions for Authors

The changes made by the authors address my concerns satisfactorily, and the revised manuscript is ready for publication.

Author Response

We thank the referee for his positive assessment.

Reviewer 3 Report

Comments and Suggestions for Authors

The authors have answered the specific points I have raised, but I have a general comment regarding the introduction that I would like to address before the paper is published.

The paper starts with
-----------------------------------------------------------

Fluid dynamics is an effective theory for the long-wavelength, low-frequency limit of any given many-body system. The fluid-dynamical equations of motion are based on very general principles: they describe the conservation of energy, momentum, and, possibly, charge quantum numbers in the system. As an effective theory, fluid dynamics is applicable if there is a clear separation between the microscopic scales, which are characteristic of the dynamics of the fluid’s constituents, and the macroscopic scales, which characterize the spatio-temporal variation of the fluid-dynamical variables, i.e., energy density, charge density, and fluid velocity. This scale separation is usually quantified by the so-called Knudsen number, the ratio of a typical microscopic to a typical macroscopic time or length

-----------------------------------------

Yet, in the reply to my report the authors concede, in a margin at the end of the introduction, that in fact there are two interrelated but distinct length scales, and the study of the effect of this second scale is still it's infancy.

This also means that the regime of the applicability of the theory is for now not understood (which is certainly true, and hydrodynamics in small systems makes this issue clear at the experimental stage).  What does "long" and "slow" mean?  WIth respect to what?  The Knudsen number answers the question as far as gradients are concerned, but not as far as stochasticity is concerned.   The authors themselves contradict the first paragraph of the review in the additions they made in section 2.4

Furthermore, the first sentence "hydrodynamics is an effective theory" begs the question of "effective theory of what?".   The generality of hydrodynamics means that it should apply generally to any quantum field theory within the particular regime of applicability (non-relativistic appearances of solids, jellies etc. rely on broken symmetries which in the long wavelength limit are incompatible with causality).   

In this case the transport coefficients should have a QFT definition, which they do, via the Kubo formulae.   Yet the Kubo formula is still relegated to a paragraph in eq. 6.

All this would be forgivable except  that the above issues are widely misunderstood in the field of nuclear physics, with misconceptions like "hydrodynamics is the limit of transport" abounding in literature and conference presentations.   The answers of the authors are fine, and I think the review is complete as it is (also because the authors are correct when they say that topics like stochastic hydrodynamics are still strongly in development) but I would like the authors to revise the first paragraphs of the introduction to make these issues clear because at this stage I would say the first paragraphs are misleading.     Both the definition of the EFT and its regime of applicability needs further elaboration.

Author Response

We disagree with the referee that we contradict ourselves. In fact, in the introduction we are saying that the Knudsen number is a ratio of a typical microscopic and  typical macroscopic scale. At that point we did not specify the microscopic scale. We did not imply that it is the mean free path, in fact, it could well be the inverse temperature, which is the scale which governs fluctuations and besides is also the only other microscopic scale in strongly interacting (conformal) theories, where there is no concept of quasiparticles that collide with each other. In fact, there could even be other microscopic scales that the referee did not yet mention in his report, for instance the Compton wavelength that appears in the discussion of spin hydrodynamics in Sec. 7.4, or the vorticity scale (which we also did not discuss in this review).

To keep the presentation concise, we decided not to mention fluctuations (or the other microscopic scales mentioned above) in the first paragraph of the introduction at all.

Nevertheless, we added some additional sentences at the end of the introduction, and we also modified the paragraph at the end of Sec. 2.4, to make it even more clear that this review does not contain a discussion of fluctuations in hydrodynamical theories.

We hope that with these revisions and clarifications, the review is acceptable for publication.

Round 3

Reviewer 3 Report

Comments and Suggestions for Authors

The authors could be clearer in the introduction,but I'll leave it to them.

The paper is suitable for publication.